# Validation of a stereological method for estimating particle size and density from 2D projections with high accuracy

Jason Seth Rothman[1]*, Carolina Borges-Merjane[2¤], Noemi Holderith[3], Peter Jonas[2], R. Angus Silver[1]*

1 Department of Neuroscience, Physiology and Pharmacology, University College London, London, United Kingdom, 2 Cellular Neuroscience, Institute of Science and Technology Austria, Klosterneuburg, Austria, 3 Laboratory of Cellular Neurophysiology, Institute of Experimental Medicine, Budapest, Hungary

¤ Current address: Biozentrum of the University of Basel, Basel, Switzerland
* j.rothman@ucl.ac.uk (JSR); a.silver@ucl.ac.uk (RAS)

**Data Availability Statement:** All relevant data are within the paper and its Supporting Information files.

## Abstract

Stereological methods for estimating the 3D particle size and density from 2D projections are essential to many research fields. These methods are, however, prone to errors arising from undetected particle profiles due to sectioning and limited resolution, known as 'lost caps'. A potential solution developed by Keiding, Jensen, and Ranek in 1972, which we refer to as the Keiding model, accounts for lost caps by quantifying the smallest detectable profile in terms of its limiting 'cap angle' ($\phi$), a size-independent measure of a particle's distance from the section surface. However, this simple solution has not been widely adopted nor tested. Rather, model-independent design-based stereological methods, which do not explicitly account for lost caps, have come to the fore. Here, we provide the first experimental validation of the Keiding model by comparing the size and density of particles estimated from 2D projections with direct measurement from 3D EM reconstructions of the same tissue. We applied the Keiding model to estimate the size and density of somata, nuclei and vesicles in the cerebellum of mice and rats, where high packing density can be problematic for design-based methods. Our analysis reveals a Gaussian distribution for $\phi$ rather than a single value. Nevertheless, curve fits of the Keiding model to the 2D diameter distribution accurately estimate the mean $\phi$ and 3D diameter distribution. While systematic testing using simulations revealed an upper limit to determining $\phi$, our analysis shows that estimated $\phi$ can be used to determine the 3D particle density from the 2D density under a wide range of conditions, and this method is potentially more accurate than minimum-size-based lost-cap corrections and disector methods. Our results show the Keiding model provides an efficient means of accurately estimating the size and density of particles from 2D projections even under conditions of a high density.

**Funding:** Funding for JSR and RAS was from the Wellcome Trust (203048; 224499; https://wellcome.org/). RAS is in receipt of a Wellcome Trust Principal Research Fellowship (224499). Funding for CBM and PJ was from Fond zur Förderung der Wissenschaftlichen Forschung (V 739-B27 Elise-Richter Programme to CBM, Z 312-B27 Wittgenstein Award to PJ; https://www.fwf.ac.at). PJ received funding from the European Research Council (ERC; https://erc.europa.eu) under the European Union's Horizon 2020 research and innovation programme (grant agreement no. 692692). NH was supported by a European Research Council Advanced Grant (ERC-AG 787157). The funders had no role in study design, data collection and analysis, decision to publish, or preparation of the manuscript.

**Competing interests:** The authors have declared that no competing interests exist.

**Abbreviations:** T, thickness of tissue section (transmission microscopy) or focal plane ($\rho_z$); D, 3D diameter of a particle; $\mu_D \pm \sigma_D$, mean and standard deviation of 3D particle diameters; $CV_D$, $\sigma_D / \mu_D$; u.d., unit diameter, length normalised to $\mu_D$ (e.g. $T/\mu_D$); planar, T < 0.1 u.d.; thin, T ≈ 0.3 u.d.; thick, T ≥ u.d; d, observed 2D diameter of a particle's projection; $d_{min}$, minimum 2D diameter of a sample of particles [47]; $h_{min}$, minimum penetration depth of a sample of particles [42]; $\delta_{min}$, minimum 2D diameter of a given particle (z-stack analysis); $d_{area}$, equivalent-area 2D diameter: $d_{area} = 2(area/\pi)^{1/2}$; $d_{short}$, $d_{long}$, short and long-axis 2D diameter; $d_{geometric}$, $(d_{short} \cdot d_{long})^{1/2}$; $d_{avg}$, ½($d_{short}$ + $d_{long}$); $\mu_d \pm \sigma_d$, mean and standard deviation of 2D diameters; F(d), probability density of 3D diameters; G(d), probability density of 2D diameters; L(d), probability density of lost caps; nonblind, particle cap detection using adjacent images from a z-stack; blind, particle cap detection without guidance from adjacent images; θ, particle cap angle from section surface: sinθ = d/D where 0 ≤ θ ≤ 90; $\theta_{min}$, equivalent cap angle of $d_{min}$: $\sin\theta_{min} = d_{min}/D$; ϕ, lower limit of θ where 0 ≤ ϕ ≤ 90; $\phi_{cutoff}$, upper cutoff limit of when ϕ is determinable; $\mu_\phi \pm \sigma_\phi$, mean and standard deviation of ϕ; $CV_\phi$, $\sigma_\phi / \mu_\phi$; $d_\phi$, equivalent 2D diameter of ϕ: $d_\phi = \mu_D \cdot \sin\phi$; ζ, section z-depth over which particle center points are sampled; $Area_{xy}$, ROI xy-area over which particles are counted; VF, Particle volume fraction within a volume of interest; AF, Particle area fraction within a ROI; $N_{3D}$, Particle count within a volume of interest; $N_{2D}$, Particle count within a ROI; $\lambda_{3D}$, 3D particle density, $\lambda_{3D} = N_{3D} / Volume_{xyz}$; $\lambda_{2D}$, 2D particle density, $\lambda_{2D} = N_{2D} / Area_{xy}$; Ω, sum of projection overlaps for a given particle where Ω ≥ 0; ψ, upper limit of Ω, i.e. 0 ≤ Ω

## Introduction

Estimating the size and density of particles from their orthogonal projection, such as a 2D image, is a common stereological endeavour in the fields of biosciences, petrography, materials science and astronomy [1]. This approach is particularly valuable in the field of biosciences where the size and density of biological structures, such as cells and organelles, are often compared before and after drug perturbations, between normal and disease conditions, species, ages or critical periods of development [2]. Moreover, measures of particle size and density, or the equivalent measure of volume fraction (VF; see Abbreviations), form the basis of our understanding of a wide range of biological phenomena. For example, the density of synaptic vesicles near the active zone has been related to measures of synaptic plasticity [3–5], the density of cerebellar granule cells (GCs) and mossy fiber terminals (MFTs) has been used to estimate the amount of information transferred across the input layer of the cerebellum [6] and the amount of energy expenditure at the cellular and subcellular level [7]. Stereological measures are also commonly used to assay disease states, such as that of the kidney, brain, liver and lung [8–11]. Hence, stereological methods for estimating particle size and density have wide application and are of great practical utility.

Recent advances in high-resolution volumetric imaging have significantly improved the morphological information available about cells and tissue structure [12–15] making them ideal for 3D analysis. However, these technologies are expensive and full reconstructions are both labour and computationally intensive. The use of stereological methods for analysing a relatively small sample of 2D projections is therefore still the most time efficient and practical solution for most laboratories.

Stereological methods for estimating size and density have developed along two distinct approaches: a model-based approach that makes basic assumptions about the geometry of the particle of interest, e.g. Wicksell's transformation [16] and the Abercrombie correction [17] that assume a spherical geometry, versus a design-based approach that makes no assumption about particle geometry, e.g. the nucleator, rotator, physical and optical disector [10, 18, 19]. The ability to analyse particles with an arbitrary shape is one of the reasons design-based methods are often referred to as 'assumption free' or 'unbiased' and considered the superior approach [2, 11, 19–23]. However, design-based methods are not free of assumptions and may contain biases [24–28]. Moreover, design-based methods can be labour intensive and costly [26, 28–30] and are not appropriate for particles with a high density [18]. A high particle density occurs in many types of preparations, including vesicles in synapses [31, 32], granules in chromaffin and mast cells [33, 34], granule cells in the cerebellum and hippocampus [6, 35] and corneal epithelial basal cells [36]. Model-based methods, on the other hand, not only have the potential to be efficient, but can offer more information about particle size [37] and higher levels of accuracy [30] and can be applied to particles with either a low or high density.

A classic problem addressed by model-based stereological approaches is the estimation of particle size from a 2D projection, such as an image. This was first studied by Wicksell [16] and is known as Wicksell's corpuscle problem (Fig 1A). Wicksell's problem was to infer the true 3D diameter distribution (F(d)) of secondary follicles from their 2D diameter distribution (G(d)) measured from images of planar sections of the human spleen, where the thickness of the sections (T) was much thinner than the mean particle diameter ($\mu_D$). Wicksell's solution to this inverse problem was to use a model-based approach to derive an analytical solution for G(d) with respect to F(d) and then use a finite-difference unfolding algorithm to estimate F(d) from G(d).

A more common scenario than measuring particle profiles from a planar section is that of measuring particle profiles observed through a transparent section of thickness T (Fig 1B).

$\leq \psi$; $\chi^2$, sum of squared differences between data and fits (or simulations); $\Delta$, Parameter estimation error: % difference or difference from true value; $\mu_\Delta \pm \sigma_\Delta$, bias and (68%) confidence interval of a parameter's estimation error; $\rho_{xyz}$, Microscope resolution; $S_{xyz}$, Image/z-stack sample resolution.

The analytical solution for G(d) with respect to F(d) for $T \geq 0$ was derived by Bach [38] and can be described as the weighted sum of two components [39]: the diameter distribution of those particles with their center points contained within the section, in which case their G(d) = F(d), and the diameter distribution of those particles with their center points just above and below the section, by less than one radius, in which case their G(d) is a distorted version of F (d) as defined by Wicksell's analytical solution [16]. These latter particles whose north and south poles appear on the bottom and top of the section are known as 'caps'. Besides distorting the diameter distribution, caps also introduce a distortion of the apparent density, an effect known as overprojection, the split-cell error or the Holmes effect [17, 40–43].

A fundamental limitation of the Wicksell [16] and Bach [38] models, however, is that they assume all caps are resolvable. While this might be true for the largest caps with diameters on the order of F(d), the smallest caps are usually unresolvable, falling below the limits of resolution and contrast or blending in with their surrounding environment [44, 45]. Other caps might simply not exist if they fall off the surfaces of the sections or if the microtome fails to transect the particles during sectioning [27, 46–48]. Wicksell noted that lost caps could be accounted for by a post-hoc correction of his unfolding algorithm, whereby the missing probabilities of the smallest bins of F(d) are estimated via extrapolation from the smallest non-zero probability down to the origin. However, this approach is problematic since it relies on a small number of outlier observations. Indeed, when the number of observations within the smallest bins are insufficient, the unfolding algorithm can generate erroneous negative probabilities.

It was not until the 1970s that a key innovation for accounting for lost caps was developed by Keiding et al. [49] whereby lost caps are defined with respect to a lower limit of the 'cap angle' (cap-angle limit, $\phi$), the half angle subtended by a particle's cap from the particle's center (Fig 1). In this conceptual model, $\phi$ is independent of particle diameter, which is important since a distribution of limiting cap sizes can arise when particle size varies while specimen contrast, rather than microscope resolution, limits cap detection. Incorporating $\phi$ into the Wicksell-Bach model, Keiding et al. derived the following relationship between F(d) and G(d) for spherical particles:

$$G(d) = \frac{T}{\zeta}F(d) + \frac{d}{\zeta}\int_d^{\frac{d}{\sin\varphi}}\frac{F(y)}{\sqrt{y^2 - d^2}}dy \qquad\qquad \text{Eq 1}$$

where d is the 2D particle diameter, $y$ is the variable of integration and $\phi$ can vary from 0°, where no caps are lost, to 90°, where all caps are lost. Here, $\zeta$ is the mean axial length spanning from below to above the section that contains the center points of those particles observed within the projection (Fig 1), defined as follows:

$$\zeta = T + \mu_D\cos\phi \qquad\qquad \text{Eq 2}$$

As expected, Eq 1 reduces to Bach's analytical solution when $\phi$ = 0° and to Wicksell's analytical solution when $\phi$ = 0° and T = 0.

Another innovation of Keiding et al. [49] was to estimate both F(d) and $\phi$ from G(d) using a maximum likelihood estimation (MLE) algorithm, rather than using an unfolding algorithm, thereby providing a better quantification of $\phi$. Knowing $\phi$ is particularly useful since it can be used via Eq 2 to estimate the 3D particle density ($\lambda_{3D}$) from the measured 2D particle density ($\lambda_{2D}$) as follows [50] (S1 Appendix in S1 File):

$$\lambda_{3D} = \lambda_{2D}/\zeta \qquad\qquad \text{Eq 3}$$

This 'correction' method for estimating $\lambda_{3D}$ is potentially more accurate than using the classic Abercrombie correction [17], which assumes no caps are lost (i.e. $\phi$ = 0°), or the Floderus

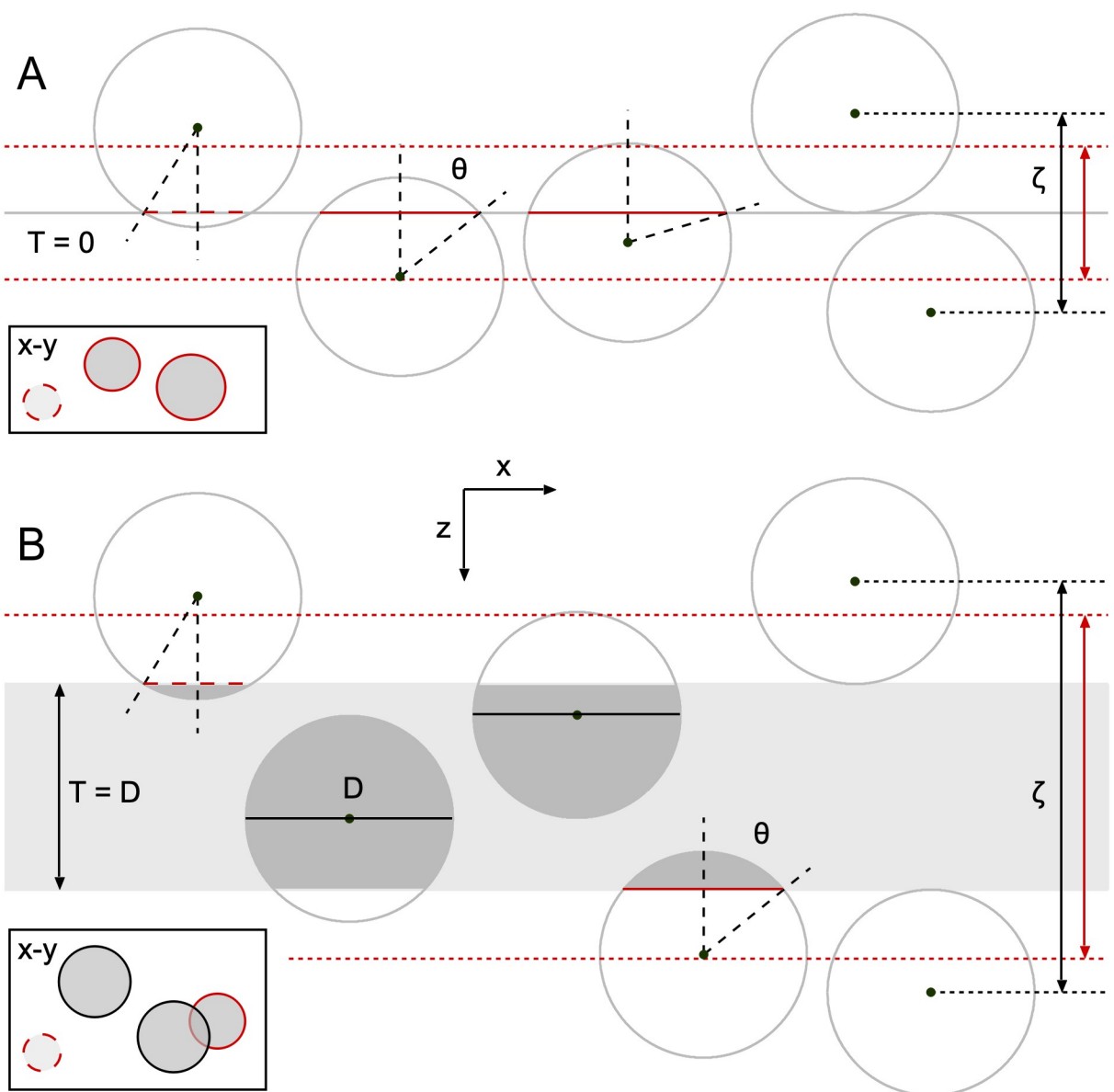

**Fig 1. Illustration of observed circular profiles when spherical particles are orthogonally viewed from above a planar or thick section. A.** Side view of a planar section (T = 0; gray line) transecting spherical particles (red solid lines). For simplicity, all particles have the same 3D diameter (D). Particles with their center above/below the section within a distance D/2 are observed as circular 'caps' in a horizontal projection (inset rectangle, top view, red circles) with apparent diameter d < D, where d = D·sinθ and θ is the cap angle (black dashed lines) that takes on values between 0˚, where d = 0, and 90˚, where d = D. Hence, those particles appearing within the projection have their center confined within a depth ζ = D (black double-headed arrow). However, due to experimental limitations, the smallest caps (with small θ) are not apparent, i.e. lost (red dashed lines). To account for lost caps, the Keiding model sets a minimum limit on θ (φ) such that caps are observed in the projection only if φ < θ < 90˚ [49]. In this case, ζ = D·cosφ (red double-headed arrow; Eq 2). For planar sections, there are no projection overlaps (inset) and the total area fraction (AF) of the projections approximately equals the 3D volume fraction (VF) of the particles so long as there are relatively few lost caps. Because all particle centers fall above/below the planar section, all particles are considered caps. **B.** Same as **A** for a thick section (T = D). In this case, particles with their center within the section have a circular projection with d = D (black circles) and ζ = T + D·cosφ. For thick sections and a high particle density, there are usually projection overlaps (inset) that make counting/outlining the projections more difficult; moreover, AF > VF, a condition known as overprojection.

[42] and Konigsmark [47] corrections that use the minimum cap penetration depth ($h_{min}$) or equivalent minimum cap diameter ($d_{min}$), both of which are likely to be outlier measures. Unfortunately, despite potentially providing the most accurate description of the lost-cap distribution via ϕ, the Keiding model has not been widely adopted nor validated. Validation of model-based approaches is important since they are based on simplifying assumptions of particle geometry [22, 24, 28, 51].

To investigate whether the Keiding model can provide a simple and accurate method for estimating the size and density of spherical particles such as synaptic vesicles and GCs, we used a distribution-based least-squares estimation (LSE) algorithm to systematically test the model's performance in estimating F(d) and ϕ from G(d) computed from 3D Monte Carlo simulations and electron-tomography (ET) reconstructions. Synaptic vesicles in MFTs and the nuclei and somata of GCs were chosen for the analysis since they contain a wide range of particle sizes and have high densities that are problematic for design-based stereological methods. This analysis confirmed the accuracy of the Keiding model in estimating the 'true' F(d) and ϕ over a range of conditions. However, the accuracy of estimated ϕ was limited by the sample size, spread of F(d) and the number (i.e. distribution) of lost caps. Finally, we tested the accuracy of Eq 3 for estimating $\lambda_{3D}$ from the measured $\lambda_{2D}$ using the same 3D simulations and ET reconstructions and found this method to be more accurate than the Abercrombie [17] and $d_{min}$ corrections [47] and the widely used disector method [18, 52]. To facilitate the adoption of the Keiding model in stereological applications, we provide an analysis workflow for estimating F(d), ϕ and $\lambda_{3D}$ from 2D projections and provide guidelines for optimising particle cap detection. Moreover, we incorporated our numerical solution of the Keiding model, including LSE curve-fit functions, into the open-source software toolkit package NeuroMatic [53] that works within Igor Pro (Key Resources).

## Results

### Properties of 2D diameter distributions from images of somata, nuclei and vesicles in the cerebellar cortex

To investigate the properties of G(d) over a range of experimental conditions, including particle size, section thickness, imaging technique and spatial resolution, we quantified the 2D diameters of GC somata and nuclei and MFT vesicles using confocal and transmission electron microscopy (TEM) images of cerebellar sections. These preparations encompassed both planar sections (where T $\ll$ $\mu_D$; Fig 1A) and thick sections (where T $\approx$ $\mu_D$; Fig 1B).

First, we computed G(d) for cerebellar GC somata from confocal images of rat brain sections from a previous study [6] (Fig 2A1). In these images, GC somata were visible due to Kv4.2 immunolabeling. However, because the GC somata had an opaque staining and were tightly packed together, there was a high probability of partial and complete overlaps of the 2D profiles, especially since the sections were not planar (T $\approx$ 1.8 μm; Table 1). To compute G(d), we drew outlines around the GC somata and computed a normalised histogram from the equivalent diameters of the areas of the outlines ($d_{area}$). As predicted by Wicksell [16], G(d) of the GC somata had an asymmetrical shape with a negative skew (Fig 2A2). However, the negative 'tail' of the distribution only extended to ~2 μm rather than 0 μm since we were unable to detect GC somata profiles with d < 2 μm (i.e. lost caps). In total, we computed 9 G(d) of GC somata that spanned 2–8 μm with a mean 2D diameter $\mu_d$ = 4.96–5.83 μm and standard deviation $\sigma_d$ = 0.61–0.90 μm (n = 3 rats, 2–3 cerebellar sections per rat, 494–638 diameters per G(d)).

To examine whether a more complete G(d) could be obtained from images where the resolution is higher, we computed G(d) for GCs in TEM images of mouse cerebellar sections (Fig

2B1). Because the sections for this preparation were ~60 nm thick, the sections were essentially planar ($T << \mu_D$; Table 1) with no overlap of 2D profiles. Here, we drew outlines around the outer contour of the GC nuclei rather than the somatic membrane since the nuclei were easier to identify due to their spotted appearance created by dark patches of heterochromatin. Similar to G(d) of the GC somata, G(d) of the GC nuclei had an asymmetrical shape negatively skewed (Fig 2B2), similar to that previously reported for rat hepatocyte nuclei in planar sections [55]. However, G(d) extended to 1 μm rather than 2 μm since smaller caps were easier to resolve in the TEM images compared to the confocal images. In total, we computed 26 G(d) of GC nuclei from 4 mice, 1–3 cerebellar sections per mouse, 6–7 TEM images per mouse. Comparison of G(d) within mice showed no significant differences (Kolmogorov-Smirnov, KS, test); hence, G(d) within mice were pooled. The resulting 4 nuclei G(d) spanned 1–6 μm with $\mu_d$ = 3.61–4.32 μm and $\sigma_d$ = 0.85–0.98 μm (416–519 diameters per G(d)). To allow a comparison of GC somata and nuclei across species, described further below, we computed $d_{area}$ of both the soma ($d_{soma}$) and nucleus ($d_{nucleus}$) of individual GCs using high-magnification TEM images of the same cerebellar sections of mice and found the $d_{soma}$-versus-$d_{nucleus}$ relation was well described by the following linear relation: $d_{soma} = 0.952 \cdot d_{nucleus} + 1.016$ μm (Pearson correlation coefficient (r) = 0.96, goodness-of-fit $R^2$ = 0.92; n = 175 GCs from 4 mice).

Finally, to investigate G(d) when the section thickness is comparable to the size of the particles, we measured 2D diameters of synaptic vesicles in MFTs in high-magnification TEM images of the same cerebellar sections of mice, where $T \approx \mu_D$ (Fig 2C1; Table 1). Because the vesicle membrane was not always apparent, we drew outlines around the outer contour of the vesicles rather than attempt to outline the inner or outer membrane leaflet. Moreover, we did not assume vesicle outlines were circular or oval, but rather followed the irregular contours of the vesicles that included membrane proteins, which are known to add at least 2 nm to the diameter of the vesicles [56]. Depending on the vesicle density and section thickness, vesicles aligned in the axial axis may show different degrees of overlap in the projection [55] (Fig 1B). Although our TEM images of vesicles in MFTs exhibited numerous overlaps, this did not necessarily preclude outlining the vesicles since they were semi-transparent. Interestingly, G(d) of MFT vesicles were quite different to that of GC somata and nuclei, having a Gaussian shape with no negative skew and a large number of lost caps with d < 30 nm (Fig 2C2), similar to that previously reported for synaptic vesicles in thick sections [55, 57]. In total, we computed 8 G(d) of MFT vesicles for 4 mice, 1 section per mouse, 2 MFTs per section, 152–428 vesicles per MFT. Comparison of the 8 G(d) showed the majority were significantly different from each other (KS test), even within mice comparisons, supporting previous findings of synapse-to-synapse variation of synaptic vesicle size [55, 57]. Analysis of the 8 G(d) of MFT vesicles showed all had Gaussian shapes spanning 23–82 nm with $\mu_d$ = 43.1–47.3 nm and $\sigma_d$ = 4.2–6.2 nm. These results highlight how the shape of G(d), especially the cap tail, depends on the relative size of the particles compared to the section thickness and the imaging method used to acquire the projections.

## Exploration of the effects of section thickness and lost caps on G(d) of the Keiding model

To better understand how section thickness and lost caps affect the shape of G(d), we computed numerical solutions of the Keiding model (Eq 1) for different section thicknesses (T) and cap-angle limits (φ). To do this, we assumed F(d) was a Gaussian distribution (Eq 5) with normalised mean, i.e. a mean of one unit diameter (u.d.) and standard deviation of 0.09 u.d., to mimic the coefficient of variation of our experimental data ($CV_D = \sigma_D/\mu_D \approx 0.07$ for GC somata, 0.08 for GC nuclei and 0.09 for MFT vesicles). For a planar section (T = 0 u.d.; Fig 1A)

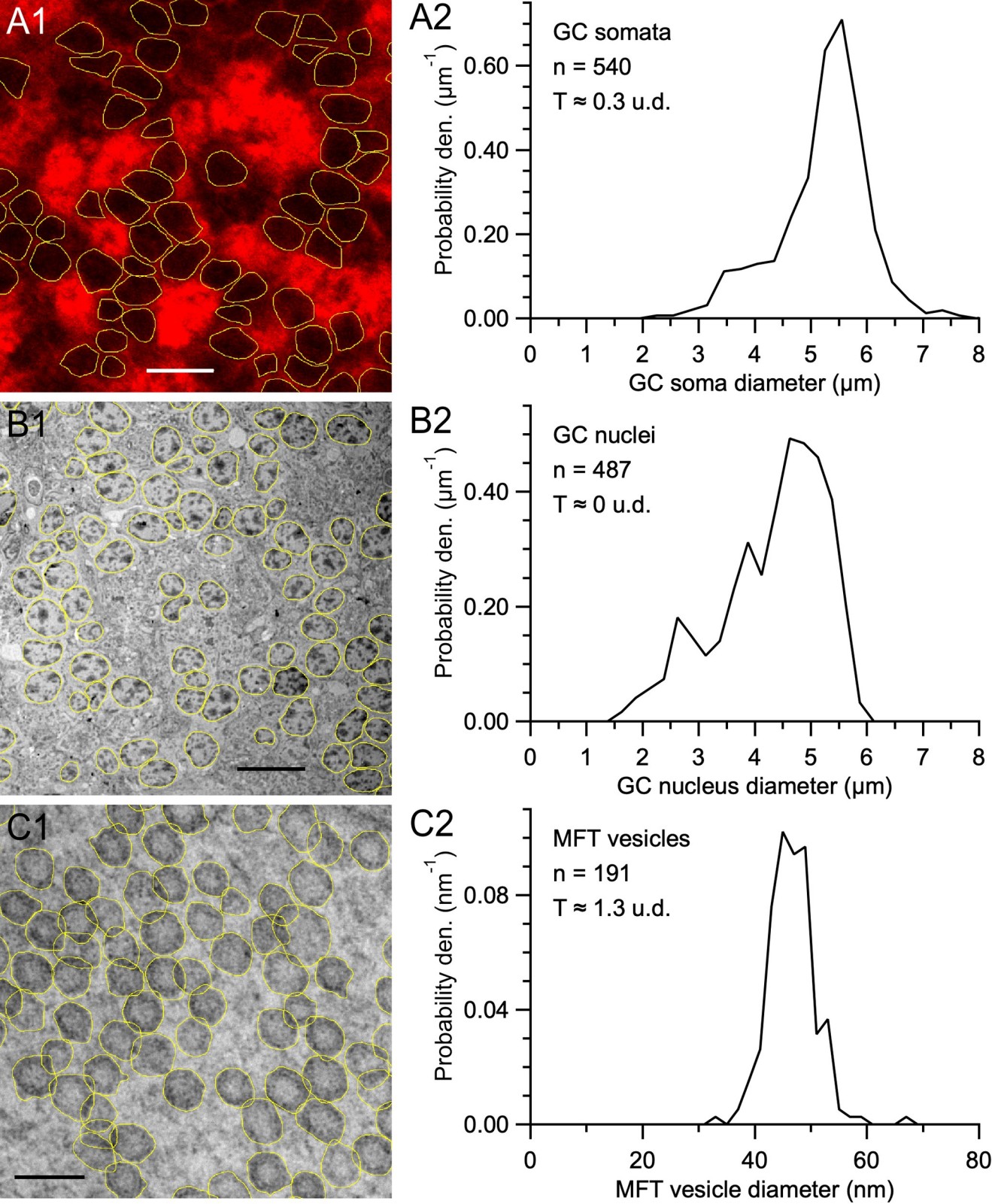

**Fig 2. Computing G(d) of GC somata and nuclei and MFT vesicles. A1.** Confocal image of a cerebellar section of a wild type (WT) rat (P30; from [6]). GC somatic plasma membranes were delineated via immunolabeling for Kv4.2. Outlines were drawn around those GC somata that were well delineated (yellow) and an equivalent diameter was computed from the area of each outline ($d_{area}$). T ≈ 1.8 μm. Scale bar 10 μm. Image ID R5.SL2.1. **A2.** Probability density of 2D diameters (G(d)) computed from the GC soma diameters (0.30 μm bins) measured from the image in **A1** plus 1 other image from the same z-stack. **B1.** Low-magnification TEM image of a cerebellar section of a WT mouse (P31). Outlines were drawn around the outer contour of visually identified GC nuclei (yellow). T ≈ 60 nm. Scale bar 10 μm. Image ID M18.N2.51. **B2.** G(d) computed from the GC nucleus diameters (0.25 μm bins) measured from the image in **B1** plus 6 other images from the same mouse. **C1.** High-magnification TEM image of a MFT in the GC layer of the same mouse in **B1**. Outlines were drawn around the outer contour of the synaptic vesicles (yellow). T ≈ 60 nm. Because vesicles are semi-transparent, 2D overlaps do not necessarily preclude drawing their outline or counting. Scale bar 80 nm. Image ID M18.N2.03. **C2.** G(d) computed from the vesicle diameters (2 nm bins) measured from the TEM image in **C1**. For **A1**, **B1** and **C1** only a subregion of the outline analysis is shown.

with no lost caps (ϕ = 0˚), the numerical solution of G(d) had a skewed distribution with pronounced tail descending to 0 u.d. (where d < D; Fig 3A). In contrast, when the section thickness equaled the mean particle diameter (T = 1 u.d.; Fig 1B), G(d) had a larger peak and less pronounced tail, since most particles (57%) had their widest central region falling within the section (where d = D). To examine how G(d) is affected by lost caps, we computed the same numerical solutions for ϕ > 0˚. When ϕ = 20˚, the cap tails of G(d) now descended to 0.3 u.d. (Fig 3B), resembling those of the GC somata and nuclei G(d) in Fig 2A2 and 2B2. When ϕ = 40˚, the cap tails of G(d) descended to 0.5 u.d. (Fig 3C). Interestingly, when ϕ = 70˚, the cap tails of G(d) were no longer apparent and G(d) ≈ F(d) (Fig 3D). In fact, G(d) ≈ F(d) for ϕ ≥ 55˚, in which case the two distributions were nearly indistinguishable (Fig 3E). The absence of cap tails in these G(d) is reminiscent of the G(d) for the MFT vesicles in Fig 2C2.

To quantify the relationship between G(d), F(d) and lost caps, we computed the distribution of lost caps (L(d)) for ϕ = 5–90˚ and compared L(d) to F(d) (Fig 3F). Results showed L(d) had a negatively skewed Gaussian shape with tail descending to 0 u.d. The upper limit of L(d) showed a dispersion, rather than a hard cutoff, since a variation in particle size in combination with a fixed ϕ resulted in limiting caps of different size. For ϕ < 55˚, there was a clear

**Table 1. Experimental conditions for confocal, TEM and ET imaging.**

| Particle | Image | #A | Prep | $T_{tissue}$ | T | $\rho_{xy/z}$ | $S_{xy/z}$ |
|---|---|---|---|---|---|---|---|
| GC soma | Confo | 3R | Fix | 40,000 | 1832† | 326† | 200–400 |
| | | | | | (0.32) | 1010 | 500–1000 |
| GC nucleus | TEM | 4M | HPF | 60 | 60 | 0.45 | 29–47 |
| | | | | | (0.01) | - | - |
| GC nucleus | TEM-z | 1M | Fix | 17,000 | 40 | 0.45 | 40 |
| | | | | | (0.01) | - | 40‡ |
| MFT vesicle | TEM | 4M | HPF | 60 | 60 | 0.45 | 0.5–0.7 |
| | | | | | (1.31) | - | - |
| MFT vesicle | ET10-z | 1M | Fix | 182 | 6.7 | 4.7 | 1.14 |
| | | | | | (0.15) | 6.7§ | 0.63‡ |
| MFT vesicle | ET11-z | 1M | Fix | 138 | 5.1 | 3.6 | 1.14 |
| | | | | | (0.12) | 5.1§ | 0.53‡ |
| Units | | | | Nm | Nm | nm | nm/px |
| | | | | | (u.d.) | nm | nm |

Confo: confocal z-stack. TEM-z: TEM z-stack [60]. ET10-z and ET11-z: ET z-stacks. #A: number of rats (R) or mice (M). Prep: tissue preparation. Fix: chemical fixation. HPF: High-pressure freezing. $T_{tissue}$: thickness of cerebellar section. T: thickness of tissue section or focal plane. ρ: microscope resolution (top: lateral; bottom: axial). S: image or z-stack sample resolution (top: lateral; bottom: axial). px: pixels. †T = 1.8·$\rho_z$, where 1.8 is a compensation factor for z-shrinkage [6]. †$\rho_{xy}$ and $\rho_z$ computed via Eqs 1 and 2 of [54] where the laser excitation wavelength is 543 nm, objective's numerical aperture is 0.85 and refractive index in air is 1.0. §Assuming z-elongation $\rho_z$ = 1.4·$\rho_x$ (Eq 13). ‡$S_z$ estimated from data (S6B and S9B Figs in S1 File).

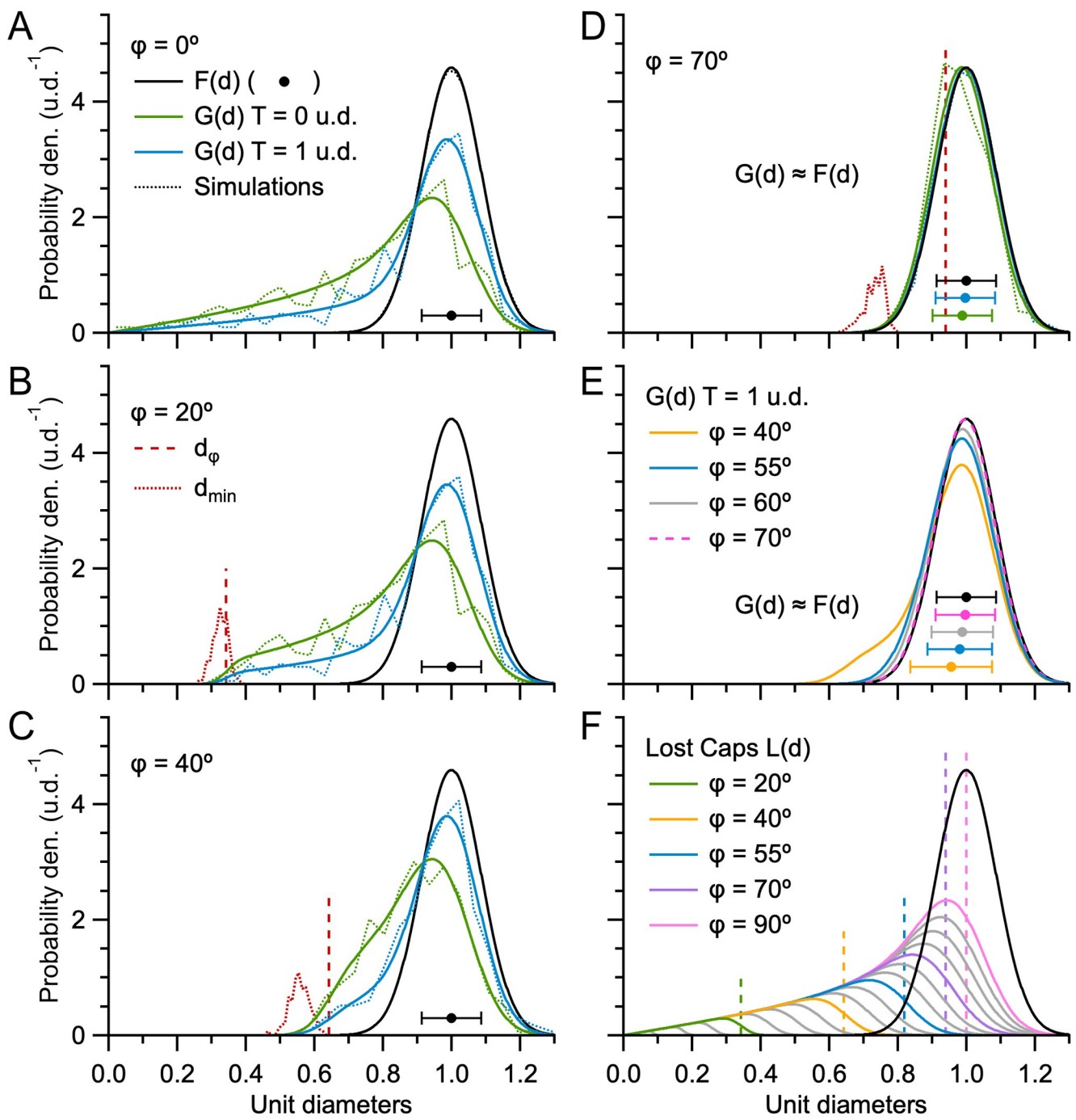

**Fig 3. Effect of section thickness and lost caps on G(d). A.** Probability density of 2D diameters (G(d)) computed via Eq 1, where $\phi = 0°$ and the probability density of 3D diameters (F(d)) is a Gaussian distribution with normalised mean (Eq 5; $\mu_D \pm \sigma_D = 1.00 \pm 0.09$ u.d.; black line and circle ± error bars). For T = 0 u.d. (green line) conditions are that of Wicksell's model [16] (Fig 1A) and for T = 1 u.d. (blue line) conditions are that of Bach's model [38] (Fig 1B). Because no caps are lost, both G(d) have tails extending to d = 0 u.d. Dotted lines denote G(d) of equivalent Monte Carlo simulations computed from ~500 diameters using 0.04 u.d. bins. **B–D.** Same as **A** for $\phi = 20, 40$ and 70˚. Here, the tails of G(d) are limited to 0.3, 0.5 and 0.7 u.d. For $\phi = 70˚$, G(d) ≈ F(d) (green and blue circles denote $\mu_d \pm \sigma_d$) since most caps are lost. Comparison of the distribution of the minimum observed 2D diameter ($d_{min}$, red dotted line, computed from simulations for both T = 0 and 1 u.d., probability densities scaled by 0.07) to $d_\phi = \mu_D \cdot \sin\phi$ (vertical red dashed line) shows $d_{min} < d_\phi$, especially at larger $\phi$. **E.** For $\phi > 55˚$, G(d) ≈ F(d). **F.** Distribution of lost caps, L(d), for $\phi = 5–90˚$ in steps of 5˚ (gray and colored solid lines; Materials and Methods) compared to F(d) (black line). Vertical dashed lines denote $d_\phi$. Note, L(d, $\phi = 90˚$) = G(d, $\phi = 0˚$).

separation between L(d) and F(d), i.e. nearly all lost caps had a diameter smaller than those that define F(d). For $\phi > 55°$, on the other hand, L(d) overlapped F(d), in which case there was a large number of lost caps with diameters equivalent to those that define F(d). At the most extreme condition where all caps are lost ($\phi = 90°$) L(d) spanned from 0 to the largest diameter of F(d). Hence, at these larger $\phi$, L(d) and F(d) cannot be delineated by diameter size.

## Estimating the 3D particle size and cap-angle limit using the Keiding model

Next, we tested the Keiding model's capacity to estimate F(d) and $\phi$ from G(d). To do this, we used a 3D Monte Carlo simulation package to compute virtual projections of spherical particles for planar and thick sections, creating lost caps according to the Keiding model (i.e. a cap was removed from the projection when its $\theta < \phi$; S1 Fig in S1 File; Materials and Methods). To make a direct comparison to our numerical solutions of Eq 1, we matched F(d) of the simulated particles to that used in the numerical solutions. Moreover, we set the number of measured 2D diameters per projection (~500) to match the average sample size of our experimental G(d) of GC somata and nuclei, the two datasets with the largest number of measured diameters. From the 2D diameters of the simulated particles, we computed G(d) for T = 0 and 1 u.d. and $\phi = 0, 20, 40$ and $70°$, all of which matched their equivalent numerical solution (Fig 3A–3D). Using an LSE routine, we then curve fitted Eq 1 to the simulated G(d) and compared the resulting estimates of $\mu_D$, $\sigma_D$ and $\phi$ to their true values. The comparison showed that, when true $\phi < 55°$, the Keiding model accurately estimated $\mu_D$, $\sigma_D$ and $\phi$ with only a small positive bias for $\sigma_D$ (Fig 4A and 4C). On the other hand, when $\phi > 55°$, estimates of $\mu_D$ and $\sigma_D$ were less accurate, with positive and negative biases, respectively, and estimates of $\phi$ often had a large negative bias (Fig 4B and 4C). In this case, the LSE routine had difficulty estimating true $\phi$ since G(d) ≈ F(d) as $\phi$ approached $90°$. The similarity between G(d) and F(d) at true $\phi > 55°$ was greatest for thick sections in which case fitting a simple Gaussian function (Eq 5) to G(d), which is equivalent to curve fitting the Keiding model with $\phi = 90°$, where all caps are lost, resulted in similar estimates of $\mu_D$ and $\sigma_D$, as did simply using the 2D measures $\mu_d$ and $\sigma_d$ as estimates. Moreover, the estimates of $\mu_D$ and $\sigma_D$ of the three approaches all showed relatively small biases (< 2 and 5%, respectively, for T = 1 u.d.). An overall comparison revealed that thick sections were moderately better for estimating $\mu_D$ and $\sigma_D$, and planar sections were moderately better for estimating $\phi$. Repeating the error analysis for G(d) computed from ~2000 diameters gave qualitatively similar results, except $\mu_D$, $\sigma_D$ and $\phi$ had smaller biases and confidence intervals, where the confidence intervals followed a $1/\sqrt{n}$ relation (S4 Fig in S1 File).

While the above results show the Keiding model works well in estimating $\mu_D$, $\sigma_D$ and $\phi$, estimates were most accurate when $\phi < 55°$ (Fig 4C). However, this upper limit of $\phi$ ($\phi_{cutoff}$) is dependent on the spread of F(d) and the number of measured diameters, both of which were set in our Monte Carlo simulations to match our experimental data ($CV_D = 0.09$, ~500 diameters). For a wider F(d) and/or smaller number of diameters, accurate parameter estimation will be limited to a smaller range of true $\phi$, and vice versa. To quantify this effect, we computed $\phi_{cutoff}$ over a range of $CV_D$ (0.04–0.17) and number of diameters (n ≈ 200–2000) for our Monte Carlo simulations (T = 0 u.d.; Fig 4C and S4 Fig in S1 File, insets). Results showed $\sin\phi_{cutoff}$ fit well to a bivariate polynomial with respect to $CV_D$ and $1/\sqrt{n}$ (Eq 8). Next, we investigated whether this $\phi_{cutoff}$ equation could be used to test the accuracy of estimated $\phi$ using estimated $\mu_D$ and $\sigma_D$, since the estimation errors of $\mu_D$ and $\sigma_D$ are relatively small even when true $\phi > \phi_{cutoff}$. Results showed that indeed this was possible with relatively small adjustments to the $\sin\phi_{cutoff}$ relations (Eq 9). Hence, if one requires an accurate estimate of $\phi$, for example

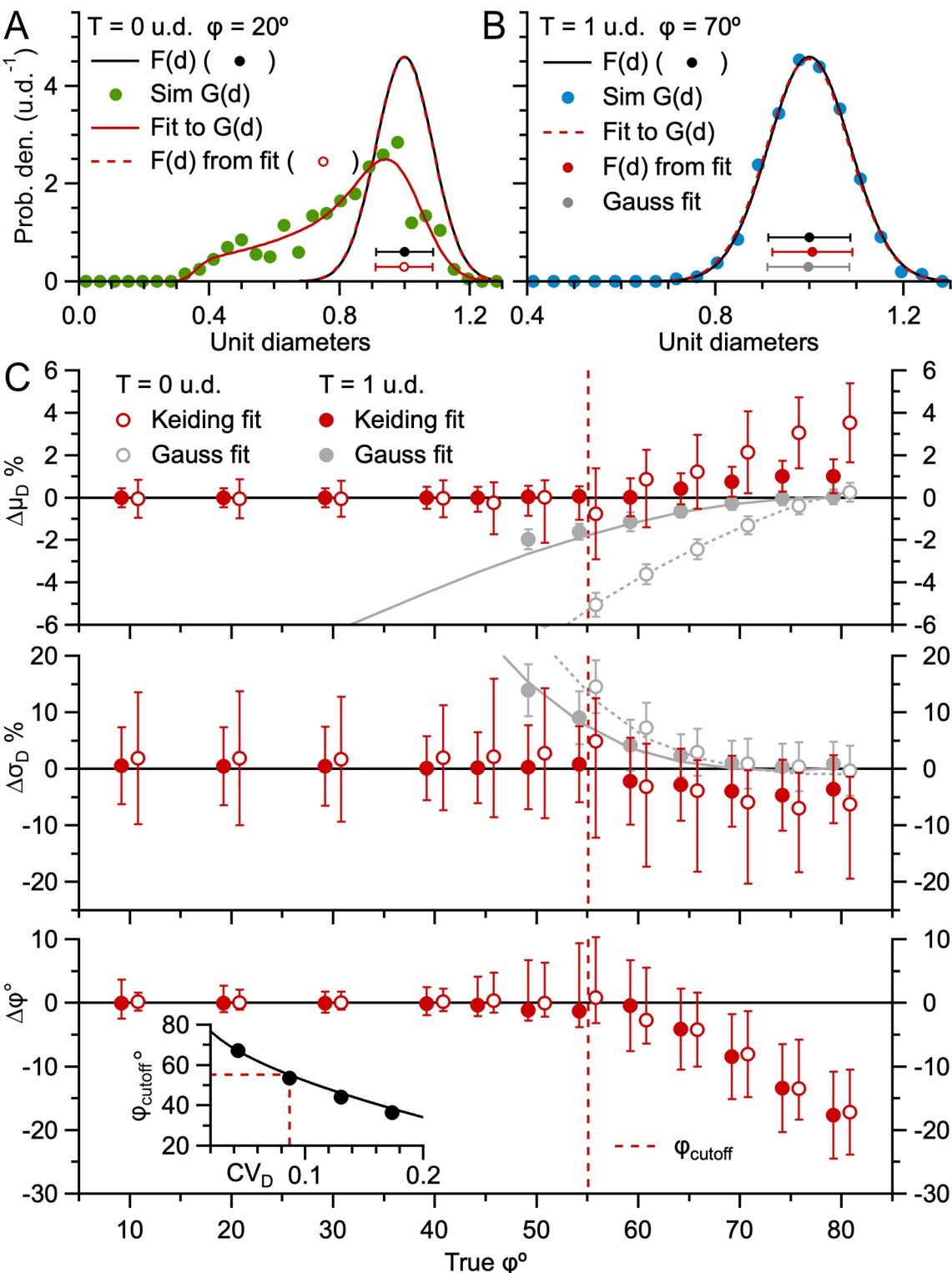

**Fig 4. The Keiding model accurately estimates F(d) and φ from G(d) for true φ < φ_cutoff (simulations). A.** Curve fit of [Eq 1](red solid line) to G(d) of the simulation in [Fig 3B](where T = 0 u.d. and φ = 20° (green circles; ~500 diameters). F(d) derived from the fit (red dashed line and circle) matches the true simulation F(d) (black line and circle) and fit φ matches true φ ($\Delta\mu_D = -0.1\%$, $\Delta\sigma_D = +0.1\%$, $\Delta\phi = +1°$). **B.** Same as **A** for G(d) of the simulation in [Fig 3D](where T = 1 u.d. and φ = 70° (blue circles). Although there is a good match between estimated and true F(d), estimation errors $\Delta\mu_D$ and $\Delta\sigma_D$ are larger than those in **A** and estimated φ < true φ ($\Delta\mu_D = +0.9\%$, $\Delta\sigma_D$

= -4.4%, Δϕ = -9˚). **C.** Average estimation errors $\Delta\mu_D$, $\Delta\sigma_D$ and Δϕ of Keiding-model fits to simulated G(d), as in **A** and **B**, for true ϕ = 10–80˚, T = 0 and 1 u.d., $CV_D$ = 0.09 (red open and closed circles; $\mu_\Delta \pm \sigma_\Delta$ for 100 repetitions per ϕ). Red dashed lines denote $\phi_{cutoff}$ (~55˚; Eq 8) above which G(d) ≈ F(d) and true ϕ becomes indeterminable. For comparison, results are shown for Gaussian fits to the same G(d) (Eq 5; gray circles) and 2D statistics $\mu_d \pm \sigma_d$ (gray lines). Data shifted ±0.8˚ to avoid overlap. Asymmetrical error bars indicate skewed distributions (Materials and Methods). Inset: $\phi_{cutoff}$ vs. $CV_D$ for simulations (black circles) and Eq 8 (black line; n = 500 diameters). See S1–S5 Figs in S1 File.

when computing particle density as discussed further below, and one only has estimates of $\mu_D$ and $\sigma_D$, then Eq 9 can be used as an accuracy test, i.e. by requiring estimated ϕ < estimated $\phi_{cutoff}$ (S5 Fig in S1 File). Although the fit error of ϕ can also be used as a measure of accuracy, ~10% of our curve fits to simulated G(d) for T = 0 u.d. and true ϕ > 55˚ showed small fit errors (< 1˚) but large estimation errors (|Δϕ| > 5˚). Hence, a combination of the fit error and Eq 9 can be used as an accuracy test for ϕ.

## Overlapping particle projections in thick sections

A potential source of error of the Bach [38] and Keiding [49] models for thick sections is that the models assume particles near the bottom of the section can be identified and outlined in a projection as well as those toward the top of the section. While this may be true for thin sections, it is unlikely to be true for thick sections with overlapping particle projections [2, 58]. To investigate what effects overlapping projections might have on the ability to accurately estimate F(d) and ϕ from G(d), we used simulations to compute the sum of 2D projection overlaps between a given particle and particles higher in a thick section, where particles had a random distribution and a modest to high VF (0.15–0.45). Results of the analysis showed that, as expected, particles toward the bottom of the section experienced more overlaps in a projection, and this effect was greater for thicker sections and higher particle density (Fig 5A). To investigate what effect projection overlaps might have on the shape of G(d), we set an upper limit to the amount of projection overlaps an observable particle can have, i.e. particles with overlaps above the set limit (ψ) were considered hidden from view (lost) and excluded from the projection. Results of these 'semi-transparent' particle simulations showed that, for thick sections and high particle density, a large number of particles near the bottom of the section were excluded from the projection, thereby reducing the effective section thickness (T) for a given projection (Fig 5B; ψ = 0.25). Nevertheless, excluding bottom-dwelling particles from the projection had minimal effect on the shape of G(d), and therefore had little effect on estimates of F(d), as long as the top-dwelling particles were simulated as transparent, i.e. small degrees of projection overlap did not interfere with counting the projections or computing their size (Fig 5C). This is in contrast to when particles were simulated as opaque such that overlapping projections reduced the number of observed projections and increased their size; in this case, opaque particles created a positive skew in G(d) (Fig 5D). However, the distortions in G(d) produced only modest changes in estimates of F(d). Hence, these results indicate overlapping projections of semi-transparent and opaque particles in thick sections create only small biases in estimates of F(d). However, the overlapping projections do have the potential to reduce the effective section thickness (Fig 5B) and therefore affect the estimates of the 3D density (see below).

## Validation of the Keiding model for estimating particle size using 3D reconstructions

In the previous section, we used Monte Carlo simulations to test the Keiding model's capacity to estimate F(d) and ϕ from G(d). However, while the simulations included variation (i.e.

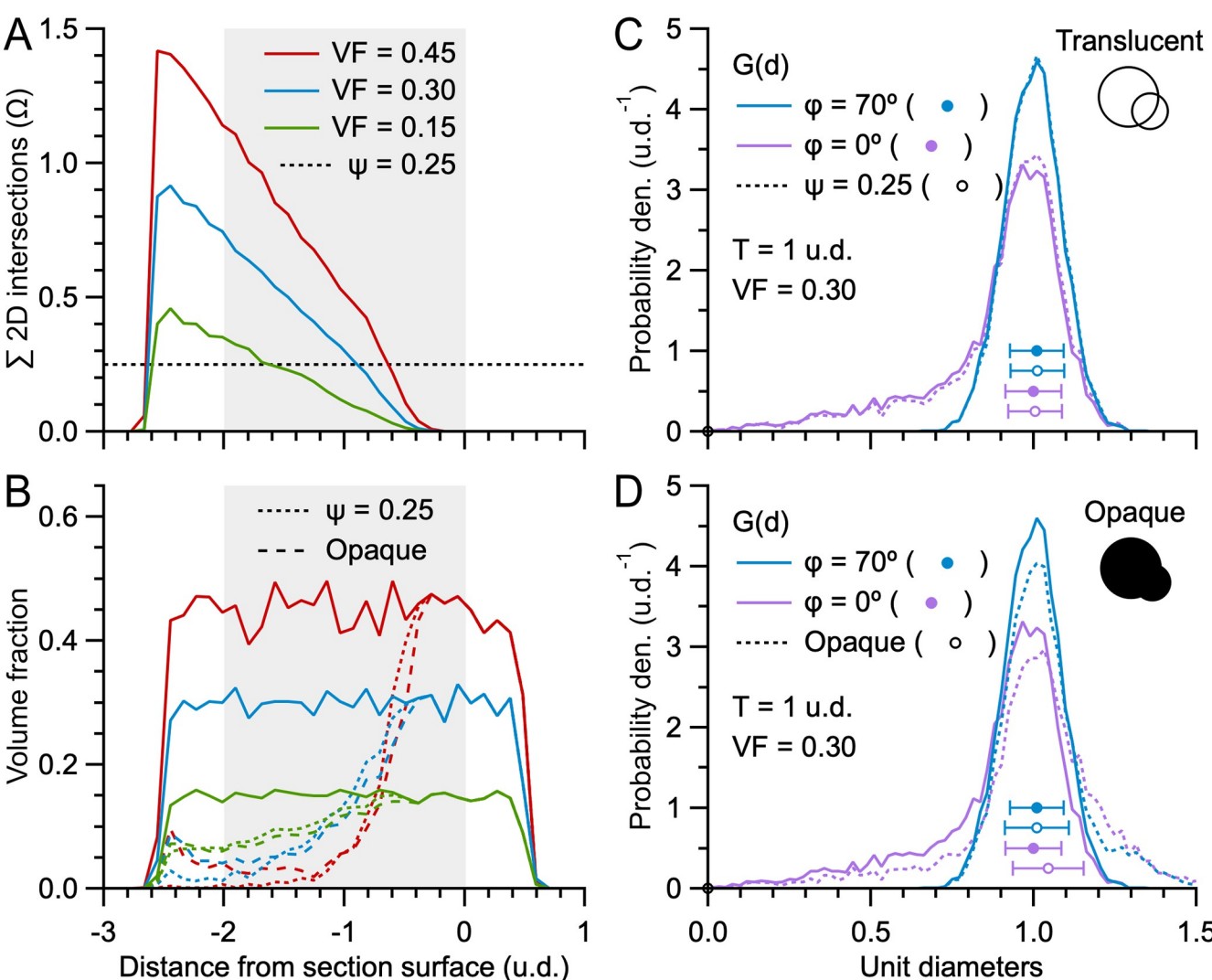

**Fig 5. Effects of projection overlaps for thick sections. A.** Sum of 2D projection overlaps (Ω) as a function of distance from the surface of a simulated section for particle VF = 0.15, 0.30 and 0.45 (green, blue and red lines) where Ω > 1 indicates a particle's projection is likely to be completely overlapping with projections of other particles closer to the section surface. Histograms were computed using particle z-center points and 0.1 u.d. bins. Gray background denotes section thickness (T = 2 u.d.). The distributions extend 0.5 u.d. below the section because of caps (Fig 1B). Black dotted line denotes upper limit ψ = 0.25 for **B** and **C**. For simplicity, ϕ = 0˚. **B.** VF of those particles appearing in a simulated projection as a function of distance from the surface of each section (T = 2 u.d.) for transparent particles (solid lines; control, ψ = ∞), semi-transparent particles (dotted lines; a particle is removed from the projection if its Ω > 0.25) and opaque particles (dashed lines; bottom-dwelling particles are merged with top-dwelling particles if -1 < α < 0; Materials and Methods). VF = 0.15, 0.30 and 0.45 as in **A**. The effect of semi-transparent and opaque particles is to reduce the effective T. Histograms were computed using particle z-center points and 0.1 u.d. bins; counts were converted to VF using the equivalent bin volume (geometry Area$_{xy}$ multiplied by bin z-width) and particle diameter distribution. For simplicity, ϕ = 0˚. **C.** Comparison of simulated probability density of 2D diameters (G(d); 0.02 u.d. bins) for transparent and semi-transparent particles (ψ = ∞ and 0.25; solid vs. dotted lines) shows little difference for two extreme conditions of lost caps (ϕ = 0˚ and 70˚; purple and blue). In these simulations, projection overlaps did not affect estimates of their size (inset). Circles and error bars denote Keiding-model fit parameters μ$_D$ ± σ$_D$. T = 1 and VF = 0.30. **D.** Comparison of simulated G(d) for transparent and opaque particles (solid vs. dotted lines). For opaque particles, overlapping projections with -1 < α < 0 were treated as a single projection with larger area (inset), thereby creating a positive skew in G(d). For **A–D**, average histograms were computed from 20 sections, ~500 particles per section.

stochasticity) in particle location and size, they lacked variation normally associated with experimental data such as that due to finite spatial resolution, limited contrast, irregular particle shape and blending with surrounding material (e.g. intracellular/extracellular proteins). Hence, to address this shortcoming, we tested the accuracy of the Keiding model using a

volumetric ET z-stack of clusters of MFT vesicles, since this allowed us to directly measure F(d) and ϕ, as well as G(d). First, F(d) was measured by outlining individual vesicles across multiple planes of the z-stack (Fig 6A; ET11). From the outlines of a given vesicle, equivalent radii were computed as a function of z-depth (Fig 6B) and curve fitted to an ellipse (Eq 10), resulting in estimates for the vesicle's 3D diameter (D), z-axis center point ($z_0$) and elliptical eccentricity (E) (S6 Fig in S1 File). Aligning the vesicle profiles at their centers revealed the profiles overlaid each other and fit well to a semi-circle (Fig 6B). From the measured 3D diameters, we computed $\mu_D \pm \sigma_D$ = 42.9 ± 3.4 nm and F(d) (Fig 6D; n = 233). To determine the shape of F(d), we fit the distribution to Gaussian, chi and gamma functions (Eqs 5–7) and found closely over-lapping fits (S7 Fig in S1 File). Moreover, we found parameter $f > 70$ for the chi and gamma fits indicating Gaussian-like distributions. Repeating the same analysis for F(d) derived from another ET z-stack (ET10) and another study of MFT vesicles [5] produced similar results. Hence, F(d) of the MFT vesicles was well described by a Gaussian function, which is consistent with the findings of a 3D analysis of vesicles in hippocampal CA1 excitatory synapses [59].

Next, we measured ϕ by computing the axial extent of each vesicle profile, which only par-tially extended to the north and south poles (Fig 6B and S6A Fig in S1 File). The missing pro-files at the pole regions, i.e. lost caps, are likely due to limited resolution and poor contrast of orthogonally oriented membranes [45]. Plotting the minimum measured diameter ($\delta_{min}$) for the vesicle poles interior to the z-stack versus their 3D diameter revealed a linear relation that was well described by the Keiding model ($\delta_{min}$ = D·sinϕ; Fig 6C1). In contrast, the minimum of all measured diameters ($d_{min}$), which has been used as a correction for lost caps (S1 Appen-dix in S1 File), was a poor fit to the $\delta_{min}$-D relation. Converting all $\delta_{min}$ to ϕ values revealed a distribution that spanned 26–67˚ (Fig 6C2) with $\mu_\phi \pm \sigma_\phi$ = 42 ± 7˚ ($CV_\phi$ = 0.2; n = 403; Table 2) and was well described by a Gaussian distribution (Fig 6C3). Hence, this analysis revealed a source of variation not accounted for in the Keiding model, which assumes all parti-cles have the same ϕ. The ramifications of a variation in ϕ are explored in the next section.

Next, we computed G(d) from all measured 2D diameters (Fig 6D). Comparison of G(d) to F(d) revealed a negative skew in G(d) as expected for planar sections (estimated T ≈ 0.1 u.d.; Table 1). Finally, using the measured G(d), F(d) and ϕ, we tested the Keiding model by curve fitting Eq 1 to G(d), assuming a Gaussian F(d) (Eq 5), which our results above confirmed is a good assumption, and comparing the resulting estimated F(d) and ϕ to their measured 'true' values. Results showed the estimated F(d) and ϕ closely matched their measured values ($\Delta\mu_D$ = -0.2%, $\Delta\sigma_D$ = +0.6% and Δϕ = -3˚). Moreover, both the measured and estimated ϕ were below $\phi_{cutoff}$ (~63˚). Similar results were obtained by repeating the same 2D versus 3D analysis for another ET z-stack of MFT vesicles (ET10; $\Delta\mu_D$ = +0.5%, $\Delta\sigma_D$ = -0.3% and Δϕ = -0.1˚; Table 2; S8 Fig in S1 File).

To examine whether our validation analysis of the Keiding model holds for larger particles, we repeated a similar 2D versus 3D analysis of a recently published TEM z-stack of cerebellar GC nuclei (S9A–S9D Fig in S1 File; Table 3; [60]). Results of the 2D analysis of individual nuclei revealed a small estimated ϕ = 10˚ (where $\phi_{cutoff}$ ≈ 45˚) and an estimated F(d) that closely matched that computed from a 3D analysis using images from the same z-stack ($\Delta\mu_D$ = -1.3%, $\Delta\sigma_D$ = +8.0%). Hence, our 2D versus 3D analyses show that, even with the added vari-ability from experimental data, including variability in ϕ, the Keiding model accurately esti-mated F(d) and ϕ from G(d) with only small error when true $\phi < \phi_{cutoff}$, confirming our previous results from simulations (Fig 4C).

## Exploration of the Keiding model's fixed-ϕ assumption

Despite the impressive capacity of the Keiding model to estimate F(d) and ϕ of our experimen-tal data, close inspection of the curve fits to G(d) measured from our ET z-stacks showed

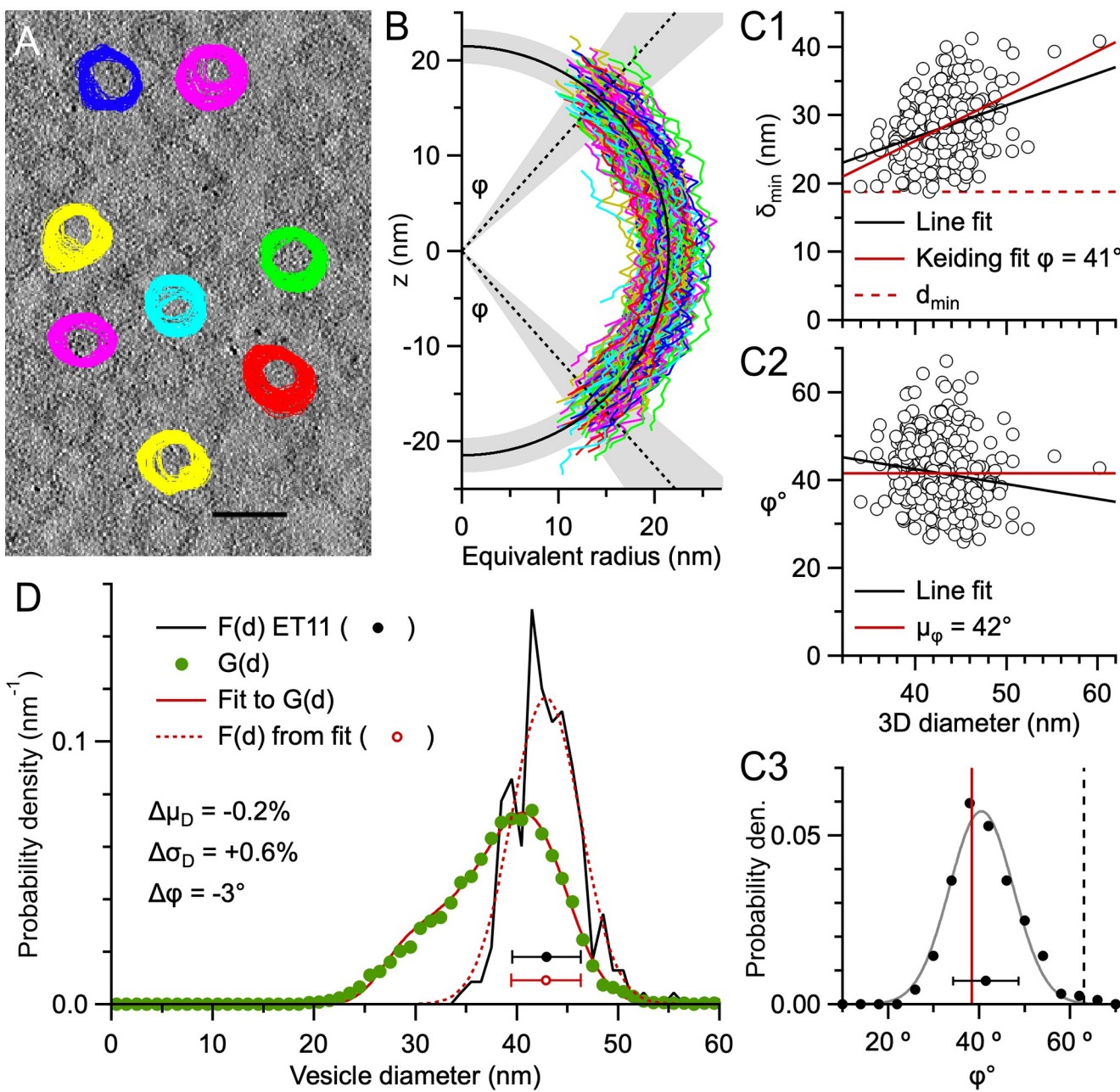

**Fig 6. The Keiding model accurately estimates F(d) and φ from G(d) for true φ < φ_cutoff (vesicles ET11). A.** One of 261 serial images of a 3D ET reconstruction (ET11) of a cerebellar MFT section 138 nm thick. 271 vesicles, including 101 caps, were tracked and outlined through multiple z-planes and their $d_{area}$ computed as a function of z-plane number ($z_\#$). This image shows outlines for 8 representative vesicles, overlaid with outlines from images above and below. Scale bar 50 nm. **B.** Vesicle xy-radius (½$d_{area}$) vs. z-depth (colored lines) with vesicle centers ($z_0$) aligned at z = 0; caps are not displayed. Black semi-circle and shading denote $\mu_D \pm \sigma_D$ = 42.9 ± 3.4 nm for all measured 3D diameters (D; n = 233). Black dotted lines and shading denote measured φ: $\mu_\phi \pm \sigma_\phi$ = 41.5 ± 7.2° (n = 403, measures of north and south poles, including caps). Parameters $z_0$ and D were estimated via a curve fit to Eq 10 and φ = $\sin^{-1}(\delta_{min}/D)$, where $\delta_{min}$ is the minimum $d_{area}$ at a given pole (S6A Fig in S1 File). Average fit E = 1.00 ± 0.16 (n = 233) where estimated $S_z$ = 0.53 nm (S6B Fig in S1 File). Fits to the smallest caps were not included (n = 38). **C1.** Minimum 2D diameter of a given vesicle ($\delta_{min}$) vs. D (circles; n = 403) with line fit (black line; $\chi^2$ = 5957, r = 0.4, $R^2$ = 0.1) and Keiding-model fit (red solid line; $\delta_{min}$ = D·sinφ; fit φ = 41.0 ± 0.3°; $\chi^2$ = 6105, r = 0.4, $R^2$ = 0.3). The smallest $\delta_{min}$ ($d_{min}$ = 19 nm; red dashed line) is a poor match to the data. **C2.** Same as **C1** but for φ = $\sin^{-1}(\delta_{min}/D)$ with line fit (black line; r = -0.1, $R^2$ = 0.02) and $\mu_\phi$ = 41.5° (red solid line). **C3.** Probability density (per °) of measured φ in **C2** (black circles; probability per degree) with Gaussian fit (gray line; Eq 5) and fit φ (red line) and $\phi_{cutoff}$ from **D** (black dashed line; ~63°; Eq 8). Note, the difference between the fit and measured φ (Δφ = -3°) can partially be accounted for by an estimated +1° discretization error of measured φ (S6D Fig in S1 File) and a -0.3° error of fit φ from assuming a fixed φ (S10C Fig in S1 File). **D.** Measured F(d) (black line and circle; 1 nm bins; see **B**) vs. G(d) (green circles; n = 13,914 outlines; 1 nm bins). A curve fit of Eq 1 to G(d) (red solid line; $\mu_D$ = 42.9 ± 0.1 nm, $\sigma_D$ = 3.4 ± 0.1 nm, φ = 38.4 ± 0.3°; T fixed to 0 nm) resulted in estimated F(d) (red dotted line and circle) nearly the same as measured F(d) and estimated φ nearly the same as $\mu_\phi$ (**C3**). See S6–S10 Figs in S1 File.

**Table 2. Summary of size and density analysis of cerebellar MFT vesicles.**

| Image | Assay | #M | $n_{size/den}$ | $\mu_D$ | $\sigma_D$ | $\phi$ | $\phi_{cutoff}$ | $\lambda_{3D}$ | VF | $VF_{AF}$ |
|---|---|---|---|---|---|---|---|---|---|---|
| TEM | 2D-B | 4 | 269 ± 48<br>332 ± 68 | 45.7±0.7 | 4.0±0.2 | > 44° | 38°±1° | | | |
| ET10-z | 3D-NB | 1 | 132<br>115 | 46.0 | 4.0 | 41† ±9° | | 8.6 | 0.45 | |
| | 2D-NB | | 7083<br>5023 | 46.2 | 4.0 | 41° | 60° | 8.7 | 0.45 | 0.46 |
| | 2D-B | | 751 | 46.4 | 4.1 | 63° | | | | |
| ET11-z | 3D-NB | 1 | 233<br>271 | 42.9 | 3.4 | 42† ±7° | | 11.0 | 0.47 | |
| | 2D-NB | | 13,914<br>13,914 | 42.9 | 3.4 | 38° | 63° | 11.0 | 0.46 | 0.46 |
| | 2D-B | | 889 | 43.0 | 3.5 | 56° | | | | |
| Units | | | | nm | nm | ° | ° | $\times 10^3 \ \mu m^{-3}$ | | |

Image: TEM (Fig 2C) or ET z-stack (Figs 6 and 11; S8 Fig in S1 File). Assay: 2D or 3D, blind (B) or nonblind (NB) particle detection. #M: number of mice. n: number of 2D or 3D diameters for size ($n_{size}$; top) and density ($n_{den}$; bottom) analysis. For 2D analysis, $\mu_D \pm \sigma_D$ and $\phi$ were computed via Keiding-model fits to G(d). $\phi_{cutoff}$ computed via Eq 9 for TEM analysis and Eq 8 for ET analysis. VF computed via Eq 4. $VF_{AF}$: VF = $K_v$·AF (Eq 14; $K_v$ = 1.09 and 1.07 for ET10 and ET11, respectively). For TEM analysis, values reported as mean ± SEM with respect to #M; it was not possible to estimate $\lambda_{3D}$. †Estimated discretization error of measured $\phi$ is +1° (S6D Fig in S1 File).

systematic deviation at the smaller diameters (d = 20–40 nm; Fig 6D; S8D and S10A Figs in S1 File. To test whether the discrepancies arose from the variation of $\phi$ within the sample of vesicles (Fig 6C3 and S8C3 Fig in S1 File), we used Monte Carlo simulations to replicate our ET10 z-stack analysis (S8 Fig in S1 File) and compared the resulting G(d) to the experimental G(d) for a fixed-$\phi$ model where all vesicles have the same $\phi$, as in the Keiding model, and a Gaussian-$\phi$ model where $\phi$ of each vesicle is randomly drawn from a Gaussian distribution ($\mu_\phi \pm \sigma_\phi$) whose shape and variation matched the measured $\phi$ ($CV_\phi$ = 0.2). Comparison of the two models confirmed that the small discrepancy between the Keiding-model fit and experimental G(d) at the smaller diameters can indeed be explained by the variation in $\phi$ (S10B1 and S10B2

**Table 3. Summary of size and density analysis of cerebellar GC somata and nuclei.**

| Image | #A | GC | Assay | $n_{size/den}$ | $\mu_D$ | $\sigma_D$ | $\phi$ | $\phi_{cutoff}$ | $\lambda_{3D}$ | VF | $VF_{AF}$ |
|---|---|---|---|---|---|---|---|---|---|---|---|
| Confo | 3R | S | 2D-B | 537 ± 12<br>301 ± 39 | 5.78±0.16 | 0.41±0.03 | 37° ±2° | 47°±1° | 3.2±0.2 | 0.32±0.01 | 0.32±0.02 |
| TEM | 4M | N | 2D-B | 471 ± 22<br>488 ± 22 | 4.80±0.17 | 0.39±0.01 | 20° ±1° | 44°±0° | 5.9†±0.4 | 0.34±0.01 | 0.33±0.01 |
| | | N→S | | | 5.60±0.16 | 0.35±0.01 | 29° ±1° | 49°±0° | † | 0.54±0.01 | |
| TEM-z | 1M | N | 3D-NB | 107<br>206 | 6.73 | 0.51 | - | | 2.0 | 0.31 | |
| | | | 2D-NB | 974<br>- | 6.63 | 0.52 | 6° | 48° | - | - | - |
| | | | 2D-B | 688<br>820 | 6.64 | 0.55 | 10° | 45° | 1.9 | 0.29 | 0.29 |
| Units | | | | | μm | μm | ° | ° | $\times 10^6 \ mm^{-3}$ | | |

Image: confocal (Confo; Fig 2A), TEM (Fig 2B) or TEM z-stack (S9 Fig in S1 File; [60]). GC soma (S), nucleus (N) and nucleus scaled to soma (N→S). #A: number of rats (R) or mice (M). n: average number of 2D diameters for size ($n_{size}$; top) and density ($n_{den}$; bottom) analysis. $\mu_D \pm \sigma_D$ and $\phi$ computed via Keiding-model fits to G(d). Estimated $\phi_{cutoff}$ computed via Eq 9. VF computed via Eq 4. $VF_{AF}$: VF = $K_v$·AF (Eq 14; $K_v$ = 0.71 and 0.99 for somata and nuclei). Values reported as mean ± SEM with respect to #A; somata data was weighted per slice. All particle detection was performed by blind analysis. †Somata $\lambda_{3D}$ = nuclei $\lambda_{3D}$.

Figs in S1 File). Moreover, the simulations allowed us to quantify the errors from assuming a fixed $\phi$ in our Keiding-model curve-fit routine, giving biases $\Delta\mu_D$ = +0.2 ± 0.5%, $\Delta\sigma_D$ = -0.8 ± 5.8% and $\Delta\phi$ = -0.4 ± 0.8˚ (average of ET10 and ET11 simulations; S10C Fig in S1 File). Hence, the Keiding-model fixed-$\phi$ assumption only introduced small biases into the estimates of $\mu_D$, $\sigma_D$ and $\phi$. To further explore the effects of assuming a fixed $\phi$ in the Keiding model, we computed errors $\Delta\mu_D$, $\Delta\sigma_D$ and $\Delta\phi$ from Keiding-model fits to simulated G(d), as just described, over a range of $\phi$ distributions ($\mu_\phi$ = 10–50˚; $CV_\phi$ = 0.2) for planar and thick sections (T = 0 and 1 u.d.; S11 Fig in S1 File). Results gave negligible estimation errors, except for conditions of a planar section and $\mu_\phi$ = 40–50˚, in which case errors showed modest biases. Hence, these results show that the Keiding-model fixed-$\phi$ assumption introduces only small errors when estimating F(d) and $\phi$ from G(d).

## Impact of blind versus nonblind particle detection in 2D images

For the size analysis of MFT vesicles in our 3D reconstructions, G(d) was computed from measurements of vesicles that were tracked through multiple images of the ET z-stack. This 'nonblind' approach provided information that made it easier to identify small caps, resulting in a tail of G(d) that descended to ~20 nm and $\phi \approx 40$˚ (Fig 6D and S8D Fig in S1 File). In contrast, G(d) of MFT vesicles in the first Results section were computed 'blind' without knowledge of a vesicle's 3D position in a z-stack. In this case, G(d) had a symmetrical Gaussian-like appearance with no tail (Fig 2C2). Hence, we wondered if the difference between the two G(d) could simply be explained by one analysis being blind and the other nonblind. To test this, we recomputed G(d) from our ET z-stacks using a blind analysis: images of the z-stack were analysed in random order, and vesicles were not tracked between adjacent images. Results showed G(d) computed blind had a symmetrical Gaussian appearance with no tail, and a similar appearance to F(d) (Fig 7A and 7B), thus confirming the difference in G(d) can be explained by a greater number of lost caps in the blind analysis. To quantify $\phi$ for the blind analysis, we simultaneously fit the Keiding model to G(d) computed blind and nonblind, sharing parameters for F(d), i.e. $\mu_D$ and $\sigma_D$. For ET11, results gave $\phi$ = 38.3 ± 0.5˚ for the nonblind analysis (similar to the analysis in Fig 6D) and $\phi$ = 55.5 ± 0.7˚ for the blind analysis. Hence, there was a 17˚ difference (bias) in $\phi$. For ET10, results gave a 22˚ difference in $\phi$. Note that, despite a large estimated $\phi$ for both blind analysis (where $\phi > \phi_{cutoff}$), which usually correlates with a large fit error (Fig 4C), the fit errors for this analysis were only ±1˚. The small fit errors are due to the sharing of parameters $\mu_D$ and $\sigma_D$ during the simultaneous fit, in which case good estimates of $\mu_D$ and $\sigma_D$ were achieved via G(d) computed nonblind, and $\phi$ was determinable for both blind and nonblind analysis. Besides the difference in $\phi$, the simultaneous fit also revealed G(d) $\approx$ F(d) for the blind analysis, with only a subtle difference between the two curves. These results highlight the difference between blind and nonblind particle detection, and that cap detection can be significantly improved by additional 3D information provided by a z-stack.

## Estimation of the 3D size of granule cell somata and nuclei

Having shown the Keiding model accurately estimates F(d) and $\phi$ from G(d), with the exception that $\phi$ is indeterminable when true $\phi > \phi_{cutoff}$ (Fig 4C), we curve fitted Eq 1 to the 9 G(d) computed from GC somata of rats (Fig 2A2), resulting in estimated $\mu_D$ = 5.48–6.32 μm, $\sigma_D$ = 0.35–0.49 μm and $\phi$ = 27–52˚ with small fit errors of 1–2˚ (Fig 8A and 8C and S12 Fig in S1 File). The $\phi$-accuracy test (Eq 9) showed estimated $\phi$ < estimated $\phi_{cutoff}$ (~44–50˚) for all but two fits where estimated $\phi$ was ~3–4˚ above its estimated $\phi_{cutoff}$. For the two fits that failed the $\phi$-accuracy test, we recomputed the fits while fixing their $\phi$ to that estimated from the other fits of the same preparation (this made little difference in estimates of $\mu_D$ and $\sigma_D$). Averaging

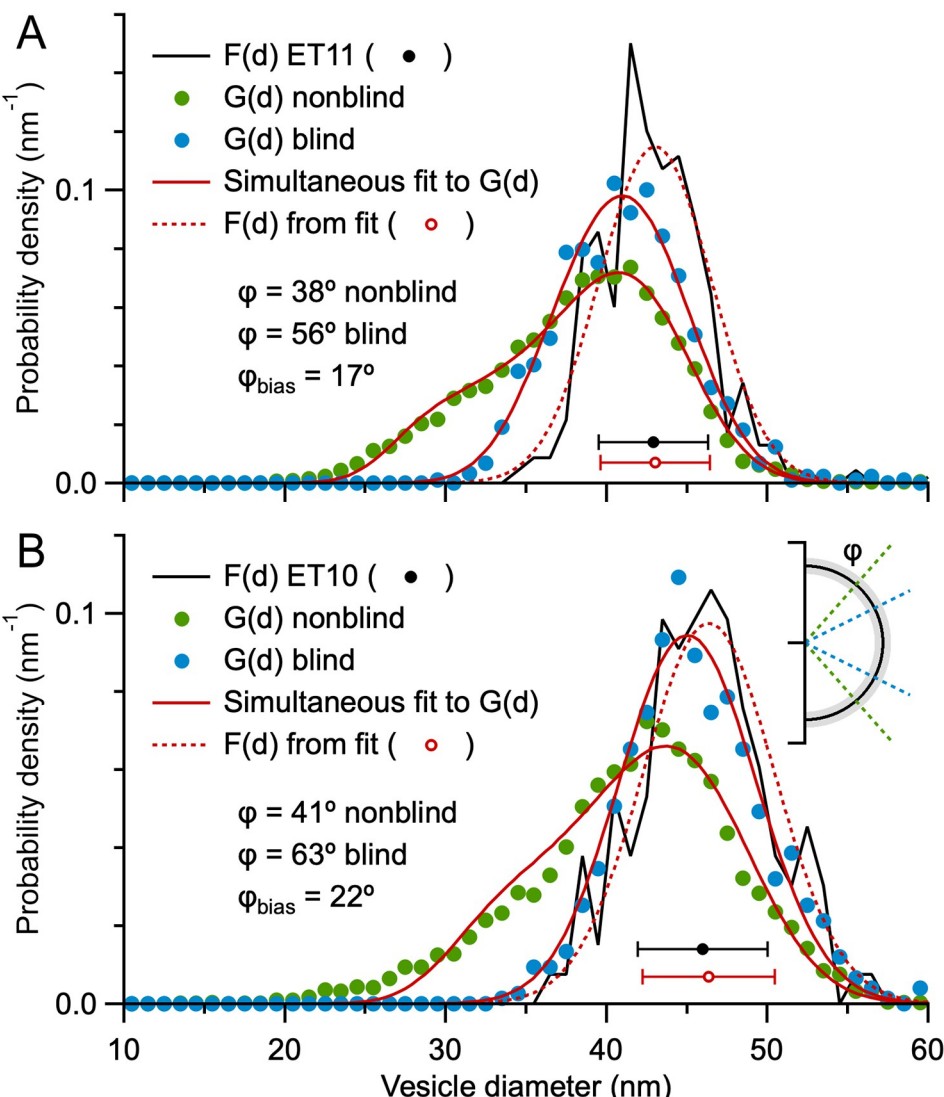

**Fig 7. Comparison of G(d) computed blind versus nonblind reveals a large bias in particle cap detection for MFT vesicles ($\phi_{bias} \approx 20°$). A.** Probability density of 2D diameters (G(d)) computed via a nonblind vesicle detection (green circles; ET11; Fig 6D) versus a blind vesicle detection (blue circles). Both G(d) were simultaneously curve fitted to Eq 1, where parameters $\mu_D$ and $\sigma_D$ were shared, revealing a 17° bias in $\phi$ (red solid lines; $\mu_D = 43.0 \pm 0.1$ nm, $\sigma_D = 3.5 \pm 0.1$ nm, nonblind $\phi = 38.3 \pm 0.5°$, blind $\phi = 55.5 \pm 0.7°$). As in Fig 6D, estimated F(d) (red dotted line and circle) is similar to measured F(d) (black solid line and circle). For the blind analysis, there were 889 diameters measured from 18 z-planes ($z_\# = 1–260$) spaced 5–15 nm apart, analysed in random order. **B.** Same as **A** for ET10 (S8D Fig in S1 File). The simultaneous curve fit revealed a 22° bias in $\phi$ (red solid lines; $\mu_D = 46.4 \pm 0.1$ nm, $\sigma_D = 4.1 \pm 0.1$ nm, nonblind $\phi = 40.9 \pm 0.7°$, blind $\phi = 63.3 \pm 1.3°$). For the blind analysis, there were 751 diameters measured from 30 z-planes ($z_\# = 51–236$) spaced 2–7 nm apart, analysed in random order. Inset cartoon depicts the bias in $\phi$ along the axial axis of a vesicle.

results across the 3 rats gave $\mu_D = 5.78 \pm 0.16$ μm, $\sigma_D = 0.41 \pm 0.03$ μm and $\phi = 37 \pm 2°$ (±SEM; Table 3). Simulations indicate these estimates have negligible biases and small confidence intervals ($\Delta\mu_D = 0.0 \pm 0.7\%$, $\Delta\sigma_D = +1.5 \pm 8.4\%$, $\Delta\phi = +0.1 \pm 1.3°$ for T = 0.3 u.d., true $\phi = 37°$, ~500 diameters).

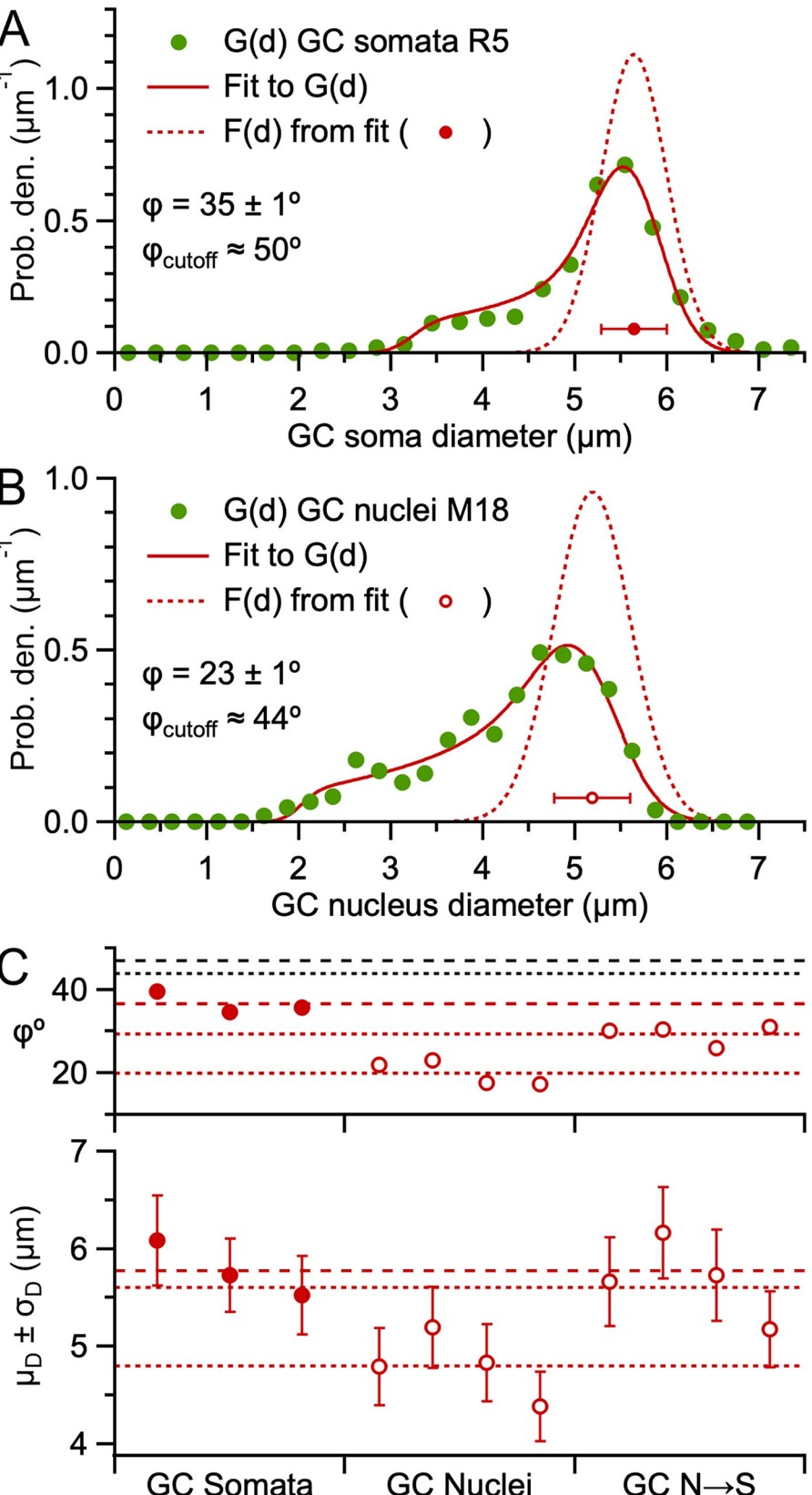

**Fig 8. Estimates of F(d) and ϕ from G(d) of cerebellar GC somata and nuclei. A.** Curve fit of Eq 1 (red solid line) to the G(d) of rat GC somata in Fig 2A2 (green circles) resulting in estimates for F(d) (red dotted line and circle), ϕ and ϕ_cutoff (Eq 9). **B.** Same as **A** for the G(d) of mouse GC nuclei in Fig 2B2. **C.** Parameters $\mu_D \pm \sigma_D$ (bottom) and ϕ (top) from Keiding-model fits to G(d) of GC somata for 3 rats (red closed circles; S12 Fig in S1 File; weighted averages with respect to number of tissue sections), G(d) of GC nuclei for 4 mice (red open circles; S13 Fig in S1 File; pooled diameters from 6–7 TEM images per mouse) and the same G(d) of GC nuclei scaled to somata dimensions (N→S). Red dashed and dotted lines denote averages across the 3 rats and 4 mice, respectively. Black dashed and dotted lines denote average estimated ϕ_cutoff for somata and nuclei (~47 and 44˚).

Next, we curve fitted the Keiding model to the 4 G(d) computed from GC nuclei of mice (Fig 2B2), resulting in $\mu_D$ = 4.39–5.19 μm, $\sigma_D$ = 0.36–0.42 μm and ϕ = 17–23˚ with small fit errors of 1–2˚ (Fig 8B and 8C and S13 Fig in S1 File) and all ϕ passing the ϕ-accuracy test (i.e. estimated ϕ < estimated ϕ_cutoff, where estimated ϕ_cutoff ≈ 43–44˚; Eq 9). The smaller less-variable ϕ for the GC nuclei compared to the GC somata reflects a more complete G(d) with fewer lost caps, achieved via the higher resolution and contrast of the TEM compared to confocal images. Moreover, since the nuclei were much larger than the section thickness, there were no overlaps of the nuclei in the TEM images and therefore less ambiguity of where the borders lay between nuclei. Across the 4 mice, $\mu_D$ = 4.80 ± 0.17 μm, $\sigma_D$ = 0.39 ± 0.01 μm and ϕ = 20 ± 1˚ (±SEM), which are similar to estimates computed from an analysis where all G(d) are aligned and pooled into a single G(d) (S14C Fig in S1 File). Simulations indicate these estimates have negligible biases and small confidence intervals (Fig 4C; $\Delta\mu_D$ = -0.1 ± 0.9%, $\Delta\sigma_D$ = +1.9 ± 11.9%, $\Delta\phi$ = 0.0 ± 1.6˚ for T = 0 u.d., true ϕ = 20˚, ~500 diameters).

To compare the above estimates for GC size between rats and mice, we scaled G(d) of the GC nuclei of mice via the $d_{soma}$-versus-$d_{nucleus}$ linear relation described above and curve fitted the Keiding model to the new G(d). Results gave $\mu_D$ = 5.60 ± 0.16 μm and $\sigma_D$ = 0.35 ± 0.01 μm (±SEM), which are similar to that estimated for rats (p = 0.49 and 0.07, respectively; *t*-test). These results are consistent with our simulations that showed overlapping projections of opaque particles (a likely scenario for the rat confocal dataset) create negligible bias in estimates of $\mu_D$ (Fig 5D). Hence, these results are consistent with F(d) being the same for GCs in rats and mice. The results also indicate the two different tissue preparations of the rat and mouse datasets (chemical versus cryo fixation) had little effect on the size of GCs.

## Estimation of the 3D size of vesicles in mossy fiber terminals

To estimate the 3D diameter of MFT vesicles, we curve fitted Eq 1 to the 8 G(d) computed from MFT vesicles of mice (Fig 2C2), resulting in estimated $\mu_D$ = 42.4–47.2 nm, $\sigma_D$ = 3.5–5.0 nm and ϕ = 45–70˚ with large fit errors most of which exceeded 80˚ (Fig 9 and S15 Fig in S1 File). Here, the large estimated ϕ with large fit errors indicate ϕ was indeterminable (Fig 4C). This conclusion was further supported by the finding that all estimated ϕ failed the ϕ-accuracy test (i.e. estimated ϕ > estimated ϕ_cutoff, where ϕ_cutoff ≈ 34–42˚). While these estimates of ϕ are larger than that computed from our ET z-stack analysis (Fig 6D and S8D Fig in S1 File; ϕ ≈ 40˚), this is expected for the G(d) analysed here given the greater difficulty in identifying vesicle caps using a blind analysis (Fig 7) and thick sections (T ≈ $\mu_D$). While the large estimates of ϕ with large errors means there may be small biases in these final estimates of $\mu_D$ and $\sigma_D$ (absolute values < 1 and 5%, respectively; Fig 4C), the estimates are not significantly different to the 3D measures computed from our two ET z-stacks (p = 0.3 and 0.6, respectively; Student's *t*-test). Again, the similarity in estimates of F(d) between our datasets indicates differences in tissue preparation (cryo versus chemical fixation) had little effect on the size of synaptic vesicles, which is consistent with other studies [31, 61].

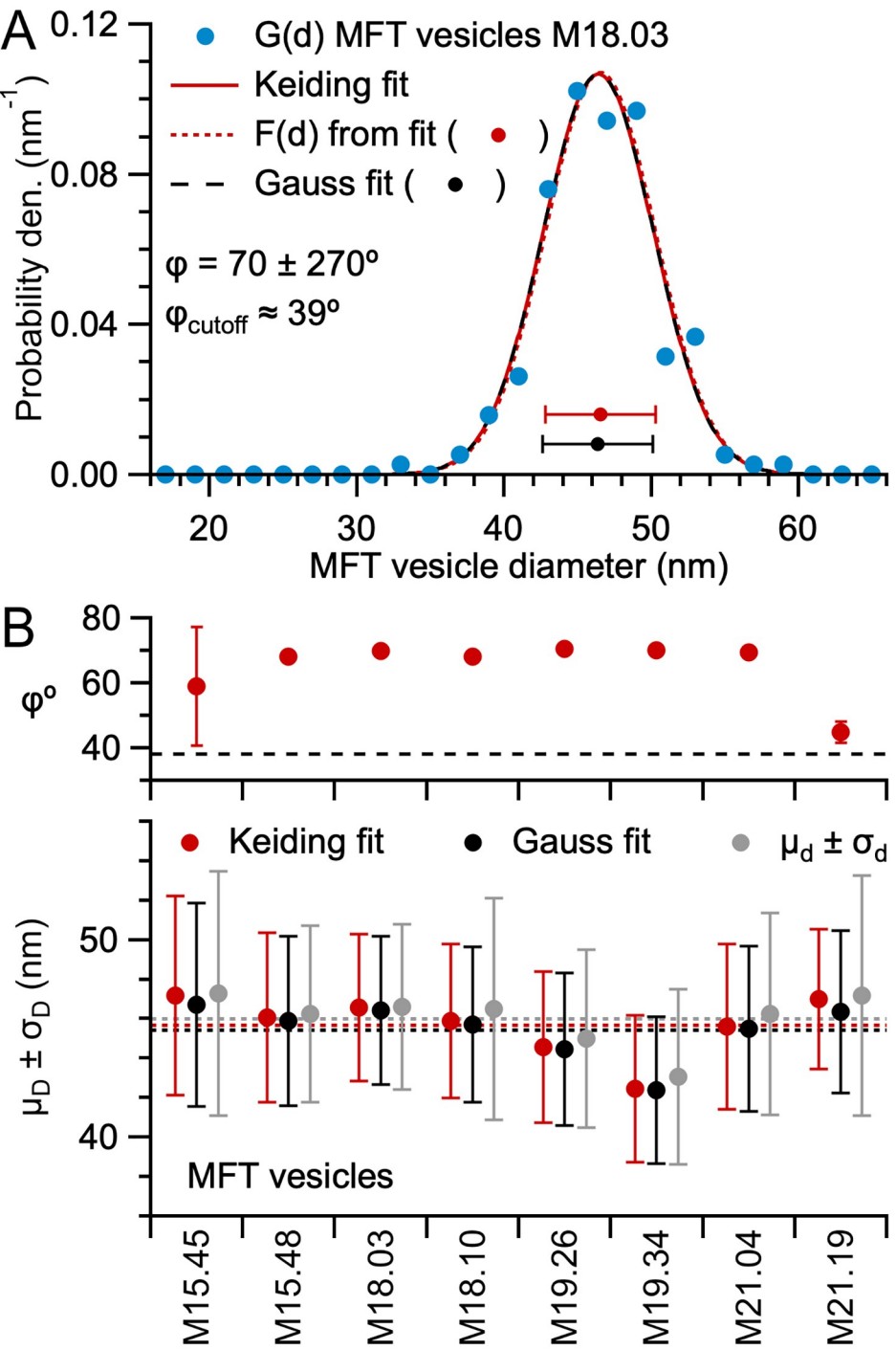

**Fig 9. Estimates of F(d) and φ from G(d) of MFT vesicles. A.** Curve fit of Eq 1 (red solid line) to the G(d) of mouse MFT vesicles in Fig 2C2 (blue circles) resulting in estimates for F(d) (red dotted line and circle), φ and $φ_{cutoff}$ (Eq 9). A Gaussian fit to the same G(d) (black dashed line) overlaps the Keiding-model fit and $μ_D ± σ_D$ of both fits overlap (red and black circles) indicating G(d) ≈ F(d). **B.** Parameters $μ_D ± σ_D$ (bottom) and φ (top) from Keiding-model curve fits to G(d) of MFT vesicles for 4 mice, 2 MFTs per mouse (red circles; S15 Fig in S1 File), compared to $μ_D ± σ_D$ from Gaussian fits (black circles) and $μ_d ± σ_d$ computed from 2D diameters. Overlapping distributions again indicate G(d) ≈ F(d). Dotted lines (bottom) denote averages. Black dashed line (top) denotes average estimated $φ_{cutoff}$ (~38˚); all estimated φ are considered inaccurate. Error bars of φ denote fit errors, 6 of which are not shown since they are off scale. See S16 Fig in S1 File.

## Estimating the 3D density of particles from their 2D projection

Having an accurate estimate of the number of particles per unit volume is essential to many fields of science. Historically, estimates of 3D particle density ($\lambda_{3D}$) were obtained by computing the number of particles per unit area ($\lambda_{2D}$; S17 Fig in S1 File) observed in a projection divided by the section thickness (T). However, particle caps on the top and bottom of the section inflate the particle count with respect to T, i.e. they create an overprojection [17, 40–43]. To correct for overprojection, one can use $\phi$ to estimate $\lambda_{3D}$ via Eq 3 (S1 Appendix in S1 File).

As a first test of the validity of Eq 3, we used the Monte Carlo simulations in Fig 4C to investigate the relation between the measured $\lambda_{2D}$ and true $\lambda_{3D}$ by computing $\zeta = \lambda_{2D}/\lambda_{3D}$. Comparison of this measured 'true' $\zeta$ to the expected $\zeta$ computed via Eq 2, using true T, $\mu_D$ and $\phi$, showed a close agreement (Fig 10A). Hence, our simulations of the Keiding model were well described by Eq 3. Next, we tested the ability of the Keiding model to accurately estimate $\lambda_{3D}$ via Eq 3 by computing $\Delta\lambda_{3D}$ for the same simulations using parameters $\mu_D$ and $\phi$ estimated via Keiding-model curve fits to G(d). Results showed accurate estimates of $\lambda_{3D}$, except when true $\phi > 45°$, in which case the estimation errors of $\mu_D$ and $\phi$ translated into estimation errors of $\lambda_{3D}$ (Fig 10B). Here, the finding that $\phi_{cutoff}$ for estimated $\lambda_{3D}$ is less than that for estimated $\mu_D$, $\sigma_D$ and $\phi$ (~55°) is consistent with an increase in variability from using estimated $\mu_D$ and $\phi$ to compute $\zeta$, and therefore consistent with our $\phi$-accuracy test, where estimated $\phi_{cutoff} \approx 43°$ (Eq 9). Hence, these results demonstrate the ability of Eq 3 to accurately estimate $\lambda_{3D}$ using Keiding-model estimates of $\mu_D$ and $\phi$ as long as estimated $\phi <$ estimated $\phi_{cutoff}$. Comparing results for planar and thick sections showed qualitatively similar results, except errors were smaller for thick sections due to smaller errors in $\mu_D$ (but see next paragraph for caveats of using thick sections). Repeating the error analysis for G(d) computed from ~2000 diameters gave qualitatively similar results compared to those for ~500 diameters, but $\zeta$ and $\lambda_{3D}$ had smaller biases and confidence intervals, where the confidence intervals followed a $1/\sqrt{n}$ relation for $\phi < 45°$ (S18 Fig in S1 File) similar to that of estimated $\mu_D$ and $\phi$ (S4 Fig in S1 File).

For thick sections, a major caveat of estimating $\lambda_{3D}$ from $\lambda_{2D}$ via Eq 3 is that the calculation assumes one has a good estimate of $\zeta$. As shown in Fig 5B, however, overlapping projections of semi-transparent and opaque particles may preclude counting particles at the bottom of the section, thereby reducing $\zeta$. To demonstrate this, we computed $\Delta\zeta$ for simulations similar to those in Fig 5 and found large positive biases for both semi-transparent and opaque particles, i.e. estimated $\zeta$ was larger than true $\zeta$ (Fig 10A). As expected, the overestimation of $\zeta$ translated into large negative biases in estimates of $\lambda_{3D}$ (Fig 10B). Given this caveat, therefore, the best approach to estimate $\lambda_{3D}$ from $\lambda_{2D}$ for particles with a high density is to use planar sections, in which case there will be little to no interference in counting particles from overlapping particle projections.

As a second independent method for computing the 3D density, we used the particle area fraction (AF) to VF relation of Weibel and Paumgartner [62] (VF = $K_v$·AF) to compute the VF of our simulations, where AF is the sum of all projection areas divided by the total projection area (Area$_{xy}$) and $K_v$ is a proportional scale factor that is a function of T and $\mu_D$, but modified to be a function of $\phi$ rather than $h_{min}$ (S2 Appendix in S1 File). Results showed excellent agreement between the estimated and true VF of our simulations (Fig 10C). However, for the simulations of overlapping projections of semi-transparent and opaque particles in thick sections, there was a significant underestimation of the VF, as expected.

## Validation of the Keiding model for estimating particle density using 3D reconstructions

To test the accuracy of applying Eq 3 to real data, we used our ET z-stacks of MFT vesicles to measure $\lambda_{2D}$ and $\lambda_{3D}$ within a subregion of the z-stacks where the vesicles were clustered

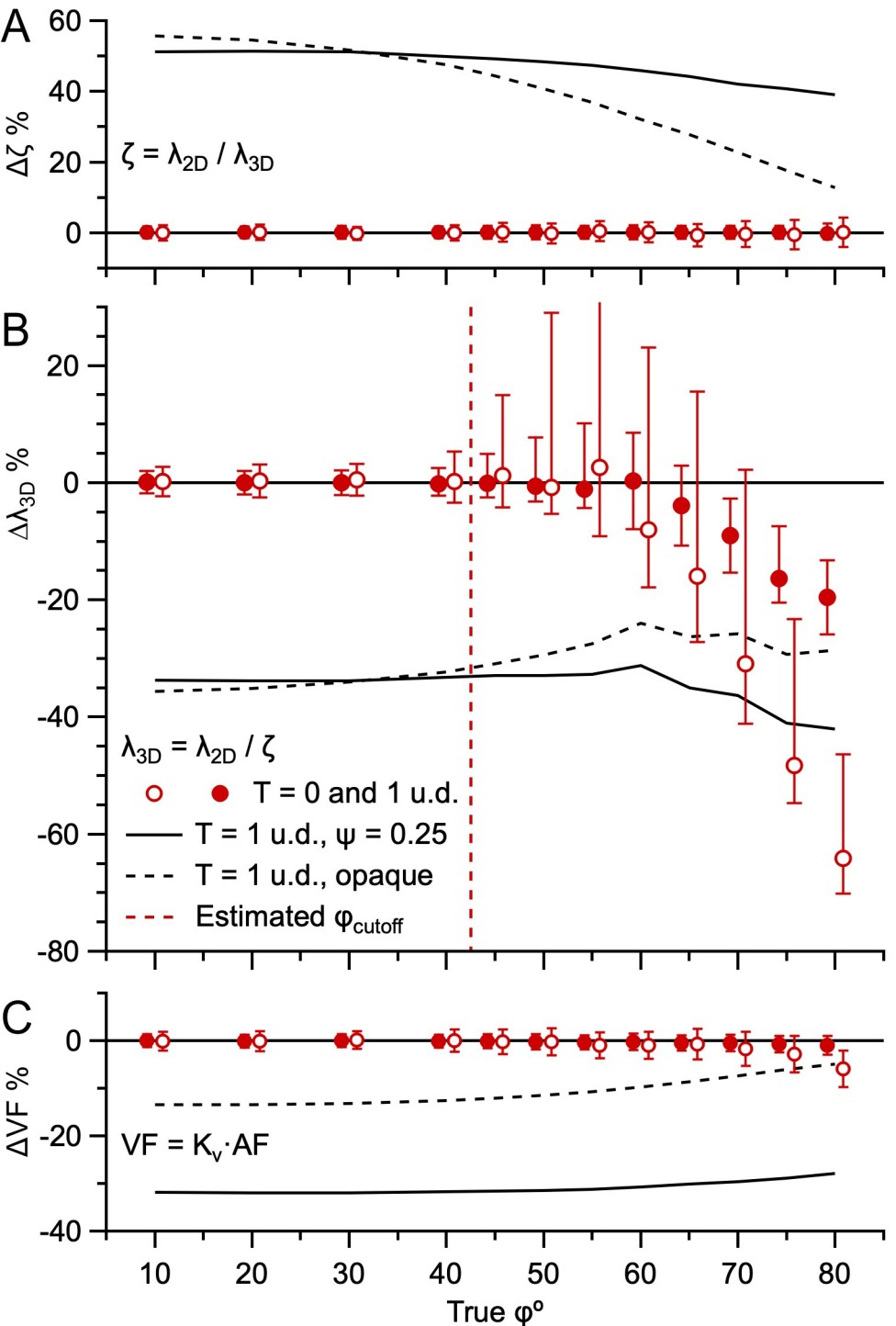

**Fig 10. The Keiding model accurately estimates $\lambda_{3D}$ from $\lambda_{2D}$ for true $\phi$ < estimated $\phi_{cutoff}$ (simulations). A.** Average error in section z-depth over which particle center points are sampled ($\Delta\zeta$) vs. true $\phi$ (top) for simulations in Fig 4C for T = 0 and 1 u.d. (red open and closed circles) where estimated $\zeta$ was computed via Eq 2 using true $\mu_D$, $\phi$ and T and 'true' $\zeta = \lambda_{2D}/\lambda_{3D}$ using measured $\lambda_{2D}$ and true $\lambda_{3D}$. **B.** Average error in 3D particle density ($\Delta\lambda_{3D}$) vs. true $\phi$ (middle) for the same simulations, computed via estimated and true $\lambda_{3D}$, where estimated $\lambda_{3D}$ was computed via Eq 3 ($\lambda_{3D} = \lambda_{2D}/\zeta$) using measured $\lambda_{2D}$ and estimated $\mu_D$ and $\phi$ from Keiding-model fits to the simulated G(d). Red dashed line denotes estimated $\phi_{cutoff}$ (~43˚; Eq 9). One error bar at true $\phi$ = 55˚ is off scale (+56%). **C.** Average error in volume fraction ($\Delta$VF) vs. true $\phi$ (bottom) for the same simulations, computed via estimated and true VF, where estimated

VF = $K_v \cdot$AF (Eq 14). For **A**, **B** and **C**, results are also shown for transparent particles (black solid line; $\psi$ = 0.25) and opaque particles (black dashed line) in thick sections (T = 1 u.d.; Fig 5), showing estimated $\zeta$ was larger than true $\zeta$ when projection overlaps hindered particle counting, creating a large underestimation of $\lambda_{3D}$ and VF. However, for opaque particles, the bias was larger for $\lambda_{3D}$ than VF since an overlap in two projections reduced the particle count by one, but only partially reduced the AF (overlapping projections coalesce into one). Data x-scales were shifted ±0.8° to avoid overlap. See S18 Fig in S1 File.

(Fig 11). For ET11, this analysis gave $\lambda_{2D}$ = 369 $\mu m^{-2}$ and $\lambda_{3D}$ = 11,022 $\mu m^{-3}$ (VF = 0.47; Eq 4) where $\lambda_{3D}$ was computed as a 3D measurement, i.e. count per volume. Using Eq 3, we then computed $\lambda_{3D} = \lambda_{2D}/\zeta$ = 11,482 $\mu m^{-3}$ (VF = 0.48; Eq 4) where $\zeta$ = 32 nm, computed via Eq 2 using the measured $\mu_D$ and $\mu_\phi$ (Table 2; 3D-NB). Hence, $\lambda_{3D}$ computed via Eq 3 was nearly the same as that computed via a direct 3D measurement ($\Delta\lambda_{3D}$ = +4%). Repeating the same computation using $\zeta$ = 34 nm computed via Keiding-model curve-fit parameters $\mu_D$ and $\phi$ (Fig 6D), where estimated $\phi$ < estimated $\phi_{cutoff}$ (~51°), gave $\lambda_{3D}$ = 10,991 $\mu m^{-3}$ (VF = 0.46; Eq 4), which was also nearly the same to $\lambda_{3D}$ computed via the 3D measurement ($\Delta\lambda_{3D}$ = -0.3%). In contrast, when we used $d_{min}$ of the z-stack analysis (19 nm) to estimate $\zeta$ (using measured $\mu_D$) rather than $\phi$ (where $d_\phi$ = 28 nm), we computed $\lambda_{3D}$ = 9560 $\mu m^{-3}$. Hence, in this example, using $d_{min}$ to estimate $\lambda_{3D}$ caused significant error ($\Delta\lambda_{3D}$ = -13%). Next, using the VF = $K_v \cdot$AF relation (Eq 14) we computed VF = 0.46 that matched that computed via the 3D analysis (0.47). Repeating the same 2D versus 3D density analysis for our other ET z-stack (ET10) produced similar results ($\Delta\lambda_{3D}$ = +2%; Fig 11B; Table 2); however, unlike ET11, the density analysis of ET10 had to be confined within a subregion of the z-stack due to a nonhomogeneous distribution of vesicles in the axial axis.

To examine whether these results hold for larger particles, we repeated a similar 2D versus 3D density analysis for a recently published TEM z-stack of GC nuclei [60]. Results showed $\lambda_{3D}$ estimated from 2D images via Eq 3 closely matched that computed from a 3D analysis using images from the same z-stack ($\Delta\lambda_{3D}$ = -4%; S9E Fig in S1 File; Table 3) and a full 3D reconstruction [60]. Hence, our 2D versus 3D analyses show that Eq 3 accurately estimates $\lambda_{3D}$ from $\lambda_{2D}$ with only small error when true $\phi$ < $\phi_{cutoff}$, confirming our previous results from simulations (Fig 10).

Finally, we investigated the range of expected error for the ET z-stack density analysis, as well as that from assuming a fixed $\phi$ (S10 Fig in S1 File), by computing $\lambda_{2D}$ and $\lambda_{3D}$ for Monte Carlo simulations that mimicked the 3D analysis. Results gave ranges of $\Delta\lambda_{3D}$ that were consistent with the above measured $\Delta\lambda_{3D}$ (average -2 ± 3% for ET10 and ET11 simulations; S19A Fig in S1 File) and showed the assumption of a fixed $\phi$ introduces only small errors for estimating $\lambda_{3D}$ from $\lambda_{2D}$ via Eq 3 for $\phi$ < 50° (S19B Fig in S1 File). Again, using $d_{min}$ to estimate $\lambda_{3D}$ caused significant error ($\Delta\lambda_{3D}$ = -19 ± 2%; n = 100 simulation repetitions).

## Estimating the 3D particle density via the disector method

The disector method [18, 52] is a popular method for estimating $\lambda_{3D}$ of particles with arbitrary geometry using two adjacent sections from a z-stack, referred to as the 'reference' and 'lookup' section. This method counts the number of particles that appear within the reference section but do not appear in the lookup section, i.e. the method counts a particle only if its leading edge appears within the reference section. To compute $\lambda_{3D}$, one determines $\lambda_{2D}$ from the leading-edge count and divides $\lambda_{2D}$ by the distance between the reference and lookup sections (i.e. the section thickness for adjacent pairs). Although theoretically, lost caps should not introduce error into this counting method [52], underestimation errors due to lost caps have been reported [63]. Given the popularity of the disector method, as well as the potential source of error due to lost caps, we decided to use the disector method to estimate $\lambda_{3D}$ of our simulated projections where $\phi$ = 10–80° and compare these results to our analysis in Fig 10B, where $\lambda_{3D}$

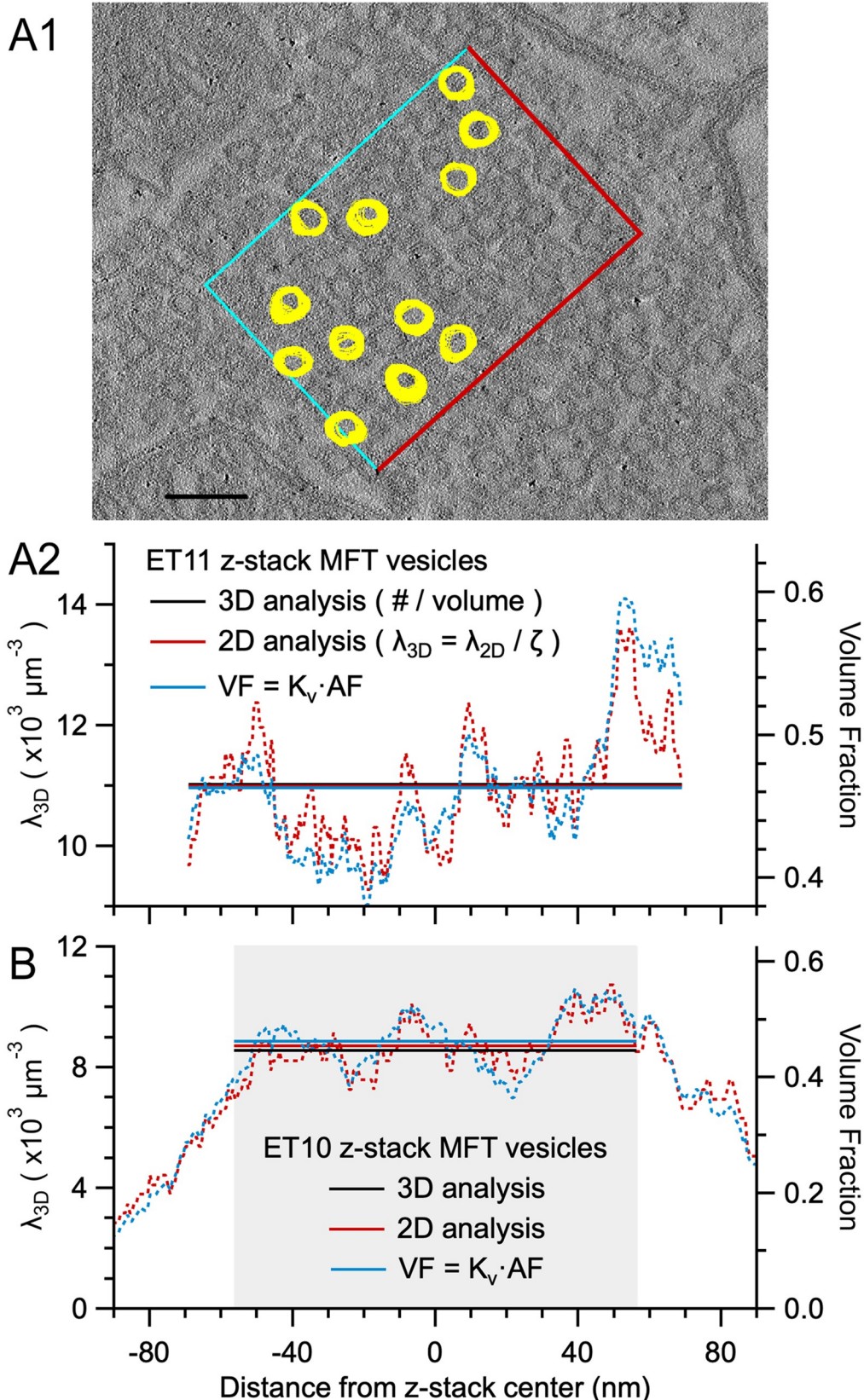

**Fig 11. The Keiding model accurately estimates $\lambda_{3D}$ from $\lambda_{2D}$ for true $\phi <$ estimated $\phi_{cutoff}$ (vesicles). A1.** One of 261 serial ET images (ET11) of a cerebellar tissue section. A ROI ($0.45 \times 0.32$ μm; Area$_{xy}$ = 0.144 μm$^2$) was placed within a large cluster of MFT vesicles and those vesicles that obeyed the inclusive/exclusive borders (blue/red; n = 271) were tracked and outlined through multiple z-planes. Because the analysis includes vesicle caps on the top/bottom of the reconstruction, the vesicle sampling space of the volume of interest (VOI) extends above/below the section such that $\zeta$ = 170 nm (Fig 1B; Eq 2; 3D measures: T = 138 nm, $\mu_D$ = 42.9 nm, $\mu_\phi$ = 42°) giving VOI = 0.025 μm$^3$. Here, outlines for 12 representative vesicles are overlaid with outlines from images above and below. Scale bar 100 nm. **A2.** Vesicle $\lambda_{3D}$ (left axis) and VF (right axis) computed within the ROI in **A1**. For the 3D analysis, $\lambda_{3D}$ = N$_{3D}$/VOI = 11,022 μm$^{-3}$ (black solid line). For the first 2D analysis, $\lambda_{3D}$ = $\lambda_{2D}/\zeta$ (Eq 3) computed for each z-plane (red dotted line) and the sum of all z-planes (red solid line), where $\lambda_{2D}$ is the number of outlines per ROI area and $\zeta$ = 34 nm computed via Keiding-model estimates $\mu_D$ and $\phi$ (Fig 6D). For the second 2D analysis, VF = K$_v$·AF (Eq 14) computed for each z-plane (blue dotted line) and the sum of all z-planes (blue solid line; 0.46), where K$_v$ = 1.07 and AF is the sum of all vesicle outline areas per ROI area. Both the measured $\phi$ (42°) and estimated $\phi$ (38°) are less than estimated $\phi_{cutoff}$ (51°; Eq 9). Left and right axes are equivalent scales for $\mu_D \pm \sigma_D$ = 42.9 ± 3.4 nm, the measured F(d). **B.** Same as **A2** for ET10. Gray shading denotes the axial subregion where the 3D analysis and averages were computed (Materials and Methods). Left and right axes are equivalent scales for $\mu_D \pm \sigma_D$ = 46.0 ± 4.0 nm. See S19 Fig in S1 File.

was estimated via Eq 3, i.e. the $\phi$-correction method. Using T = 0.3 u.d. as recommended [18], we found no bias due to lost caps for all $\phi$ tested (Fig 12, black squares; ~500 particles per projection). This result is consistent with historical thinking that lost caps do not create a bias in counting: while the effect of lost caps on any given reference section is to reduce the particle count, the effect of lost caps on the lookup section is to increase the particle count for that reference section, the two effects thereby cancelling [52]. In essence, lost caps simply shift the reference section to which particles are counted and therefore have no effect on the final particle count. However, these simulations did not take into account a potentially significant bias pointed out by Hedreen [27]: fewer caps are likely to be identified in the reference section compared to the lookup section since the former is done blind and the latter is not (the researcher searches for caps in the lookup section only after identifying a particle in the reference section). In fact, our analysis of ET z-stacks of MFT vesicles revealed a ~20° bias in $\phi$ between a blind and nonblind vesicle detection (Fig 7). To investigate the effects of such a bias, we added a bias to our disector simulations by increasing $\phi$ of the reference section with respect to the lookup section ($\phi_{bias}$ = 5–20°) thereby reducing cap identification in the reference section (i.e. increasing the number of lost caps) with respect to the lookup section. Interestingly, results of these simulations showed large underestimation errors even for small $\phi_{bias}$ (Fig 12; -5% < $\Delta\lambda_{3D}$ < -20% for $\phi_{bias}$ = 10°). Hence, the blind-versus-nonblind bias in particle detection has the potential to create a significant underestimation of $\lambda_{3D}$. Repeating the disector bias analysis using our experimental ET z-stack data gave similar results, showing a large error ($\Delta\lambda_{3D}$ = -41 ± 5%) for $\phi_{bias}$ = 20°.

Besides the potential error due to a blind-versus-nonblind bias in particle detection, our simulations show that the disector method is ~2 to 3-fold less accurate at estimating $\lambda_{3D}$ (i.e. $\Delta\lambda_{3D}$ has a larger ±$\sigma$) compared to the $\phi$-correction method (S18 Fig in S1 File; ±$\sigma$ panel) due to a lower particle count per section. Similar results were found using the disector method to compute $\lambda_{3D}$ of our ET z-stack of MFT vesicles, where $\Delta\lambda_{3D}$ = ±6% for the disector method (±$\sigma$; $\phi_{bias}$ = 0°) compared to $\Delta\lambda_{3D}$ = ±2% for $\lambda_{3D}$ computed via Eq 3 (S19A Fig in S1 File). Hence, the inherent smaller count per region of interest (ROI) of the disector method leads to a larger uncertainty in estimates of $\lambda_{3D}$.

## Estimation of the 3D density of granule cell somata and nuclei (confocal vs. TEM)

Having verified Eq 3 using simulations and experimental data, we applied the equation to the estimation of $\lambda_{3D}$ from $\lambda_{2D}$ for our sample of GC somata. Using the same confocal images used to compute the 9 G(d) of the GC somata of 3 rats, we computed $\lambda_{2D}$ = 18–23 × 10$^3$ mm$^{-2}$ (S17A

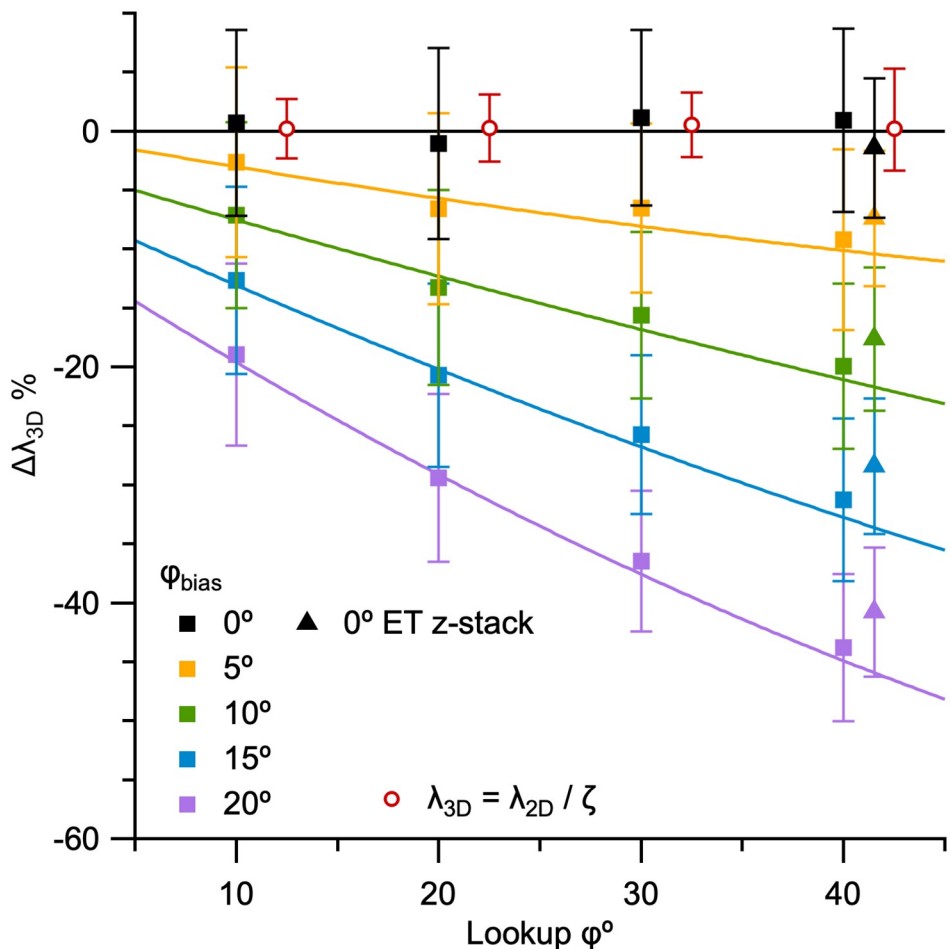

**Fig 12. Estimation errors of the 3D density computed via the disector method for different degrees of bias due to blind-versus-nonblind cap detection.** Average $\Delta\lambda_{3D}$ computed from Monte Carlo simulations of the disector method, where a particle-detection bias was simulated by increasing $\phi$ of the reference section (a blind cap detection) with respect to the lookup section (a nonblind cap detection) for $\phi_{bias} = 0–20°$ (colored squares; $\phi_{ref} = \phi_{lookup} + \phi_{bias}$; ~500 particles per reference section). Thickness of the reference and lookup sections was 0.3 u.d. An equivalent disector analysis using the ET11 z-stack data gave similar results (triangles; lookup $\phi = 42°$). Average $\Delta\lambda_{3D}$ for the simulations in Fig 10B are shown for comparison (red open circles; T = 0 u.d.; $\lambda_{3D} = \lambda_{2D}/\zeta$; x-scale is true $\phi$ and is shifted 2.5° to avoid overlap) highlighting the higher levels of accuracy compared to the disector method, i.e. smaller confidence intervals ($\sigma_A$).

Fig in S1 File; 8792–29,845 μm$^2$ ROI area per image, 196–528 somata per ROI, 3 images per rat). Using estimated $\mu_D$ and $\phi$ for each G(d), we then computed $\zeta = 5.5–7.6$ μm via Eq 2 (estimated T ≈ 1.4–2.7 μm, scaled for z-shrinkage; Table 1). Finally, we computed $\lambda_{3D} = \lambda_{2D}/\zeta = 2.7–4.1 \times 10^6$ mm$^{-3}$ with mean $\lambda_{3D} = 3.2 \pm 0.2 \times 10^6$ mm$^{-3}$ and VF = $0.32 \pm 0.01$ (±SEM; Table 3; Eq 4). To verify these results using the relation VF = $K_v \cdot$AF, we computed $K_v = 0.58–0.77$, AF = 0.39–0.50 and VF = $K_v \cdot$AF = 0.27–0.37, with mean VF = $0.32 \pm 0.02$ and $\lambda_{3D} = 3.1 \pm 0.1 \times 10^6$ mm$^{-3}$ (±SEM) which is not significantly different to $\lambda_{3D}$ computed via Eq 3 (p = 0.4, paired *t*-test).

Next, we estimated $\lambda_{2D}$ of GC nuclei from the TEM images used to compute G(d) ($\lambda_{2D} = 17–34 \times 10^3$ mm$^{-2}$; S17B Fig in S1 File; 2046–5747 μm$^2$ ROI area per image, 35–158 nuclei per ROI, 6–7 images per mouse). Using estimated $\mu_D$ and $\phi$ for each G(d), we then computed $\zeta = 4.2–4.8$ μm via Eq 2 (T = 0.06 μm) and $\lambda_{3D} = \lambda_{2D}/\zeta = 3.5–8.1 \times 10^6$ mm$^{-3}$, with mean $\lambda_{3D} = 5.9 \pm 0.4 \times 10^6$ mm$^{-3}$ and VF = $0.34 \pm 0.01$ (±SEM; Table 3; Eq 4). Simulations indicate $\lambda_{3D}$ for

this dataset should have negligible bias with a small confidence interval (Fig 10B; $\Delta\lambda_{3D}$ = +0.3 ± 2.8% for T = 0 u.d. and true $\phi$ = 20˚). To verify these results using the relation VF = $K_v \cdot$ AF, we computed $K_v$ = 0.98–0.99, AF = 0.27–0.39 and VF = $K_v \cdot$ AF = 0.27–0.39, with mean VF = 0.33 ± 0.01 and $\lambda_{3D}$ = 5.7 ± 0.5 × $10^6$ mm$^{-3}$ (±SEM; n = 4) which is not significantly different to $\lambda_{3D}$ computed via Eq 3 (p = 0.1, paired $t$-test). Finally, to scale the VF estimates from nuclei to somata, we estimated the VF for GC somata of mice via Eq 4 using estimated F (d) for GC somata of mice (Fig 8C; N→S) and assuming $\lambda_{3D}$ was the same for the nuclei and somata. Results gave VF = 0.51–0.57 with mean 0.54 ± 0.01 (±SEM), suggesting GC somata occupied half of the GC layers we analysed.

Unlike our estimates for F(d) of GCs in rats and mice, which show no difference, comparison of the above estimates for $\lambda_{3D}$ in rats and mice show a 2-fold difference: estimated $\lambda_{3D}$ of the rat dataset (3.2 × $10^6$ mm$^{-3}$) is significantly smaller than estimated $\lambda_{3D}$ of the mouse dataset (5.9 × $10^6$ mm$^{-3}$; p = 0.004). While this difference could reflect a true difference between species, a previous comparative study of the cerebellum, which used the same methodology across species, reported similar densities within the GC layer of rats and mice [64]. Hence, it is more likely that the 2-fold difference in estimated $\lambda_{3D}$ is due to one or more differences in methodology. Although there are perhaps too many differences in methodology to make a decisive conclusion, notwithstanding the confounding problem of the 'reference trap' [65], it is still instructive to consider them here. The first difference in methodology is section thickness: thick versus planar. Because the rat dataset was derived from thick sections, overlapping opaque projections could have created an underestimate of $\lambda_{3D}$ compared to the mouse dataset (Fig 10B). The second difference in methodology is imaging technology: confocal versus TEM. Because of the lower contrast and resolution of the confocal images, it was considerably harder to identify and delineate GC somata profiles compared to the GC nuclei profiles. The poor delineation of GCs would create an undercount. The third difference in methodology is tissue preparation: chemical versus cryo fixation. A change in the extracellular volume due to chemical fixation of the rat tissue preparation could have created a lower $\lambda_{3D}$. However, this explanation is unlikely since chemical fixation tends to reduce the extracellular volume [61] which would result in a higher rather than lower $\lambda_{3D}$. The final difference in methodology is the sampling space: ROIs were larger in the rat dataset compared to the mouse dataset. Because GCs are not distributed uniformly, but rather form high-density clumps interspersed by MFTs and blood vessels, there could be a bias towards a larger $\lambda_{3D}$ in the mouse dataset due to smaller ROIs. However, our analysis of the two datasets using different sized ROIs (not shown) indicates the difference in ROI size cannot account for the 2-fold difference in estimated $\lambda_{3D}$. Hence, we suspect our estimate of $\lambda_{3D}$ in the rat dataset is underestimated, and this is most likely due to overlapping projections in thick sections and poor delineation of GC profiles in the confocal images. Because of the tight packing of GC somata, estimates of $\lambda_{3D}$ via Eq 3 are best achieved by using planar sections, a higher contrast preparation and superior microscope resolution. We therefore believe our most accurate estimate of the GC $\lambda_{3D}$ is that of our mouse dataset. Although our estimated $\lambda_{3D}$ of GC somata in rats is likely to be underestimated, it is still 1.7-fold larger than our previous estimate from the same confocal images [6]. Because the latter estimate of $\lambda_{3D}$ was computed via the disector method, we suspect it is underestimated due to the blind-versus-nonblind bias in particle detection discussed in the previous section.

### Estimation of the 3D density of clustered vesicles in mossy fiber terminals (TEM vs. ET)

Although $\phi$ was indeterminable for our analysis of MFT vesicles in TEM images, it was still possible to estimate $\lambda_{3D}$ over a range of $\phi$, i.e. $\phi > \phi_{cutoff}$ or $\phi = \phi_{cutoff}$–90˚. First, we computed

$\lambda_{2D}$ of MFT vesicles (S17C Fig in S1 File) using the same TEM images used to compute their G (d) (S15 Fig in S1 File; n = 8), giving $\lambda_{2D}$ = 305–429 $\mu m^{-2}$. Next, we computed $\zeta$ = 97–60 nm via Eq 2 (T = 60 nm) for $\phi$ = 38–90˚. Finally, we computed $\lambda_{3D}$ = 3200–7200 $\mu m^{-3}$ via Eq 3. Not surprisingly, this range of $\lambda_{3D}$ is considerably smaller than that for our 3D ET analysis of MFT vesicles ($\lambda_{3D}$ = 9,000–11,000 $\mu m^{-3}$). Again, similar to our analysis of GCs in confocal images, we suspect $\lambda_{3D}$ is underestimated due to overlapping projections in thick sections (Fig 10B). To estimate the effective section depth of those vesicles counted, we assumed $\lambda_{3D}$ of the TEM dataset was the same as that of our ET dataset and computed $\zeta = \lambda_{2D}/\lambda_{3D}$ = 31–44 nm. This range of $\zeta$ is ~2-fold smaller than that estimated via Eq 2, indicating vesicles at the bottom of the tissue sections were not counted (Fig 5B), creating an underestimation of $\lambda_{3D}$. Hence, this analysis lends support to our conclusion that the optimal method for estimating $\lambda_{3D}$ from $\lambda_{2D}$ for particles with a high density is to use planar sections, in which case there will be no interference of counting from overlapping projections. Since the thinnest possible tissue section created via an ultramicrotome is currently on the order of a vesicle, the best option for computing vesicle density is via ET reconstructions (Fig 11).

## Discussion

Stereological methods for estimating the 3D size distribution (F(d)) and density ($\lambda_{3D}$) of a collection of particles from their 2D projection are essential tools in many fields of science. These methods, however, inevitably contain sources of error, one being the unresolved or nonexistent profiles known as lost caps. Surprisingly, the simple solution for lost caps developed by Keiding et al. [49], which defines lost caps of spherical particles with respect to a single (i.e. fixed) cap-angle limit ($\phi$), has not been widely adopted and has never been validated. Here, we provide the first experimental validation of the Keiding model by quantifying $\phi$ of unresolved vesicle caps within 3D reconstructions. While this analysis reveals a Gaussian distribution for $\phi$ rather than a single value, curve fits of the Keiding model to the 2D diameter distribution (G (d)) nevertheless accurately estimate the mean $\phi$, as well as F(d). Parameter space evaluation with Monte Carlo simulations revealed that the estimates are most accurate when $\phi$ falls below a specific value ($\phi_{cutoff}$). Hence, our experimental and theoretical analyses reveal that, if one only wishes to estimate F(d) from a 2D projection, then the Keiding model can be called to task, whether one is using planar or thick sections. On the other hand, if one wishes to estimate both F(d) and $\lambda_{3D}$ from a 2D projection, then one will need an accurate estimate of $\phi$ (Fig 13). As we discuss below, obtaining an accurate estimate of $\phi$ for some preparations may require optimising experimental conditions.

### Basic assumptions of the Keiding model

There are five basic assumptions of the Keiding model [49] that one must keep in mind when applying it to the estimation of particle size and density via Eqs 1–3.

The first assumption is that the particles of interest are approximately spherical, i.e. they are convex with the average shape of a sphere in rotation, which includes elliptical [16]. The assumption of a spherical shape is usually valid for vesicles, vacuoles, nuclei and cell bodies [1, 17, 44, 49, 66], but also for large structures such as follicles [16] and glomeruli [67].

The second assumption is that F(d) is well described by a probability density function (PDF), e.g. a Gaussian, chi or gamma distribution. The assumption that F(d) was a simple Gaussian distribution worked well for our analysis of MFT vesicles and GC nuclei (S7 Fig in S1 File) and for the liver cell nuclei of Keiding et al. [49] (S2 Fig in S1 File). Similarly, the assumption that F(d) was a chi distribution worked well for Wicksell's corpuscles (S3 Fig in

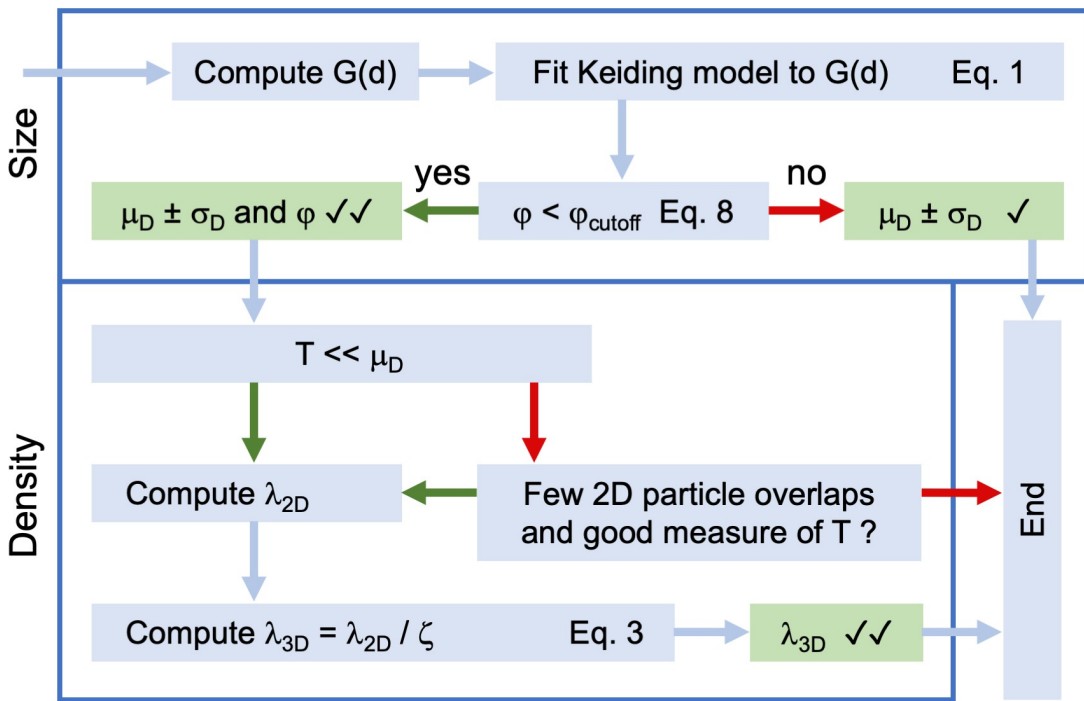

**Fig 13. Methods workflow for estimating the 3D size and density of spherical particles from their 2D projection.** Workflow diagram describing the sequence of steps for estimating F(d) ($\mu_D \pm \sigma_D$) and $\phi$ from G(d), and $\lambda_{3D}$ from $\lambda_{2D}$, where G(d) and $\lambda_{2D}$ are computed from a 2D projection of randomly distributed particles in a section of thickness T. Two check marks indicate a negligible bias and small confidence interval compared to one check mark. Final estimates of $\lambda_{3D}$ can be converted to VF using estimated $\mu_D \pm \sigma_D$ (Eq 4) and compared to that computed via the relation VF = $K_v \cdot$AF (Eq 14). Estimated $\phi$ is compared to estimated $\phi_{cutoff}$, computed via fit parameters $\mu_D$ and $\sigma_D$ (Eq 9). Green and red arrows denote 'yes' and 'no' of conditional statements.

S1 File). If uncertain of the shape of F(d), one can compare fits to distributions obtained from thick sections (T > D) where it is likely that G(d) ≈ F(d) (Fig 2C2 and S2 Fig in S1 File).

The third assumption is that the particles have a random distribution; or if the particles are clustered, then the clusters have a random distribution. It is not necessary that the spatial distribution is perfectly random (i.e. Poisson) but rather there is no order to the distribution. For example, a lattice structure (e.g. hexagonal packing) would be problematic since the particles in a 2D projection would have a discrete size distribution.

The fourth assumption is that $\phi$ is the same (fixed) for all particles. Because our 3D analysis of MFT vesicles revealed not a single value for $\phi$, but a range of $\phi$ well described by a Gaussian distribution ($CV_\phi = 0.2$), this assumption could be problematic. However, we found curve fits of the Keiding model to G(d) accurately estimated the mean of this distribution (Fig 6C3 and S8C3 Fig in S1 File). Moreover, by incorporating a Gaussian-$\phi$ model into our Monte Carlo simulations, we were able to measure the bias introduced by the fixed-$\phi$ assumption over a range of $\phi$ distributions (i.e. $\mu_\phi$) and found the bias was relatively small for $\mu_\phi < \phi_{cutoff}$ (S11 and S19B Fig in S1 File). Because vesicles are at the lower limit of resolution, we suspect the spread of our measured $\phi$ distribution, i.e. $CV_\phi = 0.2$, may be close to a worst-case scenario.

The fifth assumption pertains to thick sections: that they are perfectly transparent so that one observes 2D projections of particles at the bottom of the section as well as those of particles at the top of the section. For opaque particles with high density this assumption is clearly problematic [2, 58]. For semi-transparent particles with high density this assumption is less

problematic since overlapping projections do not necessarily preclude drawing their outline (Fig 2C1) or counting them (S17C Fig in S1 File). Yet, one can imagine a section of enough thickness that overlapping projections preclude outlining or counting the particles at the bottom of the section. Using Monte Carlo simulations, we tested the effect of such a scenario by removing particles from simulated projections if they experienced a total sum of overlaps greater than a set limit. Interestingly, results of these simulations showed only a small effect on the shape of G(d) and therefore only small biases for estimated F(d) (Fig 5C). In contrast, our simulations of opaque particles showed a larger effect on the shape of G(d), i.e. a positive skew, since overlapping projections coalesced into larger ones; yet, the end result was only small biases for estimated F(d), mostly a positive bias in $\sigma_D$ (Fig 5D). Estimates of $\lambda_{3D}$ for the same simulations of semi-transparent and opaque particles, on the other hand, showed large underestimations due to a reduction in the effective $\zeta$ (Fig 10). Hence, violation of this fifth assumption is likely to create only small biases in estimated F(d), but large biases in estimated $\lambda_{3D}$. Our analysis of GC somata in confocal images and MFT vesicles in TEM images, for example, are likely examples of this scenario.

As with other model-based stereological methods, the Keiding model has a number of core assumptions. Nevertheless, our analyses show that for approximately spherical particles these assumptions are either reasonable or can be circumvented with the correct experimental paradigm.

## Experimental considerations for optimising the detection of small caps and minimising the cap-angle limit $\phi$

Our analysis shows that estimates of F(d) and $\lambda_{3D}$ are most accurate when true $\phi < \phi_{cutoff}$ (Figs 4C and 10B). To minimise $\phi$, one needs to visually detect a wide spectrum of cap sizes within a 2D projection (i.e. sample as much of G(d) as possible). Here, we suggest experimental conditions/recommendations that could help optimise particle cap detection. First, sections should be thin (T $\approx$ 0.3 u.d.) to planar (T $\approx$ 0 u.d.) to avoid overlaps in the particle projections; the use of planar sections is particularly important for particles with a high density. Thin or planar sections can be achieved via ultrathin tissue sections (e.g. an ultramicrotome for TEM) or a high axial resolution of the microscope ($\rho_z$; e.g. confocal imaging or ET). Second, the lateral resolution of the microscope ($\rho_{xy}$) and image ($S_{xy}$) should be high with respect to the size of the particle. Third, the efficiency of particle staining and contrast between the particles and their surrounding environment should be high. Fourth, surfaces of the tissue sections should be avoided when creating images, e.g. using guard zones in the axial axis, to avoid lost caps of the nonexistent type; this scenario occurs when caps fall off the surfaces of the tissue sections or the microtome fails to transect particles during sectioning [27, 46–48]. Fifth, if the images to analyse exist within a z-stack, cap identification should be performed in a nonblind manner, i.e. by tracking particles through adjacent planes of the z-stack.

Given these considerations, our most suitable preparation for computing G(d) was that of the GC nuclei in high-resolution TEM images, where there were few lost caps and a small $\phi$ (10–20˚). For these datasets, the tissue sections were planar (T $\approx$ 0.01 u.d.) and the lateral resolution of the microscope was high with respect to the nuclei ($\rho_{xy} = 1 \times 10^{-4}$ u.d.; Table 1). Moreover, the GC nuclei were easy to identify and delineate due to their dark spotted appearance. In contrast, our analysis of GC somata in confocal images gave a larger number of lost caps and larger $\phi$ (37˚). For this dataset, the optical sections were not planar (T $\approx$ 0.3 u.d), the lateral resolution of the microscope was comparatively low ($\rho_{xy} = 0.06$ u.d.) and, due to the opaque immunolabeling and dense packing of the somata, the task of delineating between adjacent somata was more difficult compared to that of the nuclei in TEM images.

Our least successful analysis for computing a complete G(d) was that of MFT vesicles. For example, our blind analysis of MFT vesicles in ET images resulted in a large number of lost caps and a large $\phi$ = 56–63˚ (Fig 7). This result is surprising given the virtual ET sections were thin in comparison to the vesicles (estimated $\rho_z$ = 0.13 u.d.) and the relative lateral resolution ($\rho_x$ = 0.09 u.d.) was comparable to that of the confocal images of GC somata. Even when the analysis was repeated in a nonblind manner, i.e. by tracking the vesicles through multiple planes of the ET z-stack, and dramatically reducing the number of lost caps, the resulting $\phi$ (~40˚) was only on par with that of the GC somata dataset. These results suggest the small size of synaptic vesicles places them at the lower limits of cap detection with respect to microscope resolution and image contrast. Similar to our blind analysis of MFT vesicles in ET images, our analysis of MFT vesicles in high-resolution TEM images resulted in a large number of lost caps and an estimated $\phi > \phi_{cutoff}$. For this dataset, the tissue sections were thick in comparison to the vesicles (T $\approx$ 1.3 u.d.) but the lateral resolution was comparatively high ($\rho_{xy} \approx$ 0.01 u.d.). The large estimated $\phi$ for this dataset is consistent with a blind cap detection and may also reflect the difficulty of identifying caps in thick sections with a high vesicle density. The absence of vesicle caps observed in thick sections has previously been noted [55, 57]. In general, the results from our ET and TEM image analysis highlight the difficulty in computing a complete G(d) of MFT vesicles due to their small size. However, the inability to estimate $\phi$ for the MFT vesicles does not necessarily preclude obtaining an accurate estimate of their 3D size distribution (Fig 4).

## Section thickness, axial distortions and the advantages of using planar sections

To convert measures of $\lambda_{2D}$ to estimates of $\lambda_{3D}$, both 2D model-based and 3D design-based stereological methods divide $\lambda_{2D}$ by an estimate of the particle sampling space along the axial axis. For 2D methods, the axial sampling space is $\zeta$ in Eq 3, which is a function of $\mu_D$ and section thickness (T). For 3D disector methods, the axial sampling space is the distance between the reference and lookup sections (H). For both methods, sections are either physical tissue sections, in which case T and H are measures of the thickness of the tissue, or optical sections, in which case T is a measure of the axial resolution of the microscope ($\rho_z$) and H is the distance between optical sections. Hence, to obtain an accurate estimate of $\lambda_{3D}$, one must consider obtaining an accurate measure of T or H [2, 25, 26, 45, 68]. However, distortions of particle density along the axial axis of tissue sections must also be considered [2, 28, 30, 48, 69, 70]; these include uniform shrinkage and differential deformations along the axial axis, and lost caps at the surfaces of the sections (i.e. nonexistent caps).

The challenges of estimating T and avoiding axial distortions of particle density also pertain to serial 3D reconstructions. Hence, the reason 3D reconstructions should not automatically be assumed to be the 'gold standard'. For the 3D reconstructions used in this study, we avoided lost caps of the nonexistent type by avoiding the section surfaces. Moreover, the isotropic orientation of the vesicles allowed us to estimate the axial tissue shrinkage (S6B Fig in S1 File) which for EM can be considerable depending on the amount of electron beam exposure [71]. Finally, plots of vesicle density as a function of z-depth allowed us to verify that $\lambda_{3D}$ was computed within a homogeneous vesicle distribution (Fig 11A2 and 11B). Hence, estimation of $\lambda_{3D}$ from 3D reconstructions in this study was quite involved.

With these axial-axes difficulties in mind, it becomes clear that the 2D methods have a distinct advantage over the 3D methods in that they can effectively remove T from the estimation of $\lambda_{3D}$ by using planar sections. Under this condition, any bias due to an inaccurate estimate of T or uniform tissue shrinkage along the axial axis would be small. Hence, the 2D $\phi$-correction

method for estimating $\lambda_{3D}$, when applied to data derived from planar sections, is potentially the most accurate method of estimating $\lambda_{3D}$. However, planar sections come with their own challenges. First, planar sections can be technically challenging and costly to achieve, perhaps requiring TEM or ET. Second, due to the use of high-resolution microscopy, planar sections are likely to have a significantly smaller field of view, potentially creating bias if particles have a nonhomogeneous distribution. A smaller field of view, however, is not strictly prohibitive since it can be counteracted by analysing more images that have been acquired using a design-based random sampling strategy [72, 73].

Our recommendation for the use of planar sections with 2D methods is counter to previous recommendations of using thick sections [25, 26, 47]. The reasoning put forth for using thick sections is that any bias introduced by lost caps (which affects the magnitude of $\mu_D$ in Eq 3) will be relatively small in comparison to T. However, this approach will only be valid if particles have a low density, one can reliably count particles at the bottom of the sections, one has a good measure of T and one can correct for any distortions of particle density along the axial axis, as discussed above. Moreover, our analysis indicates that when sections are thick, most caps are likely to be lost, in which case $\phi$ will be close to 90° and indeterminable (Fig 9). In the case of an indeterminable $\phi$, one can use a range of $\phi$ (e.g. $\phi_{cutoff}$–90°) to estimate $\lambda_{3D}$. This would be an improvement over using the Abercrombie correction [17] that assumes no caps are lost ($\phi = 0°$) or $d_{min}$ correction [47] (or equivalent $h_{min}$ correction [42]) since $d_{min}$ is not a good measure of the lost-cap distribution when $\phi > 20°$ (S1 Appendix in S1 File).

## The cap-angle limit $\phi$ in previous studies

Although $\phi$ was never reported for Wicksell's spleen corpuscles [16], our curve-fit analysis of Wicksell's G(d), which produced an estimated F(d) that matched that of Wicksell, resulted in $\phi = 25°$ with a small fit error of 3° (S3 Fig in S1 File). In the Wicksell study, the corpuscles were large ($\mu_D = 323$ μm) in comparison to the tissue section (18 μm), in which case T ≈ 0.06 u.d. Hence, the small estimated $\phi$ for Wicksell's dataset is consistent with that of our GC nuclei in planar sections where $\phi = 20°$.

Another study of GC nuclei [66] computed a lost-caps correction factor $f = 0.978$. If one expresses $f$ as a cap-angle limit, then $\phi = \cos^{-1}(f) = 12°$. This $\phi$ is smaller than that computed for the GC nuclei in this study (20°), which is unexpected given Harvey and Napper used thin sections (T ≈ 0.3) and light microscopy compared to planar sections and TEM. It seems likely that the smaller $\phi$ of Harvey and Napper is due to their use of $d_{min}$ to estimate $f$, in which case $f$ is likely to be underestimated since $d_{min} < d_\phi$ with high probability (S1 Appendix in S1 File; Fig 3B–3D).

A previous study of vesicles within mineralized cartilage matrix estimated $\phi = 60°$ using an unfolding method that included the Keiding model [44]. In this study, matrix vesicles had an average diameter of 70 nm and the average section thickness was 28.5 nm [74], in which case T ≈ 0.4 u.d. These results are consistent with those of our blind analysis of MFT vesicles in TEM images where estimated $\phi > 50°$. Since $\phi$ was indeterminable for this blind analysis (Fig 9), the estimate of $\phi$ for the matrix vesicles is likely to suffer the same problem. To improve our estimate for MFT vesicles, we performed a nonblind cap detection in planar sections of ET z-stacks to obtain $\phi < \phi_{cuttoff}$ (Fig 6 and S8 Fig in S1 File).

The analysis of liver cell nuclei in thick sections (T ≈ 1 u.d.) by Keiding et al. [49] resulted in estimated $\phi = 70$–85° (Table 4) with a large estimation error (their Table 3). Using their estimated $\mu_D$, $\sigma_D$ and N, we computed $\phi_{cutoff} \approx 43$–54° via Eq 9, indicating their estimated $\phi > \phi_{cutoff}$, i.e. $\phi$ was indeterminable. Hence, the results of Keiding et al. also parallels that of our blind analysis of vesicles in thick sections where estimated $\phi > \phi_{cutoff}$ with a large estimation error (Fig 9).

**Table 4. Replication of the original Keiding-model fits to G(d) of human liver cell nuclei.**

| Fit | Patient 601 | | Patient 2003 | | Patient 1037 | | Units |
|---|---|---|---|---|---|---|---|
| | MLE | LSE | MLE | LSE | MLE | LSE | |
| $\mu_D$ | 6.21 | 6.19 | 6.41 | 6.39 | 7.08 | 7.07 | μm |
| $\sigma_D$ | 0.43 | 0.46 | 0.31 | 0.32 | 0.60 | 0.58 | μm |
| $\phi$ | 72.5° | 90.3° | 85.0° | 86.2° | 70.0° | 70.6° | ° |
| p1 | 0.889 | 0.888 | 0.864 | 0.866 | 0.887 | 0.874 | |
| p2 | 0.109 | 0.113 | 0.122 | 0.120 | 0.100 | 0.109 | |
| $\chi^2$ | 0.040 | 0.034 | 0.042 | 0.039 | 0.016 | 0.014 | |
| $\phi_{cutoff}$ | 48° | 46° | 54° | 54° | 43° | 44° | ° |

MLE values of Table 2 of Keiding et al. [49] where $\mu_D$ and $\sigma_D$ were computed from $f$ and $\beta$ (Eq 6), and equivalent LSE values from this study (S2 Fig in S1 File). Note, estimated $\phi > \phi_{cutoff}$, where $\phi_{cutoff}$ was computed via Eq 9 using estimated $\mu_D$ and $\sigma_D$ (n = 500). Hence, $\phi$ is considered indeterminable.

In summary, the results from this and previous studies reveal a wide range of $\phi$, highlighting the fact that $\phi$ is highly dependent on the experimental conditions. However, thin or planar sections are best for obtaining $\phi < \phi_{cutoff}$. Hence, if one wishes to accurately estimate $\phi$, which is necessary for estimating $\lambda_{3D}$ via Eq 3, then one should use thin or planar sections and, if possible, adopt the other recommendations for optimising cap detection discussed above.

## Distribution-based versus distribution-free methods and the assumption F (d) is Gaussian

Both the LSE method used in this study and the MLE method of Keiding et al. [49] are distribution-based (i.e. parametric) methods for estimating F(d) and $\phi$ from G(d) since both models assume a statistical model for F(d), e.g. a Gaussian, chi or gamma distribution. By contrast, distribution-free methods, also known as non-parametric methods or unfolding algorithms, have been extensively used in various scientific fields [2]. There are advantages and disadvantages to both methods and we refer the reader elsewhere for discussions [49, 50, 75, 76]. However, the advantages of using a distribution-based rather than distribution-free method are that it is more exact and stable and does not create implausible negative probabilities for F(d). Moreover, as we have demonstrated here, a distribution-based method allows one to accurately estimate $\phi$, which can subsequently be used to estimate $\lambda_{3D}$ from $\lambda_{2D}$ via Eq 3.

There are multiple pieces of evidence that suggest a Gaussian F(d) is valid for our samples of GC somata and nuclei and MFT vesicles. First, curve fits of the Keiding model to G(d), where F(d) is assumed to be a Gaussian distribution, showed excellent agreement for the GC somata and nuclei and MFT vesicles (S12–S16 Figs in S1 File). Moreover, repeating the same curve fits, but assuming a chi or gamma distribution for F(d), resulted in the same Gaussian solutions for F(d) (i.e. large $f$; S14C and S16C Figs in S1 File). Finally, our curve fits of Gaussian, chi and gamma distributions to F(d) computed from ET z-stacks of MFT vesicles and a TEM z-stack of GC nuclei converged to the same Gaussian solution (S7 Fig in S1 File). Interestingly, the MLE fits of Keiding et al. [49] to G(d) of liver cell nuclei, which assume a chi distribution for F(d), also converge to a Gaussian F(d) (i.e. large $f$). Our replication of the MLE fits shows that assuming either a chi or Gaussian distribution for F(d) makes little difference in the shape of the curve fits or final estimates of $\mu_D$, $\sigma_D$ and $\phi$ (Table 4; S2 Fig in S1 File).

While our study focused on F(d) described by a Gaussian model, our findings should be applicable to F(d) described by other statistical models. Moreover, our numerical solutions of the Keiding model, which have been incorporated into the latest version of the analysis package NeuroMatic [53], can be readily used as templates for creating new models that assume

other distributions for F(d). They can also be adapted to fit G(d) with multiple peaks (S2 Fig in S1 File) as previously described [49].

## Comparison of the Keiding model to the disector method

While our simulations of the disector method for estimating $\lambda_{3D}$ show no biases due to lost caps, as previously hypothesised [52], this was only true if the identification of caps on the reference and lookup sections were equally probable, since any bias due to lost caps on the reference section cancelled that on the lookup section. However, nearly all disector analyses are performed sequentially, with a blind particle detection on the reference section followed by a nonblind particle detection on the lookup section, in which case there is an increased probability of identifying caps on the lookup section (a scenario proposed by Hedreen [27]). When such an asymmetrical bias was added to our simulations, there was an underestimation error of $\lambda_{3D}$ that was large even for small degrees of bias. Using the blind-versus-nonblind bias in particle detection measured from our analysis of MFT vesicles ($\phi_{bias} = 20°$), for example, we found a large underestimation error ($\Delta\lambda_{3D} = -40\%$; Fig 12). Interestingly, an underestimation counting error of -15% for adjacent reference and lookup sections has been previously reported and attributed to lost caps [63]. To remove the blind-versus-nonblind bias in the disector method, Hedreen [27] suggests using a third section immediately below the reference section to guide the identification of caps in the reference section, i.e. a nonblind-nonblind particle detection.

Besides the potential underestimation error due to the blind-versus-nonblind bias in identifying caps, the low particle count of the disector method makes it inherently less accurate in estimating $\lambda_{3D}$ compared to the $\phi$-correction method. Our simulations indicate that, given the same number of particles per projection, the disector method is ~2 to 3-fold less accurate than the $\phi$-correction method due to the smaller counts per section. To get the same level of accuracy one would need to increase the cross-sectional area of the section (or ROI) by ~4-fold. Hence, the $\phi$-correction method for estimating $\lambda_{3D}$ is potentially more efficient and accurate than the disector method.

Finally, for particles with a high density, especially those that are touching one another (e.g. cerebellar GCs and MFT vesicles), the disector method is not recommended [18]. In this case, the $\phi$-correction method can be used in conjunction with planar sections. That said, it is important to keep in mind that the $\phi$-correction method requires an accurate estimate of $\phi$, which is not always possible for a given particle and imaging technique, and is designed for particles with a spherical geometry.

## Comparison of the Keiding model to alternative model-based and design-based stereological methods

Since the onset of design-based stereological methods in the 1980s, there has been considerable debate about the merits of these methods in comparison to the older model-based stereological methods [20, 22–30, 51, 77]. Much of the debate has stemmed from the large variability in estimates of 3D particle counts (i.e. $\lambda_{3D}$) within studies using model-based methods, i.e. the Abercrombie [17] or Floderus [42] correction, or between studies using either model-based or design-based methods. Our analysis of the two methods hopefully sheds light on the source of those variabilities: there are potential biases in both model-based and design-based counting methods. The Abercrombie correction [17], for example, is likely to underestimate $\lambda_{3D}$ since it assumes no caps are lost ($\phi = 0°$), and the Floderus $h_{min}$ correction [42], or equivalent Konigsmark $d_{min}$ correction [47], is likely to underestimate $\lambda_{3D}$ since it assumes all particles are the same size (S1 Appendix in S1 File). The use of $d_{min}$ (or $h_{min}$) to correct for lost caps is also

**Table 5. Comparison of $\lambda_{3D}$ estimated via the Keiding model, Abercrombie correction or Konigsmark ($d_{min}$) correction.**

| Particle | Image | $\phi$ | $\theta_{min}$ | $\zeta = T + \mu_D \cdot \cos\theta_{min}$ | | $\zeta = T + \mu_D$ | |
|---|---|---|---|---|---|---|---|
| | | | | $\Delta\zeta$ | $\Delta\lambda_{3D}$ | $\Delta\zeta$ | $\Delta\lambda_{3D}$ |
| GC soma | Confo | 37° | 27° | +9% | -8% | +19% | -12% |
| GC nucleus | TEM | 20° | 17° | +2% | -2% | +6% | -6% |
| MFT vesicle | ET10-z | 41° | 18° | +26% | -20% | +32% | -24% |
| MFT vesicle | ET11-z | 42° | 26° | +20% | -13% | +34% | -22% |

Confo: confocal. Mean $\phi$ from Tables 2 and 3. $\theta_{min} = \sin^{-1}(d_{min}/\mu_D)$, used in Eqs 2 and 3 as a substitute for $\phi$. Right column: Abercrombie correction [17], which assumes no lost caps ($\theta_{min} = 0°$). $\Delta = 100[X(\theta_{min}) - X(\phi)]/X(\phi)$ where $X = \zeta$ and $\lambda_{3D}$. For vesicles, comparisons are with respect to 'true' measured $\zeta$ and $\lambda_{3D}$ computed via 3D reconstructions. See S1 Appendix in S1 File.

problematic since this measure is susceptible to being an outlier (e.g. a single false-positive measurement). Moreover, those using the Abercrombie and Floderus corrections typically use approximate measures of $\mu_D$, potentially adding an additional bias. As discussed above, the design-based disector method has been deemed unbiased at its conception [52]; however, our analysis of this method has confirmed that a blind-versus-nonblind bias in particle detection, as proposed by Hedreen [27], leads to an underestimation of $\lambda_{3D}$. Moreover, biases due to inaccurate measures of section thickness (T) and z-axis tissue distortions can lead to significant biases in estimated $\lambda_{3D}$ for both design-based and model-based methods, as discussed above. Hence, it is not surprising that comparisons of estimated $\lambda_{3D}$ between model-based and design-based methods show large discrepancies. Our analysis supports the conclusion that neither model-based or design-based methods should automatically be assumed unbiased [25, 26] and both methods need to be verified/calibrated, preferably via 3D reconstructions [24, 28, 29, 51]. Yet, 3D reconstructions should not automatically be assumed to be the 'gold standard' due to potential z-axis distortions of tissue sections, as discussed above.

Here, we validated a model-based $\phi$-correction method for estimating $\lambda_{3D}$ [49, 50] that has been long overlooked since the 1970s, most likely since design-based methods have become the defacto tools in modern stereology. Results of the validation show the $\phi$-correction method can estimate $\lambda_{3D}$ with high accuracy. A high accuracy is achieved via a superior model of the lost-cap distribution, i.e. the Keiding model, that represents the mean cap-angle limit of a population of spherical particles. Moreover, use of an LSE routine (or MLE routine) allows one to use all measured 2D diameters (i.e. G(d)) to estimate $\phi$, which is a significant improvement to using $d_{min}$, a single measure that is likely to be an outlier, and also gives an accurate estimate of $\mu_D$. Comparison of estimated $\lambda_{3D}$ computed via the $\phi$-correction method to that computed via the Abercrombie and Floderus correction methods show biases (underestimations) in the latter corrections by as much as 20% for our MFT vesicle dataset (Table 5).

To estimate particle size, we used outlines to compute the cross-sectional area of a particle's projection in 2D images, which is equivalent to a high-resolution design-based point-grid method, since pixels define a grid [72]. Because particles are never perfect spheres, cross-sectional area is a better measure of 2D size than the commonly used diameter line-segment measures $d_{long}$ and $d_{short}$ (Table 6) and is consistent with methods for creating 3D reconstructions. Moreover, the cross-sectional areas can be used to estimate particle density via the VF = $K_v \cdot$AF relation (Eq 14). To estimate F(d), we computed G(d) from the equivalent diameters of the cross-sectional areas ($d_{area}$) and curve fitted Eq 1 to G(d) using an LSE algorithm. This method provides accurate measures of both $\mu_D$ and $\sigma_D$, even when true $\phi > \phi_{cutoff}$. Hence, the Keiding model offers a simple and efficient means of accurately estimating F(d). The design-based nucleator and rotator methods, on the other hand, which are used in conjunction with the

**Table 6. 2D diameter measures compared to $d_{area}$.**

| Particle | Image | $d_{long}$ | $d_{short}$ | $d_{geometric}$ | $d_{avg}$ |
|---|---|---|---|---|---|
| GC soma | Confo | +15% *** | -13% *** | -0% | +1% * |
| GC nucleus | TEM | +17% *** | -21% *** | -5% *** | -2% *** |
| MFT vesicle | TEM | +2% ** | -12% *** | -5% *** | -5% *** |
| MFT vesicle | ET11-z | +0% | -20% *** | -11% *** | -10% *** |

Confo: confocal. Percent difference (Δ) measured with respect to $d_{area}$. Significance measured via paired *t*-tests in comparison to $d_{area}$ (*$p < 0.05$

**$p < 0.01$

***$p < 0.001$). There were 50 particles per sample.

disector method [18, 21], only provide estimates of mean particle volume, are more time consuming since they require multiple line-segment measurements per particle and are potentially less accurate due to the low particle count of the disector method.

Hence, our analysis of the Keiding model demonstrates that a model-based approach to estimating the size and density of spherical particles can offer high levels of accuracy. This approach can be used in conjunction with the design-based random sampling strategies to avoid sampling biases [20, 69, 72, 73].

## Conclusions

Here, we provide the first experimental and theoretical validation of the lost-cap model for spherical particles by Keiding et al. [49], demonstrating the model estimates F(d) and ϕ from G(d) with high accuracy, so long as estimated $ϕ < ϕ_{cutoff}$ (Eq 9). The model also estimates $λ_{3D}$ from $λ_{2D}$ with high accuracy (Eq 3) and is potentially more accurate than the disector method. Our distribution-based LSE algorithm has been incorporated into the open-source software package NeuroMatic [53], making it accessible and easy to use. Our finding that density measures from thick sections were consistently smaller than those from planar sections highlights the difficulty of measuring $λ_{3D}$ for particles with a high density in thick sections. However, the necessity for using thin or planar sections is becoming less problematic with advances in transmission and scanning EM [13, 15] and super-resolution optical methods such as 3D stimulated emission depletion (STED) [78], which achieve higher axial resolutions than traditional EM and confocal microscopes. Moreover, our results should be applicable for preparations with a low particle density, where accurate estimates of size and density are likely to be achieved even with thick sections. In the future it would be interesting to combine our density and size analysis with machine learning algorithms for identifying projections of spherical particles [79, 80] as this would further speed up the analysis. Such an approach may detect lost caps more effectively than a trained researcher, leading to better estimates of F(d) and $λ_{3D}$. We hope that our validation of the lost-cap model of Keiding et al. will pave the way for the model to become more widely adopted across a wide range of research fields.

## Materials and methods

### Transmission electron microscopy of cerebellar sections

Acute sagittal sections of the cerebellar vermis (~200 μm thick) were prepared from 2 male and 2 female C57B6/J WT mice (P26–31; Charles River Germany, from the Jackson Laboratory; line #000664; RRID:IMSR_JAX:000664) in ice-cold high-sucrose artificial cerebrospinal fluid (ACSF; 87 mM NaCl, 25 mM NaHCO$_3$, 2.5 mM KCl, 1.25 mM NaH$_2$PO$_4$, 10 mM glucose, 75 mM sucrose, 0.5 mM CaCl2, and 7 mM MgCl$_2$, equilibrated with 95% O$_2$ and 5%

$CO_2$, 325 mOsm) using a Leica microsystems vibratome (VT1200S) as previously described [3]. Sections were allowed to recover in high-sucrose ACSF at 35°C for 30–45 min, then in normal ACSF (125 mM NaCl, 25 mM $NaHCO_3$, 25 mM D-glucose, 2.5 mM KCl, 1.25 mM $NaH_2PO_4$, 2 mM $CaCl_2$, and 1 mM $MgCl_2$, equilibrated with 5% $CO_2$ and 95% $O_2$) at room temperature (~23°C).

To prepare the cerebellar sections for high-pressure freezing, sections were heated to 37°C for 5–10 minutes in ACSF, then mounted into a sample 'sandwich' on a table maintained at 37°C. The sample sandwich was assembled by placing a 6 mm sapphire disk on the middle plate of a transparent cartridge system, followed by a spacer ring, the section, a drop of ACSF containing 15% of polyvinylpyrrolidone for cryoprotection and adhesion, another sapphire disk and finally a spacer ring. The sample sandwich was frozen via a Leica EM ICE high-pressure freezing machine.

Freeze substitution of the frozen samples was performed using an AFS1 or AFS4 Leica system equipped with an agitation module [81]. While in liquid nitrogen, frozen samples were transferred from storage vials to freeze-substitution vials containing 0.1% tannic acid and acetone, previously frozen in liquid nitrogen. Vials were transferred to the AFS1/AFS4 system and shaken for 22–24 hours at -90°C. Inside the AFS1/AFS4 system, samples were washed for 10 minutes in pre-chilled acetone at -90°C for 3–4 repetitions. Next, a contrasting cocktail with 2% osmium and 0.2% uranyl acetate in acetone was chilled to -90°C and added to each vial. The temperature of the vials was kept at -90°C for 7–10 hours, raised to -60°C within 2 hours (15°C/hour), kept at -60°C for 3.5 hours, raised to -30°C within 4 hours (7.5°C/hour), kept at -30°C for 3.5 hours, raised to 0°C within 3 hours (10°C/hour), kept at 0°C for ~10 min, then transferred to ice where samples were washed with acetone (3 × 10 min). Samples were transferred from the vials to glass dishes containing acetone at room temperature and inspected for intactness and proper infiltration. Samples were washed with propylene oxide (2 × 10 min) and infiltrated with Durcupan resin at 2:1, 1:1 and 1:2 propylene oxide/Durcupan resin mixtures (1 hour at room temperature). Samples were left in pure resin overnight at room temperature, embedded in BEEM capsules (Electron Microscopy Sciences, Hatfield, PA, USA) and allowed to polymerize over a second night at 100°C. Samples were trimmed with glass knives and cut into ultrathin (~60 nm) sections via a Leica EM UC7 Ultramicrotome with Diatome Histo diamond knife (6 mm, 45°). Sections were placed in formvar-coated slot grids and post-stained in 2% uranyl acetate for 10 minutes, then lead citrate for 2 minutes. Sections were imaged via a transmission electron microscope (FEI Tecnai 10, 80 kV accelerating voltage) with an OSIS Megaview III camera and Radius acquisition software.

Mice were bred in a colony maintained in the preclinical animal facility at IST Austria. All procedures strictly complied with IST Austria, Austrian, and European ethical regulations for animal experiments, and were approved by the Bundesministerium für Wissenschaft, Forschung und Wirtschaft of Austria (BMWFW-66.018/0010-WF/V/3b/2015 and BMWFW-66.018/0008-V/3b/2018).

### Electron tomography of cerebellar sections

One male C57Bl6 WT mouse (P30) was anaesthetized with ketamine and transcardially perfused with 2% paraformaldehyde and 1% glutaraldehyde in 0.1 M Na-acetate buffer for 2 min, then 2% paraformaldehyde and 1% glutaraldehyde in 0.1 M Na-borate buffer for one hour. After perfusion, the mouse's brain was dissected and 60 μm sections were cut from the cerebellar vermis. Sections were treated with 1% $OsO_4$, stained in 1% uranyl acetate, dehydrated in a graded series of ethanol and embedded in epoxy resin (Durcupan). From the embedded sections, serial sections ~200 nm thick were cut with a Leica Ultramicrotome EM UCT and

collected onto copper slot grids, where fiducial markers were introduced at both sides of the grids. Single-axis tilt series were acquired via an FEI Tecnai G2 Spirit BioTWIN transmission EM (0.34 nm line resolution) operating at 120 kV and equipped with an Eagle 4K HS digital camera (FEI, Eindhoven, The Netherlands). Tilt series were recorded between ±65˚ (with 2˚ increments between ±45˚, then 1˚ increments) at 30,000× magnification ($S_{xy}$ = $S_z$ = 0.38 nm/voxel) using FEI Xplore3D. Tomographic subvolumes were reconstructed using IMOD (RRID:SCR_003297) and exported as z-stack images ($S_{xy}$ = 1.14 nm/pixel). Two different z-stacks of MFT vesicles were analysed in this study, denoted ET10 and ET11.

The mouse was housed in the vivarium of the Institute of Experimental Medicine in a normal 12 hour/12 hour light/dark cycle and had access to water and food ad libitum. The experiment was carried out in accordance with the Hungarian Act of Animal Care and Experimentation 40/2013 (II.14) and with the ethical guidelines of the Institute of Experimental Medicine Protection of Research Subjects Committee.

## Analysis of 2D projections

From 2D images, outlines of particles (i.e. somata, nuclei and vesicles) were drawn using Fiji's freehand tool [82] (RRID:SCR_002285; https://imagej.net/Fiji) and an equivalent diameter was computed from the area of each outline ($d_{area}$ = 2(area/$\pi$)$^{1/2}$) [57, 59]. To avoid introducing bias by pooling data from multiple researchers [55], outlines were drawn by a single author (JSR). To avoid selection bias, e.g. outlining only the largest particles, an attempt was made to outline all visually identifiable particles within each selected ROI. Histograms of $d_{area}$, i.e. G(d), were computed as counts per bin, then normalised to give a probability density by dividing the count within each bin by the product of the total number of diameters and the bin size. Images and associated analyses are denoted with identification (ID) tags for the rat confocal images (R1, R5, R6) and mouse TEM images (M15, M18, M19, M21).

A numerical approximation for G(d) as defined in Eq 1 was computed via Igor Pro (RRID: SCR_000325; WaveMetrics, Portland, Oregon) where the integral in this equation was solved via an adaptive Gaussian quadrature integration routine (Integrate1D), avoiding the singularity in the denominator by setting the denominator to $1 \times 10^{-7}$ when d = $y$. The same numerical approximation was used to curve fit Eq 1 to our simulated and experimental G(d) via Igor Pro's CurveFit operation, using the Levenberg-Marquardt LSE algorithm. F(d) in Eq 1 was assumed to be a Gaussian function (Eq 5) unless specified. The estimated error of each fit parameter is reported as ±1 standard deviation (±$\sigma$). During the fit routine, parameter T was fixed at its estimated value, except where noted. The initial guess for $\phi$ was set to $\theta_{min}$, where $\theta_{min}$ = $\sin^{-1}$($d_{min}$/$\mu_D$) and $d_{min}$ is the smallest non-zero diameter bin of G(d). Initial guesses for $\mu_D$ and $\sigma_D$ were set to $\mu$ and $\sigma$ of G(d), where $\mu$ was computed as the sum of d·G(d)·h over all bins and $\sigma^2$ was computed as the sum of (d–$\mu$)$^2$·G(d)·h (h is the histogram bin size). For a small number of fits to the simulated G(d), usually for conditions of true $\phi > \phi_{cutoff}$, initial guesses had to be adjusted to get a successful fit. No parameters were constrained during the fits (e.g. 0 ≤ $\phi$ ≤ 90˚) since testing of the LSE routine using a variety of datasets showed such constraints were never active or violated during the test fits. To validate the LSE routine, the MLE fits of Keiding et al. [49] were replicated, showing nearly identical results (Table 4; S2 Fig in S1 File). Likewise, an LSE fit to Wicksell's G(d) of spleen corpuscles [16] resulted in an estimated F(d) that was nearly the same to that of Wicksell's unfolding solution (S3 Fig in S1 File).

The distribution of lost caps, L(d), was computed from G(d) via Eq 1 for T = 0 u.d. as follows: L(d, $\phi$) = G(d, $\phi$ = 0˚)–G(d, $\phi$), where G(d, $\phi$ = 0˚) and G(d, $\phi$) were computed over the range d = 0–3 u.d. and G(d, $\phi$) was normalised so that its last data point at d = 3 u.d. equaled that of G(d, $\phi$ = 0˚).

To compute $\lambda_{2D}$ from a 2D image using Fiji, a rectangular ROI was defined within a distribution of the particles of interest and two adjacent borders were designated as inclusive and the other two as exclusive (S17 Fig in S1 File). Particles were counted if they touched the inclusive borders or were completely contained within the ROI, and not counted if they touched the exclusive border [83]. $\lambda_{2D}$ was computed as the particle count ($N_{2D}$) divided by the ROI area. Using $\lambda_{2D}$, $\mu_D$ and $\phi$, $\lambda_{3D}$ was estimated via Eq 3, in which case it was important that $\lambda_{2D}$ was computed from the same image (or z-stack) from which $\mu_D$ and $\phi$ were estimated, since there was variation in $\mu_D$ and $\phi$ between sections. The following expression was used to compute the particle VF for a given $\lambda_{3D}$ and F(d):

$$VF = \lambda_{3D} \cdot max \left[ \int F(y) \frac{4\pi}{3} (1/2y)^3 dy \right] \qquad \text{Eq 4}$$

where F(d) is a PDF (Eq 5), max is the maximum value and $y$ is the variable of integration. The particle area fraction (AF) was computed by summing the area of the particle outlines located within a given ROI and dividing the summed area by the ROI area. For the confocal and TEM datasets, outlines of those particles transected by the 4 ROI borders were clipped at the borders. For the ET datasets, complete (unclipped) outlines of those particles transected by the 2 inclusive borders were included in the analysis, while none of the area of those particles transected by the 2 exclusive borders were included.

Measurements of diameters and density were analysed using NeuroMatic [53] (RRID: SCR_004186; Key Resources), an acquisition, analysis and simulation tool that runs within the Igor Pro environment. Functions for Eq 1 have been incorporated into the latest version of NeuroMatic which can be accessed via NeuroMatic's analysis Fit tab, or Igor Pro's analysis Curve Fitting graphical user interface or Global Fit package. These functions (NMKeiding-Gauss, NMKeidingChi and NMKeidingGamma) assume either a Gaussian, chi or gamma PDF for F(d) (Eqs 5–7) and can be readily used as templates for creating new Keiding models that assume other PDFs.

## Probability density functions (PDFs)

PDFs (e.g. F(d)) were described by either a Gaussian, chi or gamma distribution. The Gaussian distribution was as follows:

$$Gauss(d) = \frac{1}{\sigma(2\pi)^{1/2}} exp \left[ -\frac{1}{2} \left( \frac{d-\mu}{\sigma} \right)^2 \right] \qquad \text{Eq 5}$$

where d is the independent variable (e.g. diameter), and $\mu$ and $\sigma$ are the mean and standard deviation of the distribution. The chi distribution was the same as that used by Keding et al. [49]:

$$Chi(d) = \frac{1}{2^{\omega-1} \beta^\omega \Gamma(\omega)} d^{f-1} exp \left( -\frac{d^2}{2\beta} \right) \qquad \text{Eq 6}$$

where $f$ denotes the number of degrees of freedom, $\omega = \frac{1}{2}f$, $\beta$ the scale parameter and $\Gamma$ the gamma function. Given $f$ and $\beta$, one can compute the distribution $\mu = \gamma(2\beta)^{\frac{1}{2}}$ and $\sigma^2 = \beta(f-2\gamma^2)$ where $\gamma = \Gamma(\omega + \frac{1}{2})/\Gamma(\omega)$ (Eq 3.2 of Keiding et al.). The gamma distribution was as follows:

$$Gamma(d) = \frac{1}{\beta^f \Gamma(f)} (d-d_0)^{f-1} exp \left( -\frac{d-d_0}{\beta} \right) \qquad \text{Eq 7}$$

where $d_0$ is an x-axis offset parameter added for flexibility. Given $f$ and $\beta$, one can compute the distribution $\mu = d_0 + f \cdot \beta$ and $\sigma^2 = f \cdot \beta^2$. Note, both the chi and gamma distribution converge to a Gaussian distribution as $f \to \infty$. Hence, a large $f$ indicates a Gaussian-like distribution.

## Monte Carlo simulations

2D projections of spherical particles were simulated using D3D, a reaction-diffusion simulation package that includes a Monte Carlo algorithm for distributing non-overlapping hard spheres in arbitrary 3D geometries [5] (Key Resources). Spherical particles were randomly distributed in a rectangular cuboid using periodic boundary conditions (S1 Fig in S1 File). The xy-square dimensions of the cuboid were adjusted to accommodate the required number of particles per projection, and the z-dimension was adjusted to accommodate the required number of projections. The particle VF = 0.40 unless specified. 3D particle diameters (D) were randomly drawn from a Gaussian distribution for a given F(d). Projections in the xy-plane were computed by identifying those particles with their center point located within a given section (interior particles), and those with their center point located above or below the section (caps) at a distance dz, where dz < D/2. The 2D projected diameter (d) was computed as d = D for an interior particle and $d = (D^2 - 4dz^2)^{\frac{1}{2}}$ for a cap, derived from the trigonometric relation: $(\frac{1}{2}D)^2 = (\frac{1}{2}d)^2 + dz^2$. A particle's cap angle was computed as $\theta = \sin^{-1}(d/D)$. Other measures computed for each particle were the particle's distance from the section surface and the sum of overlaps within the xy-projection ($\Omega$; circle-circle overlaps expressed as a fraction) between the given particle and particles higher in the section. To simulate lost caps, particles were excluded from the projection if their $\theta$ was less than a fixed lower limit ($\phi$), as in the Keiding model [49]. For a few simulations, however, $\phi$ was not fixed but variable, in which case particles were assigned a $\phi$ randomly drawn from a Gaussian distribution ($\mu_\phi \pm \sigma_\phi$) for a given $CV_\phi$. To simulate the inability to observe particles deep in the section due to overlapping projections, particles were excluded from the projection if their $\Omega$ was greater than a fixed upper limit ($\psi$). To simulate the merging of circular projections for opaque particles, the xy-distance between two projections was computed according to the $\alpha$-parameter of Hilliard [58]: $\alpha = (\frac{1}{2}d_1 \cdot d_2)(4d_{12}^2 - d_1^2 - d_2^2)$, where $d_1$ and $d_2$ are the projection diameters and $d_{12}$ is the distance between the projection center points; if $-1 < \alpha < 0$, then the projections were merged into one, resulting in a decrease in projection count and increase in projection size. The procedure for merging projections began with the particle closest to the section surface, which was then merged with other particles if $-1 < \alpha < 0$. The merging procedure continued with the next particle closest to the section surface, and so on. To compute $\lambda_{2D}$, the number of particles in a projection was divided by the geometry Area$_{xy}$. Because periodic boundary conditions were used in the simulations (S1 Fig in S1 File), this $\lambda_{2D}$ is equivalent to one computed using inclusive/exclusive rectangular borders for counting (S17 Fig in S1 File).

To simulate the disector method of computing density [18, 52], particles with a Gaussian F(d) were randomly distributed within a cuboid geometry whose xy-square dimensions were adjusted to accommodate ~500 particles per projection, and the z-dimension was adjusted to accommodate 100 sections with T = 0.3 u.d. To compute the density of a given section, a projection was computed for that section (the reference projection) as well as an adjacent section of equal thickness (the lookup projection). Particles were counted if they appeared within the reference projection but not the lookup projection. $\lambda_{2D}$ was computed as particle count per Area$_{xy}$ and $\lambda_{3D} = \lambda_{2D}/T$. Simulations included parameter $\phi$, i.e. particles were excluded from a given projection if their $\theta < \phi$. To simulate a blind-versus-nonblind bias in vesicle detection, $\phi$ was defined separately for the reference section ($\phi_{ref}$) and lookup section ($\phi_{lookup}$) such that $\phi_{ref} \geq \phi_{lookup}$, with their difference (bias) defined as $\phi_{bias} = \phi_{lookup} - \phi_{ref}$.

To quantify $\phi_{cutoff}$, the estimation error $\Delta\phi$ (estimated $\phi$–true $\phi$) was computed from curve fits of Eq 1 to simulated G(d) for planar sections (T = 0 u.d.) and true $\phi$ = 10–80° (5° steps) over a range of $CV_D$ (0.04–0.17) and number of diameters (n = 200–2000; Fig 4C and S4 Fig in S1 File). $\phi_{cutoff}$ for a given $CV_D$ and n was defined as the upper limit of true $\phi$ for when $|\Delta\phi| \leq$ 5° occurs with at least 0.68 probability (i.e. at least 68 out of 100 simulation repetitions). To derive an expression relating $\phi_{cutoff}$ to $CV_D$ and n, a 2D matrix was constructed for the equivalent unit diameters of $\phi_{cutoff}$ ($d_{cutoff} = \mu_D \cdot \sin\phi_{cutoff}$, where $\mu_D$ = 1 u.d.), with the row and column dimensions defining $CV_D$ and $1/\sqrt{n}$, and bivariate polynomial with 4 dependent variables was curve fitted to the $d_{cutoff}$ matrix in Igor Pro. The inverse sine of the curve-fit solution was as follows:

$$\phi_{cutoff} \approx (1.043 - 1.534 CV_D - 0.517/\sqrt{n} - 17.106 CV_D/\sqrt{n}) \qquad \text{Eq 8}$$

where $CV_D$ is computed using true $\mu_D$ and $\sigma_D$. To investigate whether this expression can be used to test the accuracy of estimated $\phi$, $\phi_{cutoff}$ was computed using estimated $\mu_D$ and $\sigma_D$ of each simulation (rather than true $\mu_D$ and $\sigma_D$) and this 'estimated' $\phi_{cutoff}$ was compared to the corresponding estimated $\phi$. Results showed that the use of estimated $\mu_D$ and $\sigma_D$ to compute $CV_D$ translated into negative offsets in $\phi_{cutoff}$, ranging from -17° to -10° for n = 200 to 2000 diameters, respectively. To account for these offsets, diameters in the $d_{cutoff}$ matrix were adjusted to remove the offsets and the matrix was refit to the bivariate polynomial, resulting in:

$$\text{estimated } \phi_{cutoff} \approx (0.987 - 2.071 CV_D + 0.124/\sqrt{n} - 35.059 CV_D/\sqrt{n}) \qquad \text{Eq 9}$$

where $CV_D$ is computed using estimated $\mu_D$ and $\sigma_D$. In conjunction with the fit error of $\phi$, this expression was used as an accuracy test of estimated $\phi$, i.e. estimated $\phi$ was considered accurate if it was less than estimated $\phi_{cutoff}$ (S5 Fig in S1 File). For the analysis of ET z-stacks (ET10 and ET11; Fig 6 and S8 Fig in S1 File) and simulated z-stacks (S10 and S19A Figs in S1 File), *n* in Eqs 8 and 9 was reduced 3-fold to account for the reduction in sampling of F(d), a factor determined via simulations.

## 3D analysis of electron microscopy z-stacks

For the size and density analysis of MFT vesicles using ET z-stacks (ET10 and ET11) and GC nuclei using a TEM z-stack (S9 Fig in S1 File), particles were tracked and outlined through multiple planes of the z-stacks, and the equivalent xy-radii of each outline (r = ½$d_{area}$) was computed as a function of the z-stack image number ($z_\#$). The xy-diameter (D) and z-axis center point ($z_0$) of each particle was then estimated by curve fitting the particle's r-$z_\#$ relation to the following expression for an ellipse (S6A Fig in S1 File):

$$r = [(D/2)^2 - (z/E)^2]^{1/2} \qquad \text{where } z = z_\# S_z - z_0 \qquad \text{Eq 10}$$

$S_z$ is the axial sample resolution, i.e. the distance between z-stack images, which was fixed during the fit. E is an elliptical eccentricity factor where E = 1 indicates spherical dimensions and E > 1 indicates a longer diameter in the z-axis. Because tissue shrinkage in the axial axis can be significant, $S_z$ was considered unknown and estimated by adjusting its value until the mean fit E = 1.00 (S6B and S9B Figs in S1 File; $S_z$ = 0.63 and 0.53 nm for ET10 and ET11, respectively; $S_z$ = 40 nm for the TEM z-stack of nuclei). This method is based on the assumption that the vesicles and nuclei have an isotropic orientation, in which case their average diameter measured in the xy-axis should be approximately equal to that measured in the z-axis [84]. The assumption that the vesicles and nuclei have an isotropic orientation is supported by

the finding that their long axes showed no systematic orientation in the xy-plane of the ET and TEM z-stacks, consistent with a random orientation (S6C and S9C Figs in S1 File). Using fit parameters D and E, an equivalent-volume diameter ($D_{volume}$) of an ellipsoid was computed as $D_{volume} = DE^{1/3}$; however, this value was not significantly different to D (p = 0.6 and 0.5 for vesicles and 0.6 for nuclei; paired *t*-test) and is therefore not reported. To quantify lost caps, $\phi$ = $\sin^{-1}(\delta_{min}/D)$ was computed for the positive and negative pole of each particle (if the pole was interior to the z-stack) where $\delta_{min}$ was the smallest $d_{area}$ measurement near a given pole. This method of measuring $\phi$ has a discretization error that depends on the size of $S_z$ and true $\phi$ (S6D Fig in S1 File). For the vesicle ET z-stack analysis, we estimated the discretization introduced a small positive bias between the measured and true $\phi$ (estimated $\Delta\phi$ = +1 ± 1˚ for $S_z$ = 0.012 u.d. and true $\phi$ = 40˚). For the GC nuclei TEM z-stack analysis, we estimated the discretization introduced a large positive bias due to a large $S_z$ and small true $\phi$ (estimated $\Delta\phi$ = +15 ± 1˚ for $S_z$ = 0.030 u.d. and true $\phi$ = 5˚); hence, we do not report measured $\phi$ for the TEM z-stack. For ET10, one giant vesicle ~69 nm in diameter was excluded from the analysis.

To estimate the resolution of the ET z-stacks, the resolution formula of Crowther et al. [85] was used to estimate $\rho_x$ as follows:

$$\rho_x = T_{tissue} \cdot \pi / N_{tilt} \qquad \text{Eq 11}$$

where $T_{tissue}$ is the tissue thickness and $N_{tilt}$ is the number of scan tilts. For ET10, $T_{tissue}$ = 182 nm and $N_{tilt}$ = 87. Results gave $\rho_x$ = 6.6 nm. However, the Crowther formula assumes a total scan angle of 180˚, and the total scan angle was 130˚ (±65˚) for the ET scans used in this study. Hence, the Crowther formula was expressed with respect to the tilt increment ($\Delta_{tilt}$) as follows:

$$\rho_x = T_{tissue} \cdot \Delta_{tilt} \qquad \text{Eq 12}$$

where $\Delta_{tilt}$ is the total scan angle divided by $N_{tilt}$ (IMOD Tomography Guide; Key Resources). This modified formula gave $\rho_x$ = 4.7 nm. Due to the 'missing wedge effect', the resolution in the axial axis ($\rho_z$) is expected to be longer than $\rho_x$ by the following scale factor:

$$e_{xz} = \left[ \frac{(\alpha + sin\alpha \cdot cos\alpha)}{(\alpha - sin\alpha \cdot cos\alpha)} \right]^{1/2} \qquad \text{Eq 13}$$

where $\alpha$ is the maximum scan angle [86]. For this study, $\alpha$ = 65˚ in which case $e_{xz}$ = 1.4. Hence, $\rho_z = \rho_x \cdot e_{xz}$ = 6.7 nm. For ET11, where $T_{tissue}$ = 138 nm, $\rho_x$ = 3.6 nm and $\rho_z$ = 5.1 nm. Next, $\rho_z$ was estimated from experimental data by curve fitting Eq 1 to G(d) computed from the MFT vesicle analysis (Fig 6D and S8D Fig in S1 File) while fixing $\mu_D$, $\sigma_D$ and $\phi$ to their 'true' values measured from the 3D analysis (Table 2; 3D-NB) and leaving T (i.e. $\rho_z$) as the one free parameter. Results gave estimated T = 2.3 ± 0.7 nm for ET10 and -0.9 ± 0.4 nm for ET11. Hence, these analyses indicate estimated T < 7 nm of the ET z-stacks. Given the small range of estimated T, and comparatively large dimensions of the MFT vesicles, the ET analysis was simplified by assuming T = 0 nm. To test what effect this assumption might have estimates of $\mu_D$, $\sigma_D$ and $\phi$, the curve fits to G(d) were recomputed assuming T = 7 nm for ET10 (S8D Fig in S1 File) and T = 5 nm for ET11 (Fig 6D) and found $\Delta\mu_D$, $\Delta\sigma_D$ and $\Delta\phi$ were similar to those for assuming T = 0 nm: ET10 (T = 0 vs. 7 nm): $\Delta\mu_D$ = +0.5 vs. -1.4%, $\Delta\sigma_D$ = -0.3 vs. +3.9% and $\Delta\phi$ = -0.1 vs -0.9˚; ET11 (T = 0 vs. 5 nm): $\Delta\mu_D$ = -0.2 vs. -1.8%, $\Delta\sigma_D$ = +0.6 vs. +7.9% and $\Delta\phi$ = -3.1 vs. -3.2˚.

To estimate vesicle density within a cluster for ET10, it was necessary to confine the density analysis to a subregion of the original z-stack along the axial axis since the vesicle density as a function of z-depth was nonhomogeneous, being smaller at the top and bottom of the stack (Fig 11B). Within this subregion, we estimated $\lambda_{2D}$ by counting the number of outlines that

fell within a ROI ($\text{Area}_{xy}$ = 0.039 $\mu m^2$) using inclusive/exclusive borders (as in Fig 11A1) at the center of the vesicle cluster for 10 z-stack images spaced 11–16 nm apart along the axial axis, giving $\lambda_{2D}$ = 304 ± 15 $\mu m^{-2}$ (±SEM). There was an average of 12 vesicles per ROI, which is 7-fold larger than the theoretical optimal number of particles for computing density via the disector method [30]. Using the same ROI and vesicle outlines, we computed AF = 0.45 ± 0.02 and VF = 0.49 ± 0.03 (±SEM; VF = $K_v$·AF, where $K_v$ = 1.09; Eq 14). To estimate $\lambda_{3D}$, we divided the number of vesicles counted within the z-stack subregion (n = 115) by the sampling volume of interest: VOI = $\text{Area}_{xy}$·$\zeta$ = 0.013 $\mu m^3$, where $\text{Area}_{xy}$ = 0.091 $\mu m^2$ and $\zeta$ = 148.0 nm (Fig 1B; Eq 2; 3D measures: T = 113 nm, $\mu_D$ = 46.0 nm, $\phi$ = 41˚). Here, $\text{Area}_{xy}$ was the ROI area scaled to the equivalent xy-dimensions of the vesicle cluster, i.e. scale factor = (average count per image) / (average count per ROI) = 28/12 = 2.32). Results gave $\lambda_{3D}$ = 8558 $\mu m^{-3}$ with equivalent VF = 0.45 (Eq 4). Hence, the VF estimated via the 2D analysis is similar to that estimated via the 3D analysis.

To estimate vesicle $\lambda_{3D}$ for ET11 via the 'physical' disector method, a reference section with T = 12.8 nm (0.3 u.d.) was randomly located within the center of the z-stack and a corresponding adjacent lookup section with the same T was defined. Vesicles that appeared in the reference section (i.e. vesicles that had one or more of their 2D outlines from the nonblind analysis in Fig 6 appear in the reference section) but not the adjacent lookup section were counted and used to compute $\lambda_{3D}$ = count/($\text{Area}_{xy}$·T), where $\text{Area}_{xy}$ = 0.144 $\mu m^2$. This analysis resulted in ~20 vesicles per section, or $\lambda_{3D}$ ≈ 11,000 $\mu m^{-3}$. To simulate a bias between a blind reference vesicle detection and nonblind lookup vesicle detection ($\phi_{bias}$), vesicle outlines from the nonblind analysis were used for the lookup vesicle detection (mean $\phi_{lookup}$ = 42˚) and a copy of the same outlines for the reference vesicle detection, but modified to have a larger $\phi$ ($\phi_{ref}$ = $\phi_{lookup}$ + $\phi_{bias}$) by deleting the necessary number of extreme outlines from the negative and positive pole regions to achieve the desired $\phi_{ref}$. Setting $\phi_{bias}$ = 17˚ (Fig 7A) resulted in ~14 counts per section, or $\lambda_{3D}$ ≈ 7000 $\mu m^{-3}$, with estimation error $\Delta\lambda_{3D}$ = -32%. To compare these results to those of the Monte Carlo disector simulations, results of 7–9 reference sections as just described were combined to give a total of ~500 vesicles per reference section for a given $\phi_{bias}$, and an average $\Delta\lambda_{3D}$ was computed from 100 such reference sections.

## Estimation of the volume fraction of spherical particles from the area fraction of their 2D projections

The relation between the volume fraction (VF) of spherical particles and their observed area fraction (AF) in a 2D projection was derived by Weibel and Paumgartner [62] (their Eqs 13 and 37) and is as follows:

$$VF = K_v \cdot AF \qquad\qquad\qquad \text{Eq 14}$$

where

$$K_v = \frac{2m_3}{(2m_3 + 3g \cdot m_2 - 3X^2 + X^3)}$$

$$m_2 = \frac{(\mu_D^2 + \sigma_D^2)}{\mu_D^2}$$

$$m_3 = \frac{\mu_D \cdot (\mu_D^2 + 3\sigma_D^2)}{\mu_D^3}$$

$$g = \frac{T}{\mu_D}$$

$$X = 1 - cos\phi$$

where $m_2$ and $m_3$ are dimensionless moments for a Gaussian distribution and X is redefined to be a function of $\phi$ rather than $h_{min}$ (S2 Appendix in S1 File).

## Statistics

Comparisons between diameter distributions were computed via a Kolmogorov-Smirnov (KS) test (significant $p < 0.05$). Other comparisons were computed via a Student's $t$-test where noted (unpaired two-tailed equal-variance, unless specified differently; significant $p < 0.05$; F-test used to verify equal variance). Errors reported in the text and graphs (bars/shading) indicate the standard deviation ($\pm\sigma$), except in a few instances they indicate the standard error of the mean ($\pm$SEM) which is noted. Linear correlations were quantified via the Pearson correlation coefficient (r) and goodness-of-fit measure ($R^2$).

The estimation error ($\Delta$) of parameters $\mu_D$, $\sigma_D$, $\lambda_{2D}$ or $\lambda_{3D}$ was computed as the percent difference between a parameter's estimated ($\varepsilon$) and true (t) value [$\Delta = 100(\varepsilon - t)/t$], except for $\phi$, which was computed as a difference ($\Delta = \varepsilon - t$) since division by $\phi$ caused distortion at small $\phi$. For Monte Carlo simulations with multiple repetitions, the mean and standard deviation of a parameter's estimation error ($\mu_\Delta \pm \sigma_\Delta$) is referred to as the bias and (68%) confidence interval, respectively. For simulations with true $\phi < \phi_{cutoff}$, the $\Delta$-distributions were typically normal. However, for simulations with true $\phi \geq \phi_{cutoff}$, the $\Delta$-distributions were often skewed (i.e. absolute skew $> 0.5$); in this case, $\mu_\Delta$ was computed as the median of the $\Delta$-distribution and $+\sigma_\Delta$ and $-\sigma_\Delta$ were computed separately above and below $\mu_\Delta$.

## Key resources

D3D [5] https://github.com/SilverLabUCL/D3D

Igor Pro https://www.wavemetrics.com/

NeuroMatic [53] http://NeuroMatic.ThinkRandom.com

https://github.com/SilverLabUCL/NeuroMatic

Fiji [82] https://imagej.net/Fiji

IMOD Tomography Guide https://bio3d.colorado.edu/imod/doc/tomoguide.html

Figshare Repository [87] https://doi.org/10.5522/04/22117916

## Supporting information

**S1 File. Supporting S1 and S2 Appendices and S1–S19 Figs.**
(PDF)

**S1 Dataset. Supporting data analysis for estimating the size and density of GC nuclei and somata and MFT vesicles.**
(XLSX)

## Acknowledgments

We thank the IST Austria Electron Microscopy Facility for technical support, and Diccon Coyle, Andrea Lőrincz and Zoltan Nusser for their helpful comments and discussions.

## Author Contributions

**Conceptualization:** Jason Seth Rothman, R. Angus Silver.

**Data curation:** Jason Seth Rothman, Carolina Borges-Merjane, Noemi Holderith.

**Formal analysis:** Jason Seth Rothman.

**Funding acquisition:** Carolina Borges-Merjane, Peter Jonas, R. Angus Silver.

**Investigation:** Jason Seth Rothman, Carolina Borges-Merjane, Noemi Holderith.

**Methodology:** Jason Seth Rothman, R. Angus Silver.

**Project administration:** Jason Seth Rothman, Carolina Borges-Merjane, R. Angus Silver.

**Software:** Jason Seth Rothman.

**Supervision:** Peter Jonas, R. Angus Silver.

**Validation:** Jason Seth Rothman.

**Visualization:** Jason Seth Rothman.

**Writing – original draft:** Jason Seth Rothman, Carolina Borges-Merjane, R. Angus Silver.

**Writing – review & editing:** Jason Seth Rothman, Carolina Borges-Merjane, Noemi Holderith, Peter Jonas, R. Angus Silver.

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
