## [Decision Letter · Decision Letter 0]

3 Jan 2023

PONE-D-22-28847

Validation of a stereological method for estimating particle size and density from 2D projections with high accuracy

PLOS ONE

Dear Dr. Rothman,

Thank you for submitting your manuscript to PLOS ONE. After careful consideration, we feel that it has merit but does not fully meet PLOS ONE’s publication criteria as it currently stands. Therefore, we invite you to submit a revised version of the manuscript that addresses the points raised during the review process.

We look forward to receiving your revised manuscript.

Kind regards,

Shaokoon Cheng

Academic Editor

PLOS ONE

Journal Requirements:

Reviewers' comments:

Reviewer's Responses to Questions

**Comments to the Author**

1. Is the manuscript technically sound, and do the data support the conclusions?

Reviewer #1: Yes

Reviewer #2: Yes

2. Has the statistical analysis been performed appropriately and rigorously? 

Reviewer #1: Yes

Reviewer #2: I Don't Know

3. Have the authors made all data underlying the findings in their manuscript fully available?

Reviewer #1: Yes

Reviewer #2: Yes

4. Is the manuscript presented in an intelligible fashion and written in standard English?

Reviewer #1: Yes

Reviewer #2: Yes

5. Review Comments to the Author

Reviewer #1: The study presents the results of original research. In the manuscript, the author has made great efforts to validate the stereological method with TEM.

Both introduction and conclusions are well-written with sufficient and appropriate information provided to the scientific community.

The validation of innovative technique is supported by using relevant assays and mathematical model, and the data is presented in a comprehensive manner using different figures and tables with appropriate statistical analysis performed.

Reviewer #2: The manuscript evaluates the application of a pre-existing method for particles size measurement. The manuscript is well-written, however, it may require some amendments as explained below to improve before being accepted by the journal.

1. The manuscript requires a Conclusion section. Given that the authors have already mentioned about the limitations and assumptions of the particles size measurement methods focused in this research, they should discuss any further work needed to be done in the future to overcome such limitations. Or is there a need for a more conclusive method, which is not limited in terms of the particles' transparency, etc.

2. The manuscript may not be ‘validating’ the particles size measurement method, but rather ‘evaluating’ its applicability/ accuracy for certain particles studied in this work. The title and text may need to be revised from this aspect.

3. The abstract should be re-written and the main findings of the work need to be highlighted including the findings about “estimating the size and density of somata, nuclei, and vesicles in rodent cerebella”.

4. The introduction needs some amendments. Some terms such as ‘cap’, ‘section angle’, etc. could be briefly defined in the text to make the manuscript readable for readers of different backgrounds. Plus, the authors should include an explanation why they have selected certain particles for their study in the introduction.

5. The sentence in the lines 42-44 is incomplete and has to be re-written. The text should be thoroughly checked to avoid such issues.

6. PLOS authors have the option to publish the peer review history of their article (what does this mean?). If published, this will include your full peer review and any attached files.

Reviewer #1: No

Reviewer #2: No

---

## [Author Response · Author response to Decision Letter 0]

28 Feb 2023

1. The manuscript requires a Conclusion section. Given that the authors have already mentioned about the limitations and assumptions of the particles size measurement methods focused in this research, they should discuss any further work needed to be done in the future to overcome such limitations. Or is there a need for a more conclusive method, which is not limited in terms of the particles' transparency, etc.

We agree with Reviewer #2 and have added a Conclusions section to our manuscript (lines 1317-1337) including limitations of our methods, how the limitations can be overcome with technological developments and future directions for research. 

2. The manuscript may not be ‘validating’ the particles size measurement method, but rather ‘evaluating’ its applicability/accuracy for certain particles studied in this work. The title and text may need to be revised from this aspect.

We chose the word validation rather than evaluation as the former conveys the message that the method is valid, while the latter just specifies that a test was done without information on the outcome. We believe validation is justified given that we test the method both against 3D EM experimental data (now extended to include nuclei as well as vesicles; S9 Fig in S1 File; Table 3) and more generic Monte Carlo simulations.

We believe the most novel and newsworthy aspect of our analysis is the model validation, which is why we highlighted it in the title.

The words ‘validate’ and ‘verify’ have similar meaning and have been used extensively to describe the ‘verification’ of stereological methods, as cited in our manuscript on lines 160-161 [22,24,28,51]:

[22] Saper 1996: “Stereological methods have since been validated and extended by many other workers”.

[24] Geuna 2000: “Those who use a model-based approach usually have to provide a calibration study to verify the robustness of the model itself and the appropriate application of the method.”

[28] Bartheld 2001: “We need a rigorous comparison and validation of 2-D and 3-D counting and its direct comparison with counts derived from 3-D reconstruction of serial sections.”

[51] Coggeshall 1992: “Verification by serial-section methods: How can an investigator be sure estimates of neuronal or synaptic numbers using any particular method are unbiased? In my opinion, the only way is to verify the counts, which implies comparison against a standard. The obvious standard is numbers obtained from serial-section reconstructions”.

3. The abstract should be re-written and the main findings of the work need to be highlighted including the findings about “estimating the size and density of somata, nuclei, and vesicles in rodent cerebella”.

We agree that the main findings of the work were not described sufficiently in our original abstract. To address this we have restructured and reworded the abstract. We have also included the definition of ‘cap angle’ in the abstract, which was originally referred to as ‘section angle’ and included a justification of its use.

4a. The introduction needs some amendments. Some terms such as ‘cap’, ‘section angle’, etc. could be briefly defined in the text to make the manuscript readable for readers of different backgrounds.

The word ‘cap’ is defined on lines 115-117: “These latter particles whose north and south poles appear on the bottom and top of the section are known as ‘caps’.” The word is also defined in the legend of Fig 1 which is referenced in the same paragraph, in which case Fig 1 should be embedded within the Introduction of the final manuscript.

We removed ‘section angle’ from the Abstract since this phrase is not used anywhere else in the manuscript. Instead, we have used ‘cap angle’.

We updated our definition of ‘cap angle’ (θ) as follows (lines 134-135): “the half angle subtended by a particle’s cap from the particle’s center (Fig 1).” The cap angle is also defined in Fig 1 (legend and illustrations A and B) and the section “Definition of Key Terms”.

Other key terms such as section thickness (T), mean particle diameter (μD) and section z-depth (ζ) are defined in the Introduction and Fig 1, in the definition of Eqs 1-3 and in “Definition of Key Terms”.

4b. Plus, the authors should include an explanation why they have selected certain particles for their study in the introduction.

As requested, we added the following sentence to the last paragraph of the Introduction (lines 166-169):

“Synaptic vesicles in MFTs and the nuclei and somata of GCs were chosen for the analysis since they contain a wide range of particle sizes and have high densities that are problematic for design-based stereological methods.”

5. The sentence in the lines 42-44 is incomplete and has to be re-written. The text should be thoroughly checked to avoid such issues.

We believe this sentence is complete: it begins with the conjunction “Moreover”, linking this sentence with the previous sentence. We have completed a thorough proof-read of our manuscript for any other grammatical errors.

---

## [Editor Report · Decision Letter 1]

2 Mar 2023

Validation of a stereological method for estimating particle size and density from 2D projections with high accuracy

PONE-D-22-28847R1

Dear Dr. Rothman,

We’re pleased to inform you that your manuscript has been judged scientifically suitable for publication and will be formally accepted for publication once it meets all outstanding technical requirements.

Kind regards,

Shaokoon Cheng

Academic Editor

PLOS ONE

---

## [Editor Report · Acceptance letter]

7 Mar 2023

PONE-D-22-28847R1 

Validation of a stereological method for estimating particle size and density from 2D projections with high accuracy 

Dear Dr. Rothman:

I'm pleased to inform you that your manuscript has been deemed suitable for publication in PLOS ONE. Congratulations! Your manuscript is now with our production department. 

Kind regards, 

on behalf of

Dr. Shaokoon Cheng 

Academic Editor

PLOS ONE